# Fast Instrument Learning with Faster Rates

**Ziyu Wang, Yuhao Zhou, Jun Zhu**[*]
Dept. of Comp. Sci. and Tech., BNRist Center, State Key Lab for Intell. Tech. & Sys.,
Institute for AI, Tsinghua-Bosch Joint Center for ML, Tsinghua University
{wzy196,yuhaoz.cs}@gmail.com, dcszj@tsinghua.edu.cn

## Abstract

We investigate nonlinear instrumental variable (IV) regression given high-dimensional instruments. We propose a simple algorithm which combines kernelized IV methods and an arbitrary, adaptive regression algorithm, accessed as a black box. Our algorithm enjoys faster-rate convergence and adapts to the dimensionality of informative latent features, while avoiding an expensive minimax optimization procedure, which has been necessary to establish similar guarantees. It further brings the benefit of flexible machine learning models to quasi-Bayesian uncertainty quantification, likelihood-based model selection, and model averaging. Simulation studies demonstrate the competitive performance of our method.

## 1 Introduction

Instrumental variable (IV) analysis is widely used for causal inference [1–3]. Given confounded observational data, IV analysis identifies the causal effect through the use of *instruments*. Nonlinear IV regression is typically defined by the following conditional moment restrictions (CMRs):

$$\mathbb{E}(\mathbf{y} - f_0(\mathbf{x}) \mid \mathbf{z}) = 0 \quad a.s. \; [P(dz)] \tag{1}$$

where $f_0$ is the causal effect function of interest, and $\mathbf{x}, \mathbf{y}, \mathbf{z}$ denote the observed treatment, response and instrument, respectively. Similar CMR problems also appear in other applications of causal statistics and machine learning [see, e.g., 4, 5, for examples].

Starting from [6], recent works have demonstrated great promise in applying flexible machine learning (ML) methods to IV regression. Modern ML methods are appealing due to their *adaptivity* to the *informative latent structure* in data [7]: they may adapt to the low dimensionality of the informative latent features even if the observed input is high-dimensional and its signal-to-noise ratio is low. Sample complexity gaps have been established in such settings, between deep models based on neural networks (NNs) or trees, and linear models such as fixed-form kernels [8–10]. In IV regression, such adaptivity will be highly desirable when the observed instruments are high-dimensional, which is prevalent in applications such as genomics [11], and may also arise from the general desire to use structured data as instrument. Following previous work in the parametric setting [12, 13], we refer to this problem of learning informative latents in instruments as *instrument learning*. It generalizes the classical problem of instrument selection [14–16].

Comparing with standard supervised learning, IV regression is more challenging, due to the need to estimate a conditional expectation operator which defines (1). Consequently, establishing adaptivity guarantees becomes more difficult. While many recent works have demonstrated promising empirical results using deep models, they are often used as heuristics [e.g., 6], or justified with crude *slow-rate* analyses, which establish convergence rates that saturate at $\Omega(n^{-1/4})$ [e.g., 17]. This is in contrast to *faster rates* which approach $n^{-1/2}$ as the regularity of model improves. The only exception is

---

[*]JZ is the corresponding author. Code available at https://github.com/meta-inf/fil.

a minimax formulation of IV estimation [5, 18–20]. [19] establish faster rate convergence for this formulation, for models with local Rademacher complexity bounds. Though local Rademacher analysis covers many adaptive ML procedures [e.g., 10, 21], it still does not fully explain the success of modern ML approaches, with prominent alternatives including implicit regularization [22] and PAC-Bayesian analyses [23]. From a practical perspective, minimax optimization is computationally expensive, yet it cannot be avoided in the framework of [19], unless we instantiate their method with the less flexible kernel models. It requires additional hyperparameter tuning, which can be challenging in causal problems where validation is indirect and more difficult. It also prevents the use of such flexible models for *uncertainty quantification*, or *inference*, for which reliable methods have only been developed for linear nonparametric models [24–26].

This work bridges the gap between sharp theoretical guarantees and robust, practical implementation. We assume our prior knowledge about the causal effect function $f_0$ is characterized by a reproducing kernel Hilbert space (RKHS) $\mathcal{H}$, and focus on the flexibility of conditional expectation estimation. This is often possible, because the treatment variable is determined by the problem at hand, and thus has a fixed dimensionality.[2] We then present a surprisingly simple algorithm, with faster rate guarantees which in many cases match the best known in literature. The algorithm defines the conditional expectation estimates using a learned kernel, the basis of which is defined by applying adaptive regression algorithms to random draws from a Gaussian process (GP) prior. Given this learned first-stage kernel and $\mathcal{H}$, we can estimate $f_0$ using kernelized IV methods [19, 20, 27], which have closed-form solutions and can be efficiently approximated (e.g., with Nyström [19]). Our method allows easy hyperparameter tuning, and exhibits competitive performance in simulations. It accesses the regression algorithm as a black box, thus allowing for the use of any ML methods and benefits from their established theoretical guarantees. It also enables fast quasi-Bayesian uncertainty quantification [24, 28, 29] with improved flexibility.

Our algorithm connects to many ideas in literature. Most notably, it can be viewed as an infinite-dimensional generalization of [12, 13], which consider a linear outcome model with fixed dimensionality, and use ML methods to learn the *optimal instruments* [30, 31]. Our setting requires different analyses, for defining an infinite-dimensional estimation target and quantifying errors with its intrinsic complexity. Additionally, analysis of the resultant IV estimator is complicated by the ill-posedness of infinite dimensional IV models [32]. A more subtle distinction is in the choice of basis: while for finite-dimensional function spaces we can pick any set of basis (i.e., features) and apply the black-box regressor separately, in our case seemingly obvious choices of basis lead to inferior results (Appendix C.5.3). Our analysis also connects to the multi-task learning literature, to which we make technical contributions. Section 6 discusses related work in detail.

The remaining of the paper is organized as follows: Section 2 reviews background knowledge. Section 3 introduces the instrument learning problem, and reduces it to a general kernel learning problem. We solve the latter problem in Section 4, and return to IV in Section 5 with our main results. We review related work in Section 6, and present numerical experiments in Section 7.

## 2 Notations and Setup

**Notations**  Denote the joint data distribution as $P(dz \times dx \times dy)$, its marginal distributions as $P(dx), P(dz)$, etc., and their support as $\mathcal{X}, \mathcal{Z}$. For functions of observed variables (e.g., $\mathbf{x}$ or $\mathbf{z}$), $\|\cdot\|_2$ denotes the $L_2$ norm w.r.t. the respective marginal data distribution. $\|\cdot\|_\infty$ denotes the $L_\infty$ norm. We use the notation $[m] := \{1, \ldots, m\}$. Boldface $(\mathbf{x}, \mathbf{y}, \mathbf{z})$ emphasizes the denotation of random variables. For any kernel $k$, $\mathcal{GP}(0, k)$ refers to the "standard Gaussian process" [33] with zero mean, and covariance defined by $k$. $\lesssim, \gtrsim, \asymp$ represent (in)equalities up to constants; the hidden constants will not depend on any sample size. $\tilde{\mathcal{O}}(\cdot)$ denotes inequality up to logarithm factors.

**Problem Setup**  Nonparametric IV regression (NPIV) is formulated as (1) [34, 35]. Introduce the *conditional expectation operator* $E : L_2(P(dx)) \to L_2(P(dz)), f \mapsto \mathbb{E}(f(\mathbf{x}) \mid \mathbf{z} = \cdot)$, and define $g_0(z) := \mathbb{E}(\mathbf{y} \mid \mathbf{z} = z)$. We can then express (1) as a linear inverse problem:

$$E f_0 = g_0, \tag{2}$$

where we observe $g_0$ up to regression error. NPIV deviates from standard inverse problems in its need to estimate both $f_0$ and $E$. Following conventions in the two stage least square method [1], we

---

[2] Our method can also be applied when $\mathbf{x}$ contains high-dimensional exogenous covariates; see Appendix G.

refer to the modeling of $f_0$ as the *second stage*, and that of $E$ – or equivalently, that of $Ef$ for all $f$ in a hypothesis space – as the *first stage*. The following assumption describes the setup in full detail:

**Assumption 2.1** (NPIV). *(i)* The data variables $\mathbf{x}, \mathbf{y}, \mathbf{z}$ satisfy (1), and $\mathbf{y}$ is bounded by $B$. *(ii)* We observe two sets of i.i.d. samples, with matching (marginal) distributions: $\mathcal{D}_{s2}^{(n_2)} := \{(z_i, x_i, y_i) : i \in [n_2]\}$, $\mathcal{D}_{s1}^{(n_1)} := \{(\tilde{z}_i, \tilde{x}_i) : i \in [n_1]\}$.

We impose *(ii)* since such additional samples are sometimes available, as discussed in e.g. [27]. If only $n$ samples from the joint distribution are available, we can set $n_1 = n_2 = n/2$.

In the main text, we assume that our prior knowledge about $f_0$ is fully characterized by an RKHS $\mathcal{H}$, in the following sense.

**Assumption 2.2** (second stage RKHS). *(i)* $\mathcal{X}$ is a bounded subset of $\mathbb{R}^{d_x}$; the reproducing kernel $k_x$ of $\mathcal{H}$ is bounded and continuous. *(ii)* The integral operator $T_x : f \mapsto \int k_x(x, \cdot)f(x)P(dx)$ has eigenvalues $\lambda_i(T_x) \lesssim i^{-(b+1)}$, for some $b > 0$. *(iii) One of the following* holds true:

*(iii)*.a ("kernel scheme"): $f_0 \in \mathcal{H}$.

*(iii)*.b ("GP scheme"): $b > 1$; and for all $n, \exists f_n^\dagger \in \mathcal{H}$ s.t. $\|f_n^\dagger - f_0\|_{\mathcal{H}} \lesssim n^{\frac{1/2}{b+1}}$, $\|f_n^\dagger - f_0\|_2 \lesssim n^{-\frac{b/2}{b+1}}$.

In the above, *(i)* and *(ii)* are common technical assumptions: *(i)* ensures the existence of Mercer's representation, and *(ii)* is a complexity measure, with a larger value of $b$ indicating a smaller hypothesis space. *(iii)* requires $\mathcal{H}$ is correctly specified for regression; its two cases cover the different assumptions in standard RKHS-based estimation and GP modeling. *(iii)*.a is intuitive. *(iii)*.b is standard in the posterior contraction literature [36]; it roughly requires $f_0$ to be (at least) as regular as "typical" samples from $\mathcal{GP}(0, k_x)$, in the sense of [36, Theorem 2.1]. This is different from *(iii)*.a because when $\mathcal{H}$ is infinite dimensional, almost all GP samples fall out of $\mathcal{H}$ [37, 38]. Our algorithm applies to both settings, but analysis of the GP scheme requires additional effort. It is useful as it allows for quasi-Bayesian uncertainty quantification using a $\mathcal{GP}(0, k_x)$ prior [26].

The RKHS assumption has been employed in a thread of recent work [27, 20, 39, 26], and generalizes the sieve method in literature [34, 40, 4]. It is most reasonable when $\mathbf{x}$ has moderate dimensions; for example, when $f_0$ satisfies certain $L_2$-Sobolev regularity conditions, we can set $k_x$ to be a suitable Matérn kernel (Example A.1-A.2). Appendix G studies a more general setting, where $\mathbf{x}$ and $\mathbf{z}$ include additional, high-dimensional exogenous covariates. Nonetheless, the assumption will be less reasonable when the treatment variable is high-dimensional and variable selection is needed for it.

NPIV is typically an ill-posed inverse problem [35]. We now quantify the degree of ill-posedness:

**Assumption 2.3.** The operator $E$ is compact, with singular values $s_i(E) \asymp i^{-p}$, where $p > 0$.

Such *mildly ill-posed* settings [41] match our polynomial eigendecay assumption for the kernel. In the *severely ill-posed* setting where the decay of $s_i(E)$ is exponential, kernels with a similar eigendecay should be used. While the analyses of the two settings share many ideas, the Bayesian inverse problem literature typically restricts to the former for technical reasons [42, 43].

## 3 From Instrument to Kernel Learning

As we assume $\mathcal{H}$ is a correctly specified second-stage model, it remains to determine the first stage. In this section, we show that an ideal first stage model can be defined using another RKHS $\mathcal{I}$, determined by $\mathcal{H}$ and $E$. Although its kernel $k_z$ has an unknown form, we demonstrate that we can access noisy samples from $\mathcal{GP}(0, k_z)$, which, as Section 4 below shows, enable efficient learning of $\mathcal{I}$. This can be viewed as instrument learning, as $\mathcal{I}$ will only depend on the informative features in $\mathbf{z}$ (Example 3.1).

Let us first consider the GP scheme (Assumption 2.2 *(iii)*.b) which roughly requires $f_0$ to be similar to typical samples from $\mathcal{GP}(0, k_x)$. From a Bayesian perspective, an ideal prior for $Ef_0$ should match the distribution of $Ef$, for $f \sim \mathcal{GP}(0, k_x)$. This distribution is "almost equivalent" to another GP:

**Lemma 3.1** (proof in Appendix B). *Denote by $[g]_\sim$ the $L_2$ equivalence class of $g$.[3] Under Assumptions 2.1, 2.2, there exists a kernel $k_z$, with integral operator $T_z = ET_x E^\top$, s.t. for $f \sim \mathcal{GP}(0, k_x), g \sim \mathcal{GP}(0, k_z)$, $[g]_\sim$ has the same distribution as $E[f]_\sim$.*

Informally, the lemma shows that $\mathcal{GP}(0, k_z)$ matches the distribution of $Ef$. It is thus intuitive that $k_z$ could be a good choice for the first stage. The following lemma further motivates its use in the kernel scheme: its *(i)* shows that $\mathcal{I}$ fulfills the conditions in previous work [19, 27]: the restriction of $E$ on $\mathcal{H}$ has image contained in $\mathcal{I}$, and is a bounded linear map to $\mathcal{I}$. *(ii)* shows that $\mathcal{I}$, as a set of functions, cannot be made smaller while maintaining *(i)* .

**Lemma 3.2** (proof in Appendix B). *Let $\mathcal{I}$ be the RKHS defined by $k_z$. Under Assumptions 2.1, 2.2, (i) for any $f \in \mathcal{H}$, there exists $g \in \mathcal{I}$ s.t. $[g]_\sim = E[f]_\sim$; (ii) for any $g \in \mathcal{I}$, there exists $f \in \mathcal{H}$ satisfying the above. In both cases, we have $\|f\|_\mathcal{H} = \|g\|_\mathcal{I}$.*

We now demonstrate that $\mathcal{I}$ only depends on the informative latent features.

**Example 3.1** (informative latent structure). *Let $\Phi : \mathcal{Z} \to \bar{\mathcal{Z}}$ be a feature extractor that maps the observed instruments $\mathbf{z}$ to latent features $\bar{\mathbf{z}} := \Phi(\mathbf{z})$, s.t. $\mathbb{E}(f(\mathbf{x}) \mid \mathbf{z}) = \mathbb{E}(f(\mathbf{x}) \mid \Phi(\mathbf{z}))$ for all $L_2$-integrable $f$. Then we can apply Lemma 3.1, with $E$ replaced by $\bar{E} : f \mapsto \mathbb{E}(f(\mathbf{x}) \mid \bar{\mathbf{z}}) \in L_2(P(d\bar{z}))$, leading to a latent-space RKHS $\bar{\mathcal{I}}$ with kernel $\bar{k}_z$. $\bar{\mathcal{I}}$ induces the input-space RKHS*

$$\mathcal{I} := \{g = \bar{g} \circ \Phi : \bar{g} \in \bar{\mathcal{I}}\}, \quad \|\bar{g} \circ \Phi\|_\mathcal{I} := \|\bar{g}\|_{\bar{\mathcal{I}}}; \quad k_z(z, z') = \bar{k}_z(\Phi(z), \Phi(z')).$$

*The above $k_z$ satisfies Lemma 3.1-3.2.[4] Observe that $\mathcal{I}$ perfectly approximates $\{Ef : f \in \mathcal{H}\}$, but its complexity only depends on $\bar{\mathcal{I}}$. In particular, $k_z$ has the same Mercer eigenvalues as $\bar{k}_z$ (Claim B.2), the decay of which is a standard complexity measure [e.g., 44, Ch. 7].*

While $k_z$ has ideal properties, it cannot be used directly as it involves the unknown operator $E$. Instead, we need to construct an approximation from data. Our main insight is that *we can effectively draw noisy samples from $\mathcal{GP}(0, k_z)$*; as we develop in Section 4, such samples enable the approximation of $k_z$. To see how the noisy samples are obtained, consider $f \sim \mathcal{GP}(0, k_x)$. By Lemma 3.1, $g = Ef$ is $L_2$-equivalent to clean samples from $\mathcal{GP}(0, k_z)$; and we have $f(\mathbf{x}) = g(\mathbf{z}) + (f(\mathbf{x}) - (Ef)(\mathbf{z}))$, where the latter term is unpredictable given $\mathbf{z}$, and from this perspective can be viewed as noise. Thus, if we apply any regression algorithm to $f \sim \mathcal{GP}(0, k_x)$, with $\mathbf{z}$ as input, we will recover a "denoised" sample from $\mathcal{GP}(0, k_z)$, up to regression errors.

In the informative latent feature setting, optimal regression error can only be achieved by methods that adapt to such structures [8–10]. Approximating $\mathcal{I}$ with such "denoised" samples can then be viewed as a knowledge distillation procedure, which results in a compact representation of the adaptive regression algorithm. This is particularly beneficial in the NPIV setting: as discussed in the introduction, using a learned kernel eliminates the need of minimax optimization in estimation, and allows the (indirect) use of adaptive methods for uncertainty quantification.

## 4 Black-Box Kernel Learning

In this section, we address the problem of kernel learning given noisy GP samples. As our results apply to more general settings, we first state the assumptions with full generality.

**Assumption 4.1** (RKHS). *There exist a continuous function $\Phi : \mathcal{Z} \to \bar{\mathcal{Z}}$, and a reproducing kernel $\bar{k}_z$ over $\bar{\mathcal{Z}}$, s.t. (i) the random variable $\bar{\mathbf{z}} = \Phi(\mathbf{z})$ is supported on a bounded subset of $\mathbb{R}^{d_l}$; $\bar{k}_z$ is bounded. (ii) The eigenvalues of the integral operator $T_{\bar{z}} : \bar{g} \mapsto \int \bar{k}_z(\bar{z}, \cdot)g(\bar{z})P(d\bar{z})$ satisfy $\lambda_i(T_{\bar{z}}) \lesssim i^{-(\bar{b}+1)}$, for some $\bar{b} > 0$. (iii) $\bar{g} \sim \mathcal{GP}(0, \bar{k}_z)$ have finite sup norm with probability 1.*

The above assumption applies to a latent-space kernel $\bar{k}_z$. As shown in Example 3.1, $\Phi$ and $\bar{k}_z$ induce an input-space kernel $k_z$, and RKHS $\mathcal{I}$, which inherit the assumed regularity conditions. Our goal is to estimate $k_z$. This is harder than the estimation of $\bar{k}_z$, as it also involves $\Phi$.

---

[3]Recall the $L_2$ space is not a function space, and consists of equivalence classes of functions. Note that for readability, we may occasionally ignore this distinction in the main text, and use (a version of) $E$ to also denote the corresponding map between function spaces. All such denotations can be made unambiguous (Remark B.1), and all null set ambiguities in this section can be removed under mild additional assumptions (Lemma B.5).

[4]There may be multiple kernels satisfying Lemma 3.1, but they are equivalent up to null sets (Claim B.2); the ambiguity can be removed under mild assumptions (Lemma B.5).

All conditions for $\bar{k}_z$ are satisfied by Matérn kernels with a sutiable order; see Appendix A. Appendix B.1 discusses its applicability in the IV setting, where $\mathcal{I}$ is defined as in Section 3. Briefly, *(ii)* always holds for $\bar{b} \geq \max\{b, 2p-1\}$, and if $\mathcal{H}$ is further correctly specified in the sense of Assumption E.1, $\bar{b} = b + 2p$. *(i)* and *(iii)* hold under mild technical assumptions.

We will "denoise" noisy $\mathcal{GP}(0, k_z)$ samples using a regression oracle, which is specified below:

**Assumption 4.2** (regression oracle). Let $\mathcal{D}^{(n_1)} := \{(\tilde{z}_i, g(\tilde{z}_i) + e_i)\}$ be $n_1$ iid replications of the rvs $(\mathbf{z}, g(\mathbf{z}) + \mathbf{e})$, s.t. $\mathbb{E}(\mathbf{e} \mid \mathbf{z}) = 0$ and $g(\mathbf{z}) + \mathbf{e}$ has a 1-subgaussian distribution. Then the oracle returns estimator $\hat{g}_{u,n_1}$ s.t. $\mathbb{E}_{g \sim \mathcal{GP}(0, k_z)} \mathbb{E}_{\mathcal{D}^{(n_1)}} \|\hat{g}_{u,n_1} - g\|_2^2 \leq \xi_{n_1}^2$, for some $\xi_{n_1} \to 0$.

In the IV setting, we have $g = Ef \sim \mathcal{GP}(0, k_z)$, and $\mathbb{E}(\mathbf{e} \mid \mathbf{z}) = \mathbb{E}(f(\mathbf{x}) - (Ef)(\mathbf{z}) \mid \mathbf{z}) = 0$; the subgaussian condition is verified by Lemma A.4.

To provide some intuition on adaptivity, we instantiate the assumption with the DNN model in [10], and compare the resulted $\xi_n$ with fixed-form kernels:

**Example 4.1** (adaptivity of DNN oracles). *Let $\mathcal{Z} \subset \mathbb{R}^{d_z}$, $\Phi : \mathcal{Z} \to \bar{\mathcal{Z}}$ be $\beta_1$-Hölder regular, $\bar{\mathcal{I}}$ be a Matérn-$\beta_2$ RKHS, and $\beta_1, \beta_2 \geq 1$. Let the regression oracle return a $\epsilon_{opt}^2$-approximate empirical risk minimizer for the model in [10]. Then for any $\epsilon > 0$, it holds that (see Appendix C.5.1 for derivations)*

$$\xi_n = \tilde{\mathcal{O}}\Big(n^{-\frac{\beta_1}{2\beta_1 + d_z}} + n^{-\frac{\beta_2 - \epsilon}{2\beta_2 + d_l}} + \epsilon_{opt}\Big) =: \tilde{\mathcal{O}}\Big(\epsilon_{fea,n} + n^{-\frac{\beta_2 - \epsilon}{2\beta_2 + d_l}} + \epsilon_{opt}\Big) \tag{3}$$

*In the above, $\epsilon_{fea,n}$ characterizes the hardness of feature learning, i.e., learning $\Phi$. The second term characterizes that of kernelized regression given the optimal features: it matches the optimal regression rate if we* had *full knowledge about $\Phi$, or equivalently, $\mathcal{I}$, and would be attainable by kernel ridge regression (KRR) using $\mathcal{I}$.*

*As long as $\epsilon_{opt}$ is small, (3) will match the minimax rate up to logarithms. When $\beta_1/d_z < \beta_2/d_l$, the minimax rate is $\epsilon_{fea,n} \gg n^{-\beta_2/(2\beta_2 + d_l)}$, meaning that the hardness of feature learning cannot be overlooked. Otherwise, the rate $\xi_n$ nearly matches the rate given full knowledge of the unknown $\mathcal{I}$, up to the infinitesimal $\epsilon > 0$; this is realistic when, e.g., $d_z \gg d_l$ and $\Phi$ is linear ($\beta_1 = \infty$).*

*We are interested in the high-dimensional regime where $d_z \gg d_l$. In this case, fixed-form Matérn or RBF kernels could only attain the rate of $\mathcal{O}(n^{-\frac{\min\{\beta_1, \beta_2\}}{2\min\{\beta_1, \beta_2\} + d_z}})$, which can always be much worse than (3), regardless of the hardness of feature learning. This comparison suggests that fixed-form kernels cannot adapt to the latent feature structure to avoid the curse of dimensionality.*[5]

We now define the approximate RKHS. Let $\{g^{(j)} : j \in [m]\}$ be i.i.d. samples from the GP prior, and $\hat{g}_{u,n_1}^{(j)}$ be the respective estimate returned by the regression oracle, constructed from the shared dataset $\mathcal{D}^{(n_1)} = \{(\tilde{z}_i, g^{(j)}(\tilde{z}_i) + \epsilon_i^{(j)}) : i \in [n_1], j \in [m]\}$ where $\epsilon_i^{(j)}$ are subgaussian, mean-zero noise. Let $\hat{g}_{n_1}^{(j)} := \min\{C_k \log n, \hat{g}_{u,n_1}^{(j)}(\cdot)\}$, where $C_k$ is a constant determined by $\mathcal{I}$. Define $\hat{G}_n(z) := (\hat{g}_{n_1}^{(1)}(z), \ldots, \hat{g}_{n_1}^{(m)}(z))$. Our approximate RKHS is defined as

$$\tilde{\mathcal{I}} := \{g(z) = \theta^\top \hat{G}_{n_1}(z) \text{ for some } \theta \in \mathbb{R}^m\}, \quad \text{with norm } \|g\|_{\tilde{\mathcal{I}}} := \sqrt{m}\|\theta\|_2. \tag{4}$$

As $\tilde{\mathcal{I}}$ is a finite-dimensional linear space, it is an RKHS. We can check that $\|g\|_\infty \leq C_k \log n_1 \|g\|_{\tilde{\mathcal{I}}}$.

**Theoretical Results**   Under a given model, regression error is decomposed into approximation and estimation (i.e., generalization) errors. We first present the approximation error bound:

**Theorem 4.1** (proof in Appendix C.2). *Under Assumptions 4.1, 4.2, there exists a universal constant $c_r > 0$, and an event $E_{n_1}$ determined by $g^{(1 \ldots m)}$ and $\mathcal{D}^{(n_1)}$ with $\mathbb{P}_{\mathcal{D}^{(n_1)}} E_{n_1} \to 1$, on which for any $g^* \in L_2(P(dz))$, there exists $\tilde{g}^* \in \tilde{\mathcal{I}}$ s.t.*

$$\|\tilde{g}^*\|_{\tilde{\mathcal{I}}} \leq c_r \|\mathrm{Proj}_{m'} g^*\|_{\mathcal{I}}, \tag{5}$$

$$\|\tilde{g}^* - g^*\|_2 \leq c_r \|\mathrm{Proj}_{m'} g^*\|_{\mathcal{I}} (\xi_{n_1} + m^{-(\bar{b}+1)/2})\sqrt{\log n_1} + \|g^* - \mathrm{Proj}_{m'} g^*\|_2, \tag{6}$$

*where $m' = [m/2]$, and $\mathrm{Proj}_{m'}$ denotes the projection onto the top $m'$ Mercer eigenfunctions of $k_z$.*

---

[5][10] establishes formal lower bounds. Also, for small $\beta_2$, we can replace $d_z$ with a manifold dimensionality of $\mathcal{Z}$, but it can still be much larger than $d_l$.

We will use $\tilde{\mathcal{I}}$ to estimate functions on a separate dataset with $n_2$ samples. For a single regression task, the estimation error can be simply bounded as $\tilde{\mathcal{O}}(\sqrt{m/n_2})$ [45]. However, our analysis of IV estimation will require quantifying the intrinsic complexity of $\tilde{\mathcal{I}}$, which will also allow the use of a larger $m$ in practice. The following proposition provides one such result; it will be used in Section 5, to analyze IV estimation in the kernel scheme (Assumption 2.2 *(iii)*.a).

**Proposition 4.2** (proof in Appendix C.3). *Let $\tilde{\mathcal{I}}, \mathcal{D}^{(n_1)}$ be defined as above, and $\delta_{n_2}$ be the critical radius of the local Rademacher complexity of the norm ball $\tilde{\mathcal{I}}_1$ [46, Ch. 14]. On the event defined in Theorem 4.1, we have $\delta_{n_2} = \tilde{O}(n_2^{-(\bar{b}+1)/2(\bar{b}+2)} + m^{-(\bar{b}+1)/2} + \xi_{n_1})$.*

IV estimation in the GP scheme is more delicate, and requires additional analysis of $\tilde{\mathcal{I}}$, which is deferred to App. D. Before we proceed, however, we illustrate the results on a simple regression task:

**Example 4.2** (Example 4.1, cont'd). *Let $\Phi, \mathcal{I}$ be defined as before, and $\hat{g}_{n_1}^{(j)}$ be estimated by the DNN oracle. Suppose $\epsilon_{opt}$ is not greater than the other terms. Then*

   i. *Let $m = \lceil n_1^{\bar{b}/(\bar{b}+1)^2} \rceil$. On the event in Theorem 4.1, for any $g^* \in \mathcal{I}$, there exists $\tilde{g}^* \in \tilde{\mathcal{I}}$ s.t. $\|\tilde{g}^*\|_{\tilde{\mathcal{I}}} \leq c_r \|g^*\|_{\mathcal{I}}, \|\tilde{g}^* - g^*\|_2 = \tilde{O}(\|g^*\|_{\mathcal{I}} \xi_{n_1})$.*
   ii. *Let $g^* \sim \mathcal{GP}(0, k_z)$. A refined analysis, based on Corollary C.6, shows that when $m = \lceil n_1^{1/(\bar{b}+1)} \rceil$, there exists $\tilde{g}$ s.t. $\|\tilde{g}\|_{\tilde{\mathcal{I}}} \lesssim n_1^{1/2(\bar{b}+1)}, \mathbb{E}_{g^* \sim \mathcal{GP}(0, k_z)} \|\tilde{g} - g^*\|_2 = \tilde{\mathcal{O}}(\xi_{n_1})$.*

*(See Appendix C.5.2 for derivations, and another high-probability bound in the GP scheme.)*

Let $\hat{g}_n^*$ be the truncated OLS estimate using $\tilde{\mathcal{I}}$, on a dataset $\{(z_i, g^*(z_i) + e_i) : i \in [n_2]\}$ where $\mathbb{E}(e_i \mid z_i) = 0, \mathrm{Var}(e_i) \leq 1, \|g^*\|_\infty \leq B$. Then $\mathbb{E}\|\hat{g}_n^* - g^*\|_2 = \tilde{\mathcal{O}}(\|\tilde{g}^* - g^*\|_2 + B\sqrt{m/n_2})$ [45, Thm. 11.3]. When $n_1 = n_2$, the latter term is $\ll \xi_{n_2}$, and case (ii) above always matches the DNN rate. Case (i) matches the DNN rate when feature learning becomes harder ($\xi_{n_1}^2 \gtrsim n_1^{-\bar{b}/(\bar{b}+1)}$); otherwise the rate may be slightly inferior, but still approaches $n_1^{-1/2}$ as the regularity $\bar{b}$ improves.

As discussed in Example 4.1, when $d_z > d_l$, the DNN rate can outperform fixed-form kernels by a large margin. The above example demonstrates a similar superiority of the learned kernel.

## 5  Results for IV Regression

We shall use the approximate first stage $\tilde{\mathcal{I}}$ for IV regression, by plugging $\tilde{\mathcal{I}}$ and $\mathcal{H}$ to the kernelized estimators in [19, 26]; see Algorithm 1. We analyze the resulted estimators in this section, while deferring implementation details, including hyperparameter selection, to Appendix F.

---

**Algorithm 1** Kernelized IV with learned instruments.

---

**Require:** $\mathcal{D}_{s1}^{(n_1)}, \mathcal{D}_{s2}^{(n_2)}$; regression algorithm Regress; second-stage kernel $k_x$; $m \in \mathbb{N}$
1: **for** $j \leftarrow 1$ to $m$ **do**
2:     Sample $f^{(j)} \sim \mathcal{GP}(0, k_x)$
3:     $\hat{g}_{u,n_1}^{(j)} \leftarrow$ Regress$(\{(\tilde{z}_i, f^{(j)}(\tilde{x}_i)) : i \in [n_1]\})$
4: **end for** {the $m$ invocations of Regress may be replaced with a single vector-valued regression}
5: Define $\tilde{k}_z(z, z') := \frac{1}{m} \sum_{j=1}^m \hat{g}_{n_1}^{(j)}(z) \hat{g}_{n_1}^{(j)}(z')$, where $\hat{g}_{n_1}^{(j)} := \min\{\hat{g}_{u,n_1}^{(j)}(\cdot), C \log m\}$.
6: **return** KernelizedIV$(\mathcal{D}_{s2}^{(n_2)}, \tilde{k}_z, k_x)$ {See (7) below, or Appendix F for the closed-form solution}

---

Both [19] and the posterior mean estimator of [26] have the form

$$\arg\min_{f \in \mathcal{H}} \ell_{n_2}(f) + \mu \|f\|_{\mathcal{H}}^2 := \arg\min_{f \in \mathcal{H}} \max_{g \in \tilde{\mathcal{I}}} \frac{1}{n_2} \sum_{i=1}^{n_2} (y_i - f(x_i) - \kappa g(z_i)) g(z_i) - \lambda \|g\|_{\tilde{\mathcal{I}}}^2 + \mu \|f\|_{\mathcal{H}}^2. \quad (7)$$

Their difference lies in the regularization scaling, which arises from the different assumptions about $f_0$ and $\mathcal{H}$ (Assumption 2.2). Thus, we analyze the resulted two estimators separately, in Section 5.1 and Section 5.2 below. In the setting of [26] we are also able to justify the use of likelihood-based model comparison and (quasi-)Bayesian model averaging (BMA).

## 5.1 Estimation in the Kernel Scheme

[19] establish faster rate convergence of the point estimator under simple assumptions. We now provide corresponding results using our learned $\tilde{\mathcal{I}}$, by plugging in the results in Section 4.

**Proposition 5.1** (proof in Appendix E.1). *Assume Assumptions 2.1, 2.2 (kernel scheme), 4.1 and 4.2. Let $\tilde{\mathcal{I}}$ be defined by $\tilde{k}_z$ in Algorithm 1, and $\hat{f}_{n_2}$ be defined by (7), with $\kappa, \lambda, \nu$ set as in Appendix E.1. On the event defined in Theorem 4.1, we have*

$$\|E(\hat{f}_{n_2} - f_0)\|_2 = \tilde{\mathcal{O}}_p((\xi_{n_1} + n_2^{-\frac{b+1}{2(b+2)}})(1 + \|f_0\|_{\mathcal{H}}^2)). \tag{8}$$

Let us compare the result with [19] in the setting of Example 4.1. Suppose $n_1 = n_2$. [19] establishes the rate of $\mathcal{O}_p((\xi'_{n_2} + n_2^{-(b+1)/(b+2)})(1 + \|f_0\|_{\mathcal{H}}^2))$, where $\xi'_{n_2}$ is comparable with a first-stage regression rate established from local Rademacher analysis. For the DNN model in Example 4.1, we have $\xi'_n = \tilde{\Theta}(\xi_n) = \tilde{\mathcal{O}}(\epsilon_{fea,n} + n^{-\bar{b}/2(\bar{b}+1)})$ in the general case,[6] or $\xi'_n = \tilde{\mathcal{O}}(\epsilon_{fea,n} + n^{-(\bar{b}+1)/2(\bar{b}+2)})$ assuming additional regularity (Remark C.3). Thus, the two rates are equivalent if $\epsilon_{fea}$ is sufficiently large, meaning that the difficulty of feature learning cannot be ignored. Otherwise, [19] may be better if $\bar{b} < b + 1$; this is a somewhat narrow range, as $\bar{b} \geq \max\{2p - 1, b\}$. Recall that our method is more appealing computationally: directly instantiating [19] with DNNs requires solving a minimax problem similar to (7), while for our learned kernel (7) can be evaluated in closed form.

We can also compare (8) with kernelized IV using a fixed-form first stage. In the above setting, its best rate is also provided by [19], and is dominated by the kernel regression error in Ex. 4.1 which, as we discussed, can be much worse than $\xi_n$. Our improved rate has been made possible by the fact that we are approximating a first-stage model with optimal adaptivity (Ex. 3.1), at a rate that is also adaptive to the informative latent structure (Ex. 4.1).

In summary, *our algorithm combines the best of both worlds*: it maintains the sharp guarantees of adaptive models, and the simplicity of kernel methods.

## 5.2 Quasi-Bayesian Estimation and Uncertainty Quantification

Quasi-Bayesian analysis enables efficient uncertainty quantification for NPIV, without introducing extra risks of model misspecification [24, 28]. [26] studies a quasi-Bayesian posterior constructed from (7) and a $\mathcal{GP}(0, k_x)$ prior. It is defined through the Radon-Nikodym derivative w.r.t. the prior, $(d\Pi(\cdot | \mathcal{D}_{s2}^{(n_2)})/d\Pi)(f) \propto e^{-n_2 \ell_{n_2}(f)}$. For kernel first-stage models, the quasi-posterior can be evaluated in closed form (App. F). For general models, however, it is entirely unclear if approximate inference can be possible, since for any parameter $f$, evaluation of $\ell_n(f)$ involves solving a separate optimization problem. Our kernel learning algorithm enables the (indirect) use of such models.

Analysis of (quasi-)Bayesian procedures is more challenging, partly because of the weaker regularization. Thus, [26] introduced additional assumptions. Our analysis is further complicated by a different assumption on $\mathcal{I}$, and approximation errors in $\tilde{\mathcal{I}}$, which necessitate further assumptions. App. E.2 discusses these assumptions in detail. For simplicity, we state the result in a "rate-optimal" case:[7]

**Theorem 5.2** (posterior contraction; proof in App. E.3). *Assume Asms. 2.1, 2.2 (GP scheme), 2.3, 4.1, 4.2, D.1, D.2, E.1, E.2. Let $n_1$ be s.t. $\xi_{n_1}^2 \log n_2 + n_1^{-(b+2p)/(b+2p+1)} \lesssim n_2^{-1}$, and $m \asymp n_1^{1/(b+2p+1)}$. Let $\Pi_{n_1}(\cdot | \mathcal{D}_{s2}^{(n_2)})$ be defined in (52) in appendix. Then, with $\mathcal{D}_{s1}^{(n_1)}$-probability $\to 1$, we have*

$$\mathbb{E}_{\mathcal{D}_{s2}^{(n_2)}} \Pi_{n_1}(\{f : \|f - f_0\|_2 \geq M\bar{\epsilon}_{n_2}\} | \mathcal{D}_{s2}^{(n_2)}) \to 0,$$

$$\mathbb{E}_{\mathcal{D}_{s2}^{(n_2)}} \Pi_{n_1}(\{f : \|E(f - f_0)\|_2 \geq M\bar{\delta}_{n_2}\} | \mathcal{D}_{s2}^{(n_2)}) \to 0,$$

*where $\bar{\delta}_{n_2} = \tilde{\mathcal{O}}(n_2^{-(b+2p)/2(b+2p+1)})$, $\bar{\epsilon}_{n_2} = \tilde{\mathcal{O}}(n_2^{-b/2(b+2p+1)})$.*

---

[6]With some abuse of notation, we also use $\tilde{\mathcal{O}}$ to hide the infinitesimal deterioration of the polynomial order.

[7]Classical NPIV lower bounds continue to hold given full knowledge of $E$ [47], so the rate $n_2^{-b/2(b+2p+1)}$ is minimax optimal irrespective of $n_1$. In our setting, it is certainly desirable to improve the dependency on $n_1$, and our restriction is only employed to simplify proof. In simulations we find the choice of $n_1 = n_2$ works well.

Theorem 5.2 immediately implies Theorems 5, 6 in [26] for our $\tilde{\mathcal{I}}$, with the extra logarithms, as their proofs do not involve the first stage. Those results establish Sobolev norm rates, and justify uncertainty quantification by lower bounding the magnitude of posterior spread.

In the nonparametric Bayes literature, contraction results like Theorem 5.2 often lead to the justification of marginal likelihood-based model selection and averaging. This is also the case here. The key ingredient is the following marginal quasi-likelihood bound:

**Corollary 5.3** (proof in Appendix E.4). *In the setting of Theorem 5.2, for some $C > 0$ we have*

$$\mathbb{P}_{\mathcal{D}_{s2}^{(n_2)}}(C^{-1}n_2^{\frac{1}{b+2p+1}}\log^{-\frac{6}{b}}n_2 \leq -\log\Pi_{n_1}(\mathcal{D}_{s2}^{(n_2)}) \leq Cn_2^{\frac{1}{b+2p+1}}\log^2 n_2) \to 1.$$

This result allows the comparison of a finite number of second-stage RKHSes. Of particular interest is the comparison between *power RKHSes* (Defn. A.1), which often have intuitive interpretations: e.g., for a Matérn RKHS $\mathcal{H}$ and $\gamma \in (2/(b+1), 1)$, the power RKHS $\mathcal{H}^\gamma$ is equivalent to lower-order Matérn RKHSes ([48]; Ex. A.3). We can verify that such $\mathcal{H}^\gamma$ fulfills the assumptions about $\mathcal{H}$. Thus, provided the other assumptions continue to hold, Corollary 5.3 will hold for all such $\mathcal{H}^\gamma$, with $b+1$ replaced by $\gamma(b+1)$, showing the marginal likelihood has a different asymptotics. Consequently, it establishes asymptotically valid comparison between such models, and justifies the use of BMA.

Analysis of more general settings requires additional effort: NPIV is an inverse problem, and we anticipate the subtleties of model selection in nonparametric inverse problems. For example, analyses are usually restricted to the selection of $\gamma$ [43, 49, 50], and the $\gamma > 1$ case requires additional assumptions [49].[8] In the IV setting, it should also be noted that valid model comparison requires a good approximation to $E|_{\mathcal{H}}$, since otherwise the quasi-likelihood becomes less meaningful at any finite sample size. The same intuition applies to other model selection procedures [18, 20, 27] based on the estimated violation of (1). When the approximation cannot be guaranteed, it could be preferable to stick to the prior knowledge and fix a conservative choice for $\mathcal{H}$.

# 6 Related Work

**Multi-Task Learning** Our Example 4.2 can also be viewed as quantifying sample efficiency improvements in multi-task learning, if we view the GP prior draws as the labeling functions for a handful of diverse training tasks, which share the representation $\Phi$. This general idea is not new: starting from [51, 52], a line of recent work establishes similar results. Most related is [52, Sec. 5], which assumes a fixed-dimensional linear model for $\bar{g}$, and an adaptive $\Phi$ with metric entropy bounds. We assume more general models for both components, and do not require different training tasks to have separate inputs. On the flip side, [52] allows for non-iid training tasks. [52, Sec. 6] investigated infinite-dimensional $\bar{g}$, but established a slow rate. We are unaware of any work that established fast-rate convergence for infinite-dimensional top-level models, or used ML models as a black box. Both aspects may be interesting for multi-task learning, and are necessary for instrument learning.

**Causal Statistics** The double machine learning framework [53] also uses black-box ML models to estimate certain nuisance parameters in the model. While the operator $E$ can be viewed as a nuisance parameter, the structure of the NPIV problem is quite different: [54, p. 8] noted that it is very unclear if such a view can be helpful for NPIV estimation; consistent with their remarks, we have also been unable to cast our problem into the double ML framework. Note that double ML has been applied to semiparametric estimation and inference for IV [53, 55–57], which are orthogonal to our goal.

It has long been known [30, 31] that under a linear outcome model $f_0(\mathbf{x}) = \theta^\top \mathbf{x}$, using $\mathbb{E}(\mathbf{x} \mid \mathbf{z})$ as instrument leads to $\sqrt{n}$-consistent estimates. Our Section 3 can be viewed as an infinite-dimensional generalization of this observation.[9] Given high-dimensional instruments and a parametric outcome model, there is a large body of literature on efficient inference; see [12] for a review. As we move to nonparametric models, we focus on estimation which becomes much more challenging, in the spirit of [54]. Still, we have provided qualitative characterization for uncertainty estimates in Section 5.2.

For the use of ML for nonlinear IV, [6] studied a heuristic application of NNs. We discussed the minimax formulation in introduction. [5, 26] justified the use of NNs with the respective neural

---

[8]We do not cover it here for brevity, noting that it is well-understood in inverse problem settings [43, 49].

[9]As noted in [27], when $f_0 \in \mathcal{H}$ for some RKHS $\mathcal{H}$, the first stage should model $\mathbb{E}(f(\mathbf{x}) \mid \mathbf{z})$ for all $f \in \mathcal{H}$, as opposed to merely modeling $\mathbb{E}(\mathbf{x} \mid \mathbf{z})$. Note that [27] did not study the optimal choice of the first stage.

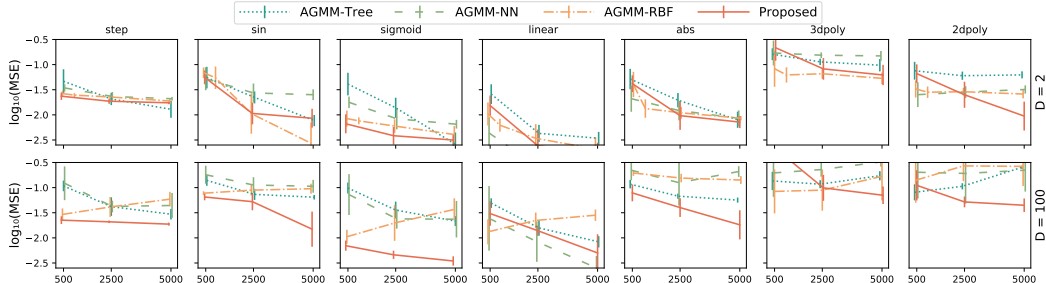

Figure 1: Predictive performance: test MSE vs sample size $n_1 = n_2$ for all method, and $D \in \{2, 100\}$. Full results are in App. H.2.

Table 1: Runtime results for all methods in the predictive experiment, for $N = 2500, D = 100$.

| Method | AGMM-Tree | AGMM-NN | AGMM-RBF | Proposed |
|---|---|---|---|---|
| Runtime / s | $1374 \pm 418$ | $303 \pm 16$ | $6.7 \pm 0.1$ | $25.9 \pm 5.6$ |

tangent kernels (NTKs) which, like other fixed-form kernels, cannot adapt to the informative latent structure [8, 9]. [39, 58] investigated the combination of an NN-based second stage and a linear first stage, which could be useful in complementary scenarios. [17] considered feature learning in both stages, but only established a slow rate; as the authors noted, it is also unclear if their algorithm reliably minimizes the empirical risk.

For model selection in the setting of Section 5.2, [59] prove the validity of bootstrap-based selection for the sieve estimator [34]. [39, 60] investigate the use of marginal likelihood for two different kernel-based IV estimators: [60] establish a crude $-1/4 \log n$ upper bound for the log marginal likelihood, and [39] connect it to the empirical leave-one-out validation error. Neither result fully justifies model selection as our Corollary 5.3. For kernelized IV models, [20, 27, 61] proposed validation statistics for comparing *a finite number of* first stage models.

## 7 Simulation Study

Our main simulation setup is adapted from [18, 19]; Appendix H.4 presents additional experiment on the demand dataset [6, 17]. In [18, 19], the observed $\mathbf{z}, \mathbf{x}, \mathbf{y}$ are generated by

$$\bar{\mathbf{z}} \sim \text{Unif}[-3, 3]^{\lfloor \frac{D}{2} \rfloor}, \; \mathbf{z} = h(\bar{\mathbf{z}}), \; \mathbf{u} \sim \mathcal{N}(0, 1), \; \mathbf{x} := \bar{\mathbf{z}}_1 + \mathbf{u} + \mathbf{e}_x, \; \mathbf{y} := (f_0(\mathbf{x}) + \mathbf{u} + \mathbf{e}_y - \mu)/\sigma,$$

where $\mathbf{u}$ is the confounder, $\mathbf{e}_x, \mathbf{e}_y \sim \mathcal{N}(0, 0.1^2)$ are independent noise, and the constants $\mu, \sigma$ standardize $\mathbf{y}$. We consider three choices for $h$: **(i)** $D = 2$, $h$ is the identity function; this recovers the setup in previous work, and quantifies the hardness of the NPIV problem given true instruments. **(ii)** $\dim \mathbf{z} = D \in \{40, 100\}$, $h$ is a three-layer DNN; this simulates a feature learning scenario, and ensures the observation has a low signal-to-noise ratio $(O(1/D))$. **(iii)** $h$ maps $\bar{\mathbf{z}}_1$ to a MNIST [62] or CIFAR-10 [63] image with matching label; the MNIST setting also appeared in previous work.

We consider two choices for $f_0$: **(i)** a widely used collection of functions (e.g., $\sin, \text{abs}$) in [18]. **(ii)** $f_0 \sim \mathcal{GP}(0, k_x)$. (ii) ensures the correct specification of $\mathcal{H}$ and allows us to focus on the first stage.

We use a DNN as the black-box learner, and a RBF kernel for $\mathcal{H}$, with bandwidth determined by marginal likelihood (72). We set $N_1 = N_2 \in \{500, 2500, 5000\}$. We defer setup details and full results to Appendix H, and summarize the findings below:

**Hyperparameter Selection (App. H.1)** We first study hyperparameter selection in instrument learning. We set $f_0 \sim \mathcal{GP}$, $D \in \{2, 40, 100\}$. We find our validation statistics (71) always correlates with the counterfactual MSE $\|\hat{f}_n - f_0\|_2^2$, and that across a large hyperparameter space, trained DNNs always outperform first-stage models based on RBF kernels, or randomly initialized DNNs.

**Predictive Performance (App. H.2)** For $h$ defined as in (i-ii), we compare our algorithm with [19, AGMM], instantiated with kernel, tree and NN models. As shown in [19], the baselines have

Table 2: Test MSE, radius and estimated coverage rate of the $90\%$ $L_2$ credible ball (CB), and the average coverage of pointwise $90\%$ credible interval (CI), for $f_0 \sim \mathcal{GP}, D = 100$. For the CB coverage rate estimate, we report its $95\%$ Wilson score interval [64]. Full results are in App. H.3.

| Method | $n_1 = n_2$ | Test MSE | 90% CB. Rad. | 90% CB. Cvg. | 90% CI. Cvg. |
|---|---|---|---|---|---|
| Proposed | 500 | $.097 \pm_{.065}$ | $.201 \pm_{.025}$ | $.923$ [.888, .948] | $.915 \pm_{.123}$ |
| | 2500 | $.035 \pm_{.024}$ | $.074 \pm_{.008}$ | $.917$ [.880, .943] | $.908 \pm_{.127}$ |
| | 5000 | $.024 \pm_{.016}$ | $.049 \pm_{.004}$ | $.920$ [.884, .946] | $.905 \pm_{.134}$ |
| RBF | 500 | $.431 \pm_{.192}$ | $.240 \pm_{.036}$ | $.187$ [.147, .235] | $.640 \pm_{.191}$ |
| | 2500 | $.176 \pm_{.089}$ | $.175 \pm_{.023}$ | $.517$ [.460, .573] | $.822 \pm_{.136}$ |
| | 5000 | $.126 \pm_{.072}$ | $.156 \pm_{.019}$ | $.660$ [.605, .711] | $.855 \pm_{.143}$ |

competitive performance on this setup; the latter two models also enjoy adaptivity guarantees. A representative subset of results are plotted in Fig. 1: our method has stable performance as we move to high dimensions, demonstrating excellent adaptivity. In contrast, fixed-form kernels fail to identify the informative features. AGMM-tree and AGMM-NN also have deteriorated performance as $D$ increases, despite their theoretical guarantees, presumably due to the challenges in optimization. Table 1 reports the run time of all methods in this experiment. As we can see, our method is more efficient than both adaptive baselines.

For image-based $h$, we compare with AGMM-NN and [39], which report the best results in the MNIST setting. Our method outperforms both baselines.

**Uncertainty Quantification (App. H.3)**  Table 2 presents a subset of results for $f_0 \sim \mathcal{GP}$. Comparing with a fixed-form RBF first stage, our method produces sharper credible intervals, which also have better coverage. For $f_0$ specified as in [18], we experiment with BMA over a grid of RBF kernels, and present visualizations in Appendix H.3. We find that when the model is more correctly specified, BMA produces conservative uncertainty estimates which are nonetheless informative. However, when all models are severely misspecified (e.g., when $f_0$ is a step function), we cannot expect model-based uncertainty estimates to have ideal coverage.

**Exogenous Covariates (App. H.4)**  We evaluate the extended algorithm in Appendix G on the demand dataset [6], which is a widely used simulation design with high-dimensional exogenous covariates. As shown in the appendix, our extended algorithm has competitive performance.

## Acknowledgements

This work was supported by the National Key Research and Development Program of China (2020AAA0106302); NSFC Projects (Nos. 62061136001, 62076145, 62076147, U19B2034, U1811461, U19A2081, 61972224), Beijing NSF Project (No. JQ19016), BNRist (BNR2022RC01006), Tsinghua Institute for Guo Qiang, and the High Performance Computing Center, Tsinghua University. J.Z is also supported by the XPlorer Prize.

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
