# Appendix

## Table of Contents

## A  Background and Technical Lemmas

**Kernels**   The two following lemma applies to our $\mathcal{H}$ satisfying Assumption 2.2, but we will also apply them to the RKHS $\bar{\mathcal{I}}$ defined in Section 4.[10] In the latter case, $x, \mathcal{X}$ should be replaced by $\bar{z}, \bar{\mathcal{Z}}$; and as discussed in the main text, the results will transfer to the RKHS $\mathcal{I} = \{\bar{g} \circ \Phi : \bar{g} \in \mathcal{I}\}$.

**Lemma A.1** (Mercer's representation). *Let $\mathcal{H}$ be any RKHS with kernel $k_x$ s.t. $\int P(dx)k_x(x,x) < \infty$. Then*

    *i. $\mathcal{H}$ can be embedded into $L_2(P(dx))$, and the natural inclusion operator $\iota_x : \mathcal{H} \to L_2(P(dx))$ and $\iota_x^\top$ are Hilbert-Schmidt; the map $T_x : f \mapsto \int P(dx)k_x(x,\cdot)f(x)$ defines a positive, self-adjoint and trace-class operator; $T_x = \iota_x \iota_x^\top$.*

---

[10]$\mathcal{I}$ may not necessarily satisfy the requirement in Lemma A.1 (iv), but a weaker version always holds; see Appendix B.

ii. $T_x$ *has the decomposition*

$$T_x f = \sum_{i \in I} \mu_i \langle \bar{e}_i, f \rangle_2 \bar{e}_i,$$

*where the index set $I \subset \mathbb{N}$ is at most countable, and $\{\bar{e}_i\}$ is an orthonormal system in $L_2(P(dx))$.*

iii. *There exists an orthogonal system $\{e_i : i \in I\}$ of $\mathcal{H}$ s.t. $[e_i]_\sim = \sqrt{\lambda_i} \bar{e}_i$.*

iv. *If $k_x$ is additionally bounded and continuous, $\{e_i : i \in I\}$ will define a Mercer's representation whose convergence is absolute and uniform.*

*Proof.* [65, Lemma 2.3, 2.2 (for i), 2.12 (for ii-iii), Corollary 3.5 (for iv)]. □

The following material on power spaces are adapted from [26], which collected them from [65, 48].

**Definition A.1** (power space, embedding property). Let $\mathcal{H}$ be an RKHS with Mercer's representation $\{(\lambda_i, \varphi_i) : i \in \mathbb{N}\}$. For $\gamma \geq 1$, the *power space* $[\mathcal{H}]^\gamma \subset L_2(P(dx))$ is defined as

$$[\mathcal{H}]^\gamma = \left\{ [f]_\sim := \sum_{i=1}^\infty a_i [\varphi_i]_\sim : \|[f]_\sim\|_{[\mathcal{H}]^\gamma}^2 := \sum_{i=1}^\infty \lambda_i^{-\gamma} a_i^2 < \infty \right\}.$$

We say $\mathcal{H}$ satisfies an *embedding property* with order $\gamma$ if $[\mathcal{H}]^\gamma$ is continuously embedded into $L_\infty(P(dx))$, denoted as

$$[\mathcal{H}]^\gamma \hookrightarrow L_\infty(P(dx)). \tag{EMB}$$

Clearly, $\mathcal{I}$ and $\bar{\mathcal{I}}$ will satisfy (EMB) with the same order.

**Lemma A.2.** *Under* (EMB), *(i) the function space*

$$\mathcal{H}^\gamma = \left\{ f := \sum_{i=1}^\infty a_i \varphi_i : (\lambda_i^{-\gamma/2} a_i)_{i \in \mathbb{N}} \in \ell_2(\mathbb{N}) \right\},$$

*with a similarly defined norm, will be an RKHS (a "power RKHS") with a bounded kernel; (ii) the kernel has a pointwise convergent Mercer representation $\{(\lambda_i^\gamma, \varphi_i) : i \in \mathbb{N}\}$. (iii) We have the interpolation inequality*

$$\|f\|_\infty \lesssim \|f\|_{\mathcal{H}}^\gamma \|f\|_2^{1-\gamma}, \quad \forall f \in \mathcal{H}. \tag{9}$$

*Proof.* [66, Theorem 5.5 (for i)], [65, Theorem 3.1 (for ii), Thm. 5.3 (for iii)]. □

The embedding property is stronger when $\gamma$ can be chosen to be smaller. The following example shows that for Matérn kernels, we can choose the best posible $\gamma$:

**Example A.1** (Matérn kernels and Sobolev regularity). *Let $\mathcal{X}$ be a bounded open set in $\mathbb{R}^d$ with a smooth boundary, $P(dx)$ have its Lebesgue density bounded from both sides, and $\mathcal{H}$ be the Matérn-$\alpha$ RKHS. Then*

i. *$\mathcal{H}$ is norm-equivalent to the $L_2$-Sobolev space $W^{\alpha + \frac{d}{2}, 2}$ [67, Example 2.6].*

ii. *Its Mercer eigenvalues decay at $\lambda_i \asymp i^{-(1 + \frac{2\alpha}{d})}$, and it satisfies (EMB) for all $\gamma > (1 + \frac{2\alpha}{d})^{-1}$ [48, Section 4].*

*Among kernels with the same eigendecay, this is the best possible $\gamma$ [68].*

We now provide some intuition on the "GP scheme" approximation condition, Assumption 2.2 *(iii)*.b:

**Example A.2.** *For any $\mathcal{H}$ satisfying the eigendecay assumption, simple calculation shows that any $f_0 \in [\mathcal{H}]^{b/(b+1)}$ satisfies Assumption 2.2 (iii).b [26, Lemma 23]. If we are further in the setting of Example A.1, $f_0$ will satisfy Assumption 2.2 (iii).b if $f_0 \in W^{\alpha, 2}$ [69, Chapter 7].*

**Lemma A.3.** *Let $\mathcal{I}$ satisfy Assumption 4.1. Then for all $g \in \mathcal{I}, m \in \mathbb{N}$,*

$$\|\mathrm{Proj}_m g\|_{\mathcal{I}} \leq \|g\|_{\mathcal{I}}, \quad \|g - \mathrm{Proj}_m g\|_2^2 \lesssim \|g\|_{\mathcal{I}}^2 m^{-(\bar{b}+1)}, \tag{10}$$

*where the constant hidden in $\lesssim$ only depends on $\mathcal{I}$.*

*Proof.* [11] Let $\{(\lambda_i, \varphi_i)\}$ be the Mercer eigendecomposition, so that $\{\sqrt{\lambda_i}\varphi_i\}$ constintute a countable ONB for $\mathcal{I}$ (Lemma A.1). Thus, the RKHS norm bound holds, and

$$\|g - \text{Proj}_m g\|_2^2 = \sum_{j=m+1}^{\infty} \langle f, \sqrt{\lambda_i}\varphi_i \rangle_{\mathcal{I}}^2 \|\sqrt{\lambda_i}\varphi_i\|_2^2 \leq \|g - \text{Proj}_m g\|_{\mathcal{I}}^2 \cdot \lambda_m \|\varphi_m\|_2^2 \lesssim \|g\|_{\mathcal{I}}^2 m^{-(\bar{b}+1)}.$$

$\square$

**Gaussian Measure and Gaussian Process**

**Definition A.2** (Gaussian measure, [70]). Let $(\mathbb{B}, \|\cdot\|)$ be a Banach space, $W \sim \mu$ be a Borel measurable map. $\mu$ is a *Gaussian measure* if for any $b^* \in \mathbb{B}^*$, the pushforward measure $b^*_{\#\mu}$ is normally distributed.

A Gaussian measure defines a bilinear form on $\mathbb{B}^*$: $q(f, g) = \mathbb{E}f(W)g(W)$. When $\mathbb{B}$ is additionally a Hilbert space, $q$ will correspond to a bilinear form on $\mathbb{B}$, denoted as $\Lambda$. We then introduce the notation $W \sim N(0, \Lambda)$, meaning that for all $l \in \mathbb{B}$, the random variable $\langle l, W \rangle_H \sim \mathcal{N}(0, \langle l, \Lambda l \rangle_{\mathbb{B}})$.

**Lemma A.4** (Borell-TIS, [71], Proposition I.8). *Let $W$ be any mean-zero Gaussian process defined on a Banach space $\mathbb{B}$, and $\|\cdot\|$ denote any Banach norm. If $\|W\| < \infty$ a.s., it will hold that*

$$\mathbb{P}(|\|W\| - E\|W\|| > x) \leq 2e^{-x^2/(2\sigma^2(W))} \quad \forall x > 0,$$

*where $\sigma(W) := \sup_{b^* \in \mathbb{B}^*: \|b^*\|=1} \sqrt{\mathbb{E}_W[b^*(W)^2]}$ is less than the median of $\|W\|$.*

In the following, $C^\beta$ denotes the Hölder space of order $\beta$ on $\mathcal{X}$.

**Example A.3** (Matérn processes). *Let $k_x$ be a Matérn-$\alpha$ kernel, $\mathcal{X}$ be as in Example A.1, $\underline{\alpha} < \alpha$ be any positive number. Then there exists a modification of $\mathcal{GP}(0, k_x)$ which always has finite $C^{\underline{\alpha}}$ norm [38, p. 2104].*

**Miscellaneous Results**

**Definition A.3** (entropy number). Let $H, J$ be Banach spaces, $A \subset J$ be a bounded set. For all $i \in \mathbb{N}$, the $i$-th entropy number is defined as

$$e_i(A, J) = \inf\{\epsilon > 0 : N(A, \|\cdot\|_J, \epsilon) \leq 2^i\},$$

where $N$ denotes the covering number. Further, let $T : H \to J$ be any bounded linear operator. Then the $i$-th entropy number of the operator $T$ is defined using the image of the unit-norm ball $H_1$ under $T$:

$$e_i(T) = e_i(T(H_1), J).$$

The following singular value inequality will be frequently used, both to $s_j(AB)$ and the $j$-th largest eigenvalue $\lambda_j(ABB^\top A^\top) = s_j(AB)^2$:

**Lemma A.5** (72, Problem III.6.2). *Let $A, B$ be any two operators, $\|\cdot\|$ denote the operator norm, and $s_j$ denote the $j$-th largest singular value. Then*

$$s_j(AB) \leq \min\{\|B\| s_j(A), \|A\| s_j(B)\}. \tag{11}$$

# B Deferred Proofs: Function Spaces

*Remark* B.1 (versions of $E$). Conditional expectations are only defined up to $P(dz)$-null sets. As $\mathbf{x}$ is supported on a bounded open subset of $\mathbb{R}^{d_x}$, there exists a regular conditional probability $\mu$, which defines a version of conditional expectation [73]

$$\text{for all square integrable } f, \quad \mathbb{E}(f(\mathbf{x}) \mid \mathbf{z}) = \int \mu(dx, \mathbf{z}) f(x) \ a.s. \ [P(dz)].$$

Throughout the work, we work with the above version of conditional expectation.[12] It represents a linear operator between spaces of functions, denoted as

$$(E_r f)(z) := \int \mu(dx, z) f(x).$$

---

[11]Similar result has been stated in [26]. We restate the proof to drop some unnecessary assumptions.

[12]The choice of $\mu$ is only unique up to a $P(dz)$-null set; we fix an arbitrary version to define $E_r$. What matters to us is the fact that $E_r$ is defined with a regular conditional probability, so that (12) always holds.

As $\mu(\cdot, z)$ is a probability measure for all $z \in \mathcal{Z}$, we now have

$$\|E_r f\|_\infty \leq \|f\|_\infty. \tag{12}$$

Our focus in this work is in estimation; thus, in the main text and other sections of the appendix, we will abuse notation, and use $E$ to also refer to $E_r$ for readability. In this section, however, we make the distinction clear for full clarity.

The following claim is well-known. Note that by requiring $[f]_\sim$ to be a Gaussian measure in $L_2(P(dx))$, we are requiring our Gaussian process to possess a possibly richer $\sigma$-algebra than e.g., the version returned by the Kolmogorov extension theorem. However, they will induce the same marginal distributions for $f(X)$, and the resulted estimators. See Definition A.2, and e.g. van Zanten and van der Vaart [33], Stuart [74] for an accessible review of related issues.

**Claim B.1.** *Let $\mathcal{X}$ be a bounded subset of $\mathbb{R}^d$, $k_x$ be a reproducing kernel on $\mathcal{X}$, s.t. $\mathbb{E}_{P(dx)} k_x(x, x) < \infty$. Let $[f]_\sim \sim N(0, C)$ be a Gaussian random element on $L_2(P(dx))$ (Definition A.2), s.t. the marginal distributions of $f$ match $\mathcal{GP}(0, k_x)$ s.t. $[f]_\sim$. Then $C$ equals the integral operator of $k_x$.*

*Proof.* Stuart [74, p. 538]. □

Note that for our $k_x$, the integral operator $T_x = \iota_x \iota_x^\top$ (Lemma A.1).

*Proof for Lemma 3.1.* By definition of Gaussian measure (A.2) and Claim B.1, we have $E[f]_\sim \sim N(0, ET_x E^\top)$, so it suffices to construct a $k_z$ with integral operator $ET_x E^\top$.

By Lemma A.1 (i) and boundedness of $E : L_2(P(dx)) \to L_2(P(dz))$, the operator $E\iota_x : \mathcal{H} \to L_2(P(dz))$ is Hilbert-Schmidt. Thus, the operator $E\iota_x \iota_x^\top E^\top$ is trace-class, and we can invoke Steinwart and Scovel [65, Theorem 3.10], which shows the existence of an RKHS $\mathcal{I}$ with a measurable reproducing kernel $k_0$, such that

i. The integral operator of $k_0$ equals $E\iota_x \iota_x^\top E^\top$.

ii. For appropriate choices of $e_i^z$ s.t. $[e_i^z]_\sim$ diagonalizes $E\iota_x \iota_x^\top E^\top$, $\{\sqrt{\lambda_i^z} e_i^z : i \in \mathbb{N}, \lambda_i^z > 0\}$ form an ONB of $\mathcal{I}$.

iii. $k_0(z, z') = \sum_{i \in I} \lambda_i^z e_i^z(z) e_i^z(z')$, $\mathbb{E}_{P(dz)} k_0(z, z) < \infty$.

Combining (i, iii) and Claim B.1 above completes the proof. □

Observe the statement (ii) above shows that, the RKHS $\mathcal{I}$ satisfies

$$\mathcal{I} = \left\{ \sum_{i \in I} b_i \sqrt{\lambda_i^z} e_i^z : (b_i) \in \ell_2(I) \right\}, \quad \left\| \sum_{i \in I} b_i \sqrt{\lambda_i^z} e_i^z \right\|_\mathcal{I} = \|(b_i)\|_{\ell_2(I)}, \tag{13}$$

where $I \subset \mathbb{N}$ denotes an index set which is at most countable.

*Proof for Lemma 3.2.* The operator $\iota_x^\top E^\top E \iota_x$ is also trace-class. Let $\{(\lambda_i^z, e_i^z) : i \in I\}$ be defined as in the proof of Lemma 3.1, so that $\{e_i^x := (\lambda_i^z)^{-1/2} \iota_x^\top E^\top [e_i^z]_\sim : i \in I\} \subset \mathcal{H}$ diagonalizes $\iota_x^\top E^\top E \iota_x$. Then for any $f \in \mathcal{H}$, it holds that

$$\infty > \sum_{i \in I} \langle f, e_i^x \rangle_\mathcal{H}^2 = \sum_{i \in I} (\lambda_i^z)^{-2} \langle f, \iota_x^\top E^\top E \iota_x e_i^x \rangle_\mathcal{H}^2 = \sum_{i \in I} (\lambda_i^z)^{-1} \langle E \iota_x f, [e_i^z]_\sim \rangle_2^2.$$

Comparing with (13), we can see that for any function $g$ s.t. $[g]_\sim = E\iota_x f$, the RHS shows that $g \in \mathcal{I}$, and equals $\|g\|_\mathcal{I}^2$.

Conversely, for any $g \in \mathcal{I}$, the sequence $\{(\lambda_i^z)^{-1/2} \langle [g]_\sim, [e_i^z]_\sim \rangle_2 : i \in I\}$ must be in $\ell_2$. Additionally, $\{e_i^x\}_{i \in I}$ is an ONS of $\mathcal{H}$, so the limit

$$\sum_{i \in I} (\lambda_i^z)^{-1/2} \langle [g]_\sim, [e_i^z]_\sim \rangle_2 e_i^x =: f$$

must exists in $\mathcal{H}$, and $\|f\|_{\mathcal{H}} = \|g\|_{\mathcal{I}}$ holds by the above display. Similarly, it holds that

$$E[f]_\sim = \sum_{i \in I} (\lambda_i^z)^{-1/2} \langle [g]_\sim, [e_i^z]_\sim \rangle_2 E\iota_x e_i^x = \sum_{i \in I} \langle [g]_\sim, [e_i^z]_\sim \rangle_2 [e_i^z]_\sim = [g]_\sim.$$

This completes the proof. $\qquad\square$

We prove the following claim, made in Example 3.1.

**Claim B.2.** *(i) Let $k_z, k_z'$ satisfy Lemma 3.1. Then $\mathbb{E}_{\mathbf{z}, \mathbf{z}' \sim P}(k_z(\mathbf{z}, \mathbf{z}') - k_z'(\mathbf{z}, \mathbf{z}'))^2 = 0$. (ii) Let $\bar{k}_z, k_z$ be defined as in the example. Then the non-zero Mercer eigenvalues of $\bar{k}_z, k_z$ coincide.*

*Proof.* *(i)* Both kernels are $L_2(P(dz) \otimes P(dz))$ bounded; the claim thus follows from the isometry between $L_2$-bounded kernels and their (Hilbert-Schmidt) integral operators, and the fact that both kernels have the same integral operator. *(ii)* Let $\{(\lambda_i, [e_i^{\bar{z}}]_\sim)\}$ denote the eigendecomposition of the integral operator $T_{\bar{k}}$. Following the definitions we find that $\{[e_i^{\bar{z}} \circ \Phi]_\sim\}$ are eigenfunctions of $T_z$, with the same eigenvalues; and it is not possible for $T_z$ to have additional non-zero eigenpairs. $\qquad\square$

## B.1 Further Regularity Properties of $\mathcal{I}$

**Eigendecay** Recall the proof of Lemma 3.1 invokes [65], and leads to the following results:

**Claim B.3.** *The kernel $k_z$ is measurable, satisfies $\mathbb{E}_{P(dz)} k_z(z, z) < \infty$, and has integral operator equal to $E\iota_x \iota_x^\top E^\top$.*

The last statement bounds the decay of the Mercer eigenvalues: using Assumption 2.2, 2.3, and (11), we immediately find $\lambda_i^z = \lambda_i(E\iota_x \iota_x^\top E^\top) \lesssim i^{-\max\{b+1, 2p\}}$. We further have the following:

**Claim B.4.** *Under Assumption E.1, it holds that $\lambda_i^z \lesssim i^{-(b+2p+1)}$.*

*Proof.* Let $\{[\psi_i]_\sim : i \in \mathbb{N}\}$ be the Mercer eigenfunctions of $\mathcal{H}$, s.t. $\{\sqrt{\lambda_i(\iota_x \iota_x^\top)}\psi_i : i \in \mathbb{N}\}$ form an ONB of $\mathcal{H}$ (Lemma A.1). By the min-max theorem for eigenvalues [e.g., 75, Theorem 3.2.4],

$$\lambda_i(E\iota_x \iota_x^\top E^\top) = \lambda_i(\iota_x^\top E^\top E \iota_x) = \inf_{V \subset \mathcal{H}, \dim V = i-1} \sup_{e \perp V, \|e\|_{\mathcal{H}} = 1} e^\top \iota_x^\top E^\top E \iota_x e$$

$$\leq \|E\iota_x(\sqrt{\lambda_i(\iota_x \iota_x^\top)}\psi_i)\|_2^2 \lesssim i^{-(b+1)} \|E[\psi_i]_\sim\|_2^2 \lesssim i^{-(b+2p+1)}.$$

The last inequality follows by the link condition. $\qquad\square$

**Bounded Kernel and GP Prior Draws** To establish boundedness of the kernel $k_z$ and (a version of) $g \sim \mathcal{GP}(0, k_z)$, we need the following additional assumptions:

(A.I) A version of $\mathcal{GP}(0, k_x)$ takes value on a separable subspace $\mathbb{B} \subset L_\infty(P(dx))$.

(A.II) The operator $E_r$ (Remark B.1) maps $\mathbb{B}$ to a space of continuous, bounded functions on $\mathcal{Z}$.

Note (A.I) will hold given our Assumption E.2, in which case we can take the subspace as a power RKHS $\mathcal{H}^\alpha$; see Steinwart [66]. As discussed around that assumption, for some valid choice of $\alpha$, $\mathcal{H}^\alpha$ should match the regularity of the second-stage RKHS assumed in previous work on kernelized IV, so such a boundedness assumption matches previous work.

(A.II) will hold if we assume $E_r$ maps $f \in \mathcal{H}^\alpha$ to another RKHS over $\mathcal{Z}$, with a continuous, bounded reproducing kernel. Such an RKHS is often assumed in previous work; note that it does not have have the optimal regularity. Alternatively, the assumption can also be fulfilled by the assumption that $P(dx \times dz)$ have a continuous Lebesgue density and the marginal density $p(z)$ does not vanish.

We now establish the following lemma. It shows the $\mathcal{I}$ defined in Sec. 3 fulfills the conditions in Asm. 4.1. It also shows that by defining $k_z$ with $E_r$ as below, we can remove the null set indeterminacies in Sec. 3: all possible $k_z$'s have the same integral operator (i.) and are thus equivalent up to null sets (Claim B.2 *(i)*), yet they are shown to be continuous (iii.).

**Lemma B.5** (bounded kernel and GP draws). *Let $f$ be a Gaussian measure with marginal distributions matching $\mathcal{GP}(0, k_x)$. Let $E_r$ be defined in Remark B.1. Then under (A.I),*

*i. There exists a kernel $k_z$, s.t. $E_r f \sim \mathcal{GP}(0, k_z)$, with integral operator equaling $ET_x E^\top$.*

*ii. $k_z$ is bounded, and there exists a version of $g \sim \mathcal{GP}(0, k_z)$ which always has a finite sup norm.*

*iii. If additionally (A.II) holds, $k_z$ will be continuous, and its RKHS $\mathcal{I}$ will also satisfy Lemma 3.2.*

*Proof.* ii.: Observe that by (12), for all $z_0 \in \mathcal{Z}$, the linear functional $e_{z_0} : f \mapsto (E_r f)(z_0)$ is bounded on $L_\infty$. As $k_x$ is a bounded kernel, we have $\|\cdot\|_{\mathcal{H}} \geq (\sup_{x \in \mathcal{X}} k_x(x, x))^{-1} \|\cdot\|_\infty$; thus, $e_{z_0}$ is bounded on $\mathcal{H}$, and its Riesz representer $h_{z_0} \in \mathcal{H}$ has norm $\|h_{z_0}\|_{\mathcal{H}} \leq \sup_{x \in \mathcal{X}} k_x(x, x) =: \sigma_x$. Moreover, for any $\{z_1, z_1, \ldots, z_m\} \subset \mathcal{Z}$ and $a \in \mathbb{R}^m$, the linear map $f \mapsto \sum_{j=1}^m a_j e_{z_j}(f)$ is also bounded on $L_\infty$, and thus $\mathcal{H}$; and its representer $h_{\{z_j, a_j\}}$ has $\mathcal{H}$-norm bounded by $\|a\|_2 \sigma_x$. By our assumptions on $f \sim \mathcal{GP}(0, k_x)$, we can invoke Ghosal and Van der Vaart [71, Definition 11.12-11.13, Lemma 11.14] which show that, for all $m, \{z_i\} \in \mathcal{Z}^m, a \in \mathbb{R}^m$ and $f \sim \mathcal{GP}(0, k_x)$,

$$\sum_{j=1}^m a_j (E_r f)(z_j) \sim \mathcal{N}(0, \|h_{\{z_j, a_j\}}\|_{\mathcal{H}}^2), \quad \text{where } \|h_{\{z_j, a_j\}}\|_{\mathcal{H}} \leq \|a\|_2 \sigma_x,$$

meaning that $E_r f$ distributes as a GP. Its reproducing kernel [71, Definition 11.12] $k_z$ satisfies

$$\sup_{z \in \mathcal{Z}} k_z(z, z) \leq \sigma_x. \tag{14}$$

We also have

$$\|E_r f\|_\infty \overset{(12)}{\leq} \|f\|_\infty < \infty. \tag{15}$$

As $E_r f$ is a version of $\mathcal{GP}(0, k_z)$, (14) and (15) prove the second claim.

i.: Let $k_z$ be defined as above, $T_x$ denote the integral operator of $k_x$. We claim $k_z$ has integral operator $ET_x E^\top$: this is because by Claim B.1 applied to $f \sim \mathcal{GP}(0, k_x)$, we have $[f]_\sim \sim N(0, T_x)$; moreover, we have $E_r f \sim \mathcal{GP}(0, k_z)$, and $[E_r f]_\sim = E[f]_\sim \sim N(0, ET_x E^\top)$ by definition of Gaussian measure, and the boundedness of $E$. Thus, by Claim B.1 applied to $k_z$, its integral operator is $ET_x E^\top$.

iii.: The continuity of $k_z$ follows from (A.II), and the fact that continuous GP samples must correspond to an RKHS with a continuous kernel [33, Example 8.1]. Now it remains to re-establish Lemma 3.2.

Following the proofs for Lemma 3.1, 3.2, let $\{(\lambda_i^z, [e_i^z]_\sim) : i \in I\}$ be a set of eigenfunctions for $ET_x E^\top$. By Lemma A.1, $\{[e_i^z]_\sim : i \in I\}$ then determine an ONS $\{\sqrt{\lambda_i^z} e_i^z : i \in I\}$ for $\mathcal{I}$. It suffices to show this is an ONB, after which we can follow the proof of Lemma 3.2. But as $k_z$ is bounded and continuous, the inclusion operator $\iota_z : \mathcal{I} \to L_2(P(dz))$ is now injective [44, Exercise 4.6]; thus, by Steinwart and Scovel [65, Theorem 3.1], $\{\sqrt{\lambda_i^z} e_i^z : i \in I\}$ is an ONB. This completes the proof. $\quad\square$

# C  Deferred Proofs: Kernel Learning

## C.1  Notations and Preliminary Observations

Let $m' = [m/2]$, $\text{Proj}_{m'}(\cdot)$ denote the projection onto the top $m'$ Mercer basis $\psi_1, \ldots, \psi_{m'}$, and the respective Mercer eigenvalues be $\lambda_i$. Then there exists i.i.d. normal rvs $\bar{e}_{ij}$ s.t.

$$\begin{pmatrix} \text{Proj}_{m'} g^{(1)} \\ \ldots, \\ \text{Proj}_{m'} g^{(m)} \end{pmatrix} = \begin{pmatrix} \bar{e}_{11} & \ldots & \bar{e}_{1m'} \\ \ldots & \ldots & \ldots \\ \bar{e}_{m1} & \ldots & \bar{e}_{mm'} \end{pmatrix} \begin{pmatrix} \sqrt{\lambda_1} \psi_1 \\ \ldots \\ \sqrt{\lambda_{m'}} \psi_{m'} \end{pmatrix}.$$

Denote the $m \times m'$ matrix as $\Xi$. Introduce the notation $\hat{G} := (\hat{g}_{n_1}^{(1)}; \ldots; \hat{g}_{n_1}^{(m)})$, and $\bar{G}, \Psi$ so that we can write the above as

$$\hat{G} + (\bar{G} - \hat{G}) = \Xi \Psi.$$

**Note our slight abuse of notation**: throughout the proof, we will use $\hat{G}$ to refer to both the vector-valued function as in the main text, and a "column of $m$ functions", i.e., a linear map from $L_2(P(dx))$ (or other suitable function spaces) to $\mathbb{R}^m$. We define the norm

$$\|\hat{G}\|_2^2 := \sum_{i=1}^m \|\hat{g}_{n_1}^{(i)}\|_2^2 = \int \sum_{i=1}^m (\hat{g}_{n_1}^{(i)}(z))^2 P(dz),$$

and similarly for $\bar{G}, \Psi$; and use notations such as $\hat{G}_j$ to refer to the $j$-th row of this "column vector of functions", so e.g., $\hat{G}_j$ refers to $\hat{g}_{n_1}^{(j)}$.

As $\Xi$ is a $m \times m'$ Gaussian random matrix, where $m' = [m/2]$, we have the high-probability singular value bounds

$$c'' \sqrt{m} \leq s_{min}(\Xi) \leq s_{max}(\Xi) = s_{max}(\Xi^\top) \leq c' \sqrt{m}, \tag{16}$$

where $c' > c'' > 0$ are universal constants [76]. Thus,

$$s_{max}((\Xi^\top \Xi)^{-1} \Xi^\top) \leq (c'')^{-2} c' m^{-1/2} =: c_r m^{-1/2}. \tag{17}$$

And for all $j \leq m'$,

$$s_j((\Xi^\top \Xi)^{-1} \Xi^\top) \overset{(11)}{\geq} s_j(\Xi^\top) \|\Xi^\top \Xi\|^{-1} \geq s_{min}(\Xi) \|\Xi^\top \Xi\|^{-1} \overset{(16)}{\geq} c'' \sqrt{m} \cdot (c' \sqrt{m})^{-2} =: c_r' m^{-1/2}. \tag{18}$$

(By the other inequality in (11), the remaining singular values are zero.)

We will condition on the event defined in (16) throughout all proofs below. On this event, we have $\Psi = (\Xi^\top \Xi)^{-1} \Xi^\top \bar{G}$, and we can define the transformed feature map

$$\hat{\Psi} := (\Xi^\top \Xi)^{-1} \Xi^\top \hat{G}.$$

## C.2 Proof for Theroem 4.1

Recall the notations and observations in Appendix C.1.

As we will work with the truncated estimators $\hat{g}_{n_1}^{(j)}$, we first show that the truncation does not affect the error rate. By Borell's inequality, we have, for any $B > 1$

$$\mathbb{P}(\max_{i \in [m]} \|g^{(i)}\|_\infty \geq B) \leq m \, \mathbb{P}(\|g^{(i)}\|_\infty \geq B) < C_1 m e^{-C_2 B^2}. \tag{19}$$

Thus the above event has high probability for $B = 4 C_2^{-1/2} \sqrt{\log m}$. On this event, the truncated estimator will have the same $L_2$ error as the original estimators, leading to

$$\mathbb{E}_{\mathcal{D}^{(n_1)}, G} \|\hat{G} - G\|_2^2 := \mathbb{E}_{\mathcal{D}^{(n_1)}, G} \sum_{j=1}^m \|\hat{g}_{n_1}^{(j)} - g^{(j)}\|_2^2 = m \xi_n^2.$$

And by Markov's inequality we have, with high probability

$$\|\hat{G} - G\|_2^2 \leq m \xi_n^2 \log n. \tag{20}$$

We further restrict to the event on which the above holds. Now, for any $g^* \in L_2(P(dx))$, let

$$\bar{e}_* := (\lambda_1^{-1/2} \langle g^*, \psi_1 \rangle_2, \ldots, \lambda_{m'}^{-1/2} \langle g^*, \psi_{m'} \rangle_2) \in \mathbb{R}^{m'}$$

so that $\|\bar{e}_*\|_2 = \|\mathrm{Proj}_{m'} g^*\|_{\mathcal{I}}, \bar{e}_*^\top \Psi = \mathrm{Proj}_{m'} g^*$. Then for $\tilde{g}^* := \bar{e}_*^\top \hat{\Psi}$, we have

$$\|\tilde{g}^*\|_{\tilde{\mathcal{I}}} = m^{1/2} \|\Xi(\Xi^\top \Xi)^{-1} \bar{e}_*\|_2 \overset{(17)}{\leq} c_r \|\bar{e}_*\|_2 = c_r \|\mathrm{Proj}_{m'} g^*\|_{\mathcal{I}}, \tag{21}$$

$$\|\tilde{g}^* - g^*\|_2 \leq \|\bar{e}_*^\top (\hat{\Psi} - \Psi)\|_2 + \|g^* - \mathrm{Proj}_{m'} g^*\|_2$$

$$\leq c_r \|\mathrm{Proj}_{m'} g^*\|_{\mathcal{I}} (\xi_n + m^{-(\bar{b}+1)/2}) \sqrt{\log n} + \|g^* - \mathrm{Proj}_{m'} g^*\|_2, \tag{22}$$

where (22) holds on a high-probability event independent of $g^*$, by Lemma C.2 which we prove below. $\qquad \square$

*Remark* C.1. The failure probability comes from four sources: a Borell's inequality in (19), a Markov's inequality in (20), the singular value bound (17), and a Markov's inequality in the above lemma, on $\bar{G} - G$. The failure probability of (19) is $O(m^{-3})$, which can be easily improved by increasing $B$. The failure probability of (17) is exponentially small [77, Theorem 4.6.1]. The Markov's inequality on $\bar{G} - G$ can be sharpened with another use of Borell-TIS inequality. Therefore, the main source of failure probability comes from our lack of further assumptions on the black-box learner, and can be improved given such assumptions.

**Lemma C.1.** *For any $j \in [m]$,*

$$\mathbb{E}_{g^{(j)}}\|g^{(j)} - \mathrm{Proj}_{m'}g^{(j)}\|_2^2 = \sum_{j=m'+1}^{\infty} \lambda_j \asymp m^{-\bar{b}}. \tag{23}$$

*Proof.* By the $L_2$ convergence of K-L expansion, and the eigendecay assumption. $\qquad\square$

**Lemma C.2.** *On a high-probability event determined by $G$ and $\mathcal{D}^{(n_1)}$, we have, for any $\bar{e}_* \in \mathbb{R}^m$,*

$$\|\bar{e}_*^\top(\hat{\Psi} - \Psi)\|_2 \leq c_r\|\bar{e}_*\|_2(\xi_n + m^{-(\bar{b}+1)/2})\sqrt{\log n}.$$

*Proof.* Introduce the notation $\tilde{e} := \Xi(\Xi^\top\Xi)^{-1}\bar{e}_*$, so that we can write the LHS as

$$\|\bar{e}_*^\top(\hat{\Psi} - \Psi)\|_2 = \|\bar{e}_*^\top(\Xi^\top\Xi)^{-1}\Xi^\top(\bar{G} - \hat{G})\|_2 = \|\tilde{e}(\bar{G} - \hat{G})\|_2 \leq \|\tilde{e}(\bar{G} - G)\|_2 + \|\tilde{e}(\hat{G} - G)\|_2.$$

We consider the two terms in turn.

1. For the first term, observe

$$\mathbb{E}_{G-\bar{G}}\langle(G - \bar{G})_i, (G - \bar{G})_j\rangle = \mathbf{1}_{\{i=j\}}\mathbb{E}\|(G - \bar{G})_1\|_2^2, \text{ where } \mathbb{E}\|(G - \bar{G})_1\|_2^2 \overset{(23)}{\asymp} m^{-\bar{b}}. \tag{24}$$

Moreover, $\Xi$ and $G - \bar{G}$ depends on disjoint subsets of the generalized Fourier coefficients of the GP samples, and are thus independent. Thus, $\tilde{e}$ and $G - \bar{G}$ are also independent, and

$$\mathbb{E}_{G-\bar{G}}\|(G - \bar{G})^\top\tilde{e}\|_2^2 = \mathbb{E}_{G-\bar{G}}\,\tilde{e}^\top(G - \bar{G})(G - \bar{G})^\top\tilde{e} \asymp m^{-\bar{b}}\|\tilde{e}\|_2^2.$$

By the Markov inequality we have, w.h.p. w.r.t. $G - \bar{G}$,

$$\|(G - \bar{G})^\top\tilde{e}\|_2^2 \leq m^{-\bar{b}}\log n\|\tilde{e}\|_2^2. \tag{25}$$

2. For the second term we have

$$\|\tilde{e}^\top(G - \hat{G})\|_2^2 = \int(\tilde{e}^\top(G - \hat{G})(z))^2 P(dz) \leq \int\|\tilde{e}\|_2^2\|(G - \hat{G})(z)\|^2 P(dz)$$

$$= \|\tilde{e}\|_2^2\|G - \hat{G}\|_2^2 \overset{(20)}{\leq} \|\tilde{e}\|_2^2 m\xi_n^2 \log n. \tag{26}$$

In the above, the first inequality is the Cauchy-Schwarz inequality in $\mathbb{R}^m$.

Combining these results and the observation that

$$\|\tilde{e}\|_2 = \|\Xi(\Xi^\top\Xi)^{-1}\bar{e}_*\|_2 \overset{(17)}{\leq} c_r m^{-1/2}\|\bar{e}_*\|_2, \tag{27}$$

we have, with high probability w.r.t. $G$ and $\mathcal{D}^{(n_1)}$,

$$\|\bar{e}_*^\top(\hat{\Psi} - \Psi)\|_2^2 \leq \|\tilde{e}\|_2^2(m\xi_n^2 + m^{-\bar{b}})\log n \leq c_r^2\|\bar{e}_*\|_2^2(\xi_n^2 + m^{-(\bar{b}+1)})\log n.$$

$\qquad\square$

### C.3 Proof for Proposition 4.2

Recall the notations and observations in Appendix C.1. In the following, the first two lemmas bound the *entropy number* of $\tilde{\mathcal{I}}_1$, and the final proof uses it to bound the local Rademacher complexity.

**Lemma C.3.** *There exists an RKHS $\bar{\mathcal{I}}$ s.t. $\tilde{\mathcal{I}}_1 \subset \bar{\mathcal{I}}_1$, and, on the event in Theorem 4.1, we have, for all $0 \leq i_s \leq i_e$, the Mercer eigenvalue bound*

$$\sum_{i=i_s}^{i_e} \lambda_i(\bar{C}) \lesssim i_s^{-\bar{b}} + \chi_n^2, \tag{28}$$

*where $\chi_n := (m^{-\frac{\bar{b}+1}{2}} + \xi_n)\sqrt{\log n}$, and the constant hidden in $\lesssim$ is independent of $j, m$ or $n$.*

*Proof.* We first define $\bar{\mathcal{I}}$. Define $\Delta_1 G_n = \hat{G}_n - G_n, \Delta_2 G_n = G_n - \bar{G}_n$. Then

$$\tilde{\mathcal{I}}_1 = \{\theta^\top(\bar{G}_n + \Delta_1 G_n + \Delta_2 G_n)(\cdot) : \|\theta\|_2^2 \leq m^{-1}\}$$
$$\subset \{\theta_1^\top \bar{G}_n(\cdot) + \theta_2^\top \Delta_1 G_n(\cdot) + \theta_3^\top \Delta_2 G_n(\cdot) : \|\theta_1\|_2^2 + \|\theta_2\|_2^2 + \|\theta_3\|_2^2 \leq 3m^{-1}\}.$$

The RHS is the unit-norm ball of an RKHS, denoted as $\bar{\mathcal{I}}$. Its reproducing kernel and integral operator are (recall our notations in Appendix C.1)

$$\bar{k}(z, z') = \frac{3}{m}(\bar{G}_n(z)^\top \bar{G}_n(z') + \Delta_1 G_n(z)^\top \Delta_1 G_n(z') + \Delta_2 G_n(z)^\top \Delta_2 G_n(z')),$$

$$\bar{C} = \frac{3}{m}(\bar{G}_n^\top \bar{G}_n + (\Delta_1 G_n)^\top \Delta_1 G_n + (\Delta_2 G_n)^\top \Delta_2 G_n).$$

By Wielandt [78, Theorem 2], we have, for any $0 \leq i_s \leq i_e$,

$$\sum_{i=i_s}^{i_e} \lambda_i(\bar{C}) \leq \frac{3}{m}\left(\sum_{i=i_s}^{i_e} \lambda_i(\bar{G}_n^\top \bar{G}_n) + \text{Tr}((\Delta_1 G_n)^\top \Delta_1 G_n) + \text{Tr}((\Delta_2 G_n)^\top \Delta_2 G_n)\right).$$

For the first part above, recall $\bar{G}_n = \Xi\Psi$; and on the event defined in Theorem 4.1, we have, by (16), $(c'')^2 m \leq \lambda_i(\Xi^\top \Xi) \leq (c')^2 m$. Thus,

$$\lambda_i(\bar{G}_n^\top \bar{G}_n) = \lambda_i(\Psi^\top \Xi^\top \Xi \Psi) \overset{(11)}{\asymp} m\lambda_i(\Psi^\top \Psi) \lesssim mi^{-(\bar{b}+1)},$$

The second term is bounded as

$$\text{Tr}((\Delta_1 G_n)^\top(\Delta_1 G_n)) = \|\Delta_1 G_n\|_2^2 \overset{(20)}{\lesssim} m\xi_n^2 \log n.$$

For the third, by Markov's inequality on (24), we have, w.h.p. w.r.t. $G_n - \bar{G}_n$,

$$\text{Tr}((\Delta_2 G_n)^\top(\Delta_2 G_n)) \lesssim m^{-\bar{b}} \log n.$$

Note this event has been included in Theorem 4.1 via Lemma C.2. Combining the above, we have, for all $0 \leq i_s \leq i_e$,

$$\sum_{i=i_s}^{i_e} \lambda_i(\bar{C}) \lesssim \left(\sum_{i=i_s}^{i_e} i^{-(\bar{b}+1)}\right) + (\xi_n^2 + m^{-(\bar{b}+1)})\log n \lesssim i_s^{-\bar{b}} + (\xi_n^2 + m^{-(\bar{b}+1)})\log n \leq i_s^{-\bar{b}} + \chi_n^2.$$

$\square$

**Lemma C.4.** *Let $Z^{n_2} := \{z_1, \ldots, z_{n_2}\}$ be a sample of $n_2$ iid inputs independent of $\mathcal{D}^{(n_1)}$, and $L_2(Z^{n_2})$ denote the $L_2$ space defined by the respective empirical measure. Then, on the high-probability event defined in Theorem 4.1, we have*

$$\mathbb{E}_{Z^{n_2}} e_j(\text{id} : \tilde{\mathcal{I}} \to L_2(Z^{n_2})) \lesssim j^{-(\bar{b}+1)}(\min\{j, n_2\}^{\frac{\bar{b}+1}{2}} + \min\{j, n_2\}^{\bar{b}+\frac{1}{2}}\chi_n)),$$

*where $\chi_n := (m^{-\frac{\bar{b}+1}{2}} + \xi_n)\sqrt{\log n}$, and the constant hidden in $\lesssim$ is independent of $j, m$ or $n$.*

*Proof.* As $\tilde{\mathcal{I}}_1 \subset \bar{\mathcal{I}}_1$, it suffices to establish the bound for $\bar{\mathcal{I}}_1$. For this, we first invoke Steinwart and Christmann [44, Theorem 7.30], but with the RHS of the last display on p. 275 replaced by (28). Following the proof we find, for all $p \in (0, 1)$,

$$\mathbb{E}_{Z^{n_2}} e_j(S_{\bar{k}, D}^*) \lesssim j^{-1/p} \sum_{i=1}^{\min\{j, n_2\}} i^{1/p-1}\sqrt{i^{-1}(i^{-\bar{b}} + \chi_n^2)}.$$

As stated in their Corollary 7.31, the LHS above is our desired empirical entropy number. Following the proof for that corollary, we set $p = (\bar{b} + 1)^{-1}$ above, leading to

$$\mathbb{E}_{Z^{n_2}} e_j(\bar{\mathcal{I}}_1 \to L_2(Z^{n_2})) \lesssim j^{-(\bar{b}+1)} \sum_{i=1}^{\min\{j, n_2\}} i^{\bar{b}}(i^{-(\bar{b}+1)/2} + i^{-1/2}\chi_n)$$

$$\lesssim j^{-(\bar{b}+1)}(\min\{j, n_2\}^{\frac{\bar{b}+1}{2}} + \min\{j, n_2\}^{\bar{b}+\frac{1}{2}}\chi_n).$$

$\square$

We now prove Proposition 4.2, by plugging in our new entropy number bound to the chaining argument in Steinwart and Christmann [44, Theorem 7.12 – Theorem 7.16].

**Proposition C.5** (Proposition 4.2, restated). *Let*

$$\bar{R}_{n_2}(\tilde{\mathcal{I}}_1;\sigma) := \mathbb{E}_{Z^{n_2},\varepsilon_i} \sup_{g\in\tilde{\mathcal{I}}_1,\|g\|_2\le\sigma} \left| \frac{1}{n_2} \sum_{i=1}^{n_2} \varepsilon_i g(z_i) \right|$$

*be the local Rademacher complexity, and $B := \sup_{g\in\tilde{\mathcal{I}}_1}\|g\|_\infty$. Then its critical radius, defined as the solution $\delta_{n_2}$ to*

$$\bar{R}_{n_2}(\tilde{\mathcal{I}}_1;\delta) \le \delta^2, \tag{29}$$

*is bounded as*

$$\delta_{n_2}^2 \lesssim B^{\frac{\bar{b}}{\bar{b}+2}} n_2^{-\frac{\bar{b}+1}{\bar{b}+2}} + \frac{\log^2 n_2}{n_2} + \chi_n^2.$$

*Proof.* Define $I_\sigma := \{g\in\tilde{\mathcal{I}}_1, \|g\|_2\le\sigma\}$. By Steinwart and Christmann [44, Lemma 7.14 and the last display on p. 254], we have

$$e_i(I_\sigma, L_2(Z^{n_2})) \le e_{i-1}(I_\sigma, L_2(Z^{n_2})),$$

$$\mathbb{E}_{Z^{n_2}} e_1(I_\sigma, L_2(Z^{n_2})) \le \mathbb{E}_{Z^{n_2}} \sup_{g\in I_\sigma}\|g\|_{L_2(Z^{n_2})} \le (\sigma^2 + 8B\bar{R}_{n_2}(\tilde{\mathcal{I}}_1;\sigma))^{1/2} =: s_1.$$

Taking expectation on both sides of Steinwart and Christmann [44, Theorem 7.13], we find

$$\bar{R}_{n_2}(\tilde{\mathcal{I}};\sigma) \le \sqrt{\frac{\ln 16}{n_2}} \mathbb{E}_{Z^{n_2}} \Big( \sum_{i=1}^\infty 2^{i/2} e_{2^i}(I_\sigma, L_2(Z^{n_2})) + \underbrace{\sup_{g\in I_\sigma}\|g\|_{L_2(Z^{n_2})}}_{\le s_1} \Big).$$

Plugging in our Lemma C.4 to the first term in RHS above, we have, for $j := 2^i$ and $m_2 := [\log_2 n_2]$,

$$\mathbb{E}_{Z^{n_2}} \sum_{i=1}^\infty 2^{\frac{i}{2}} e_{2^i}(I_\sigma, L_2(Z^{n_2}))$$

$$\le \sum_{i=1}^\infty 2^{\frac{i}{2}} \min\{s_1, j^{-\frac{\bar{b}+1}{2}}\} + \sum_{i=1}^{m_2} 2^{\frac{i}{2}} \cancel{j^{-\frac{1}{2}}} \chi_n + \sum_{i=m_2+1}^\infty 2^{\frac{i}{2}} n_2^{\bar{b}+\frac{1}{2}} j^{-(\bar{b}+1)} \chi_n$$

$$=: S_1 + \chi_n \log n_2 + S_2.$$

By Steinwart and Christmann [44, Lemma 7.15] with $p \leftarrow (\bar{b}+1)^{-1}$, we have $S_1 \lesssim s_1^{\frac{\bar{b}}{\bar{b}+1}}$, where the constant in $\lesssim$ is independent of all sample sizes. For $S_2$, we have

$$S_2 \le n_2^{\bar{b}+\frac{1}{2}} \chi_n \int_{x=m_2}^\infty 2^{\frac{x}{2}-(\bar{b}+1)x} dx \lesssim n_2^{\bar{b}+\frac{1}{2}} \chi_n 2^{\frac{m_2}{2}-(\bar{b}+1)m_2} \le \chi_n.$$

Combining, we have

$$\mathbb{E}_{Z^{n_2}} \sum_{i=1}^\infty 2^{\frac{i}{2}} e_{2^i}(I_\sigma, L_2(Z^{n_2})) \lesssim s_1^{\frac{\bar{b}}{\bar{b}+1}} + \chi_n \log n_2,$$

$$\bar{R}_{n_2}(\tilde{\mathcal{I}};\sigma) \le C n_2^{-\frac{1}{2}} \left( (\sigma^2 + B\bar{R}_{n_2}(\tilde{\mathcal{I}};\sigma))^{\frac{\bar{b}}{2(\bar{b}+1)}} + \chi_n \log n_2 \right),$$

for some constant $C$ which does not depend on $\sigma$ or any sample sizes. To find solutions to (29) using the above bound, we will restrict to

$$\sigma^2 \ge 2C\Big(\frac{\log^2 n_2}{n_2} + \chi_n^2\Big) \ge 4C n_2^{-\frac{1}{2}} \chi_n \log n_2. \tag{30}$$

For such $\sigma$ we now have

$$\bar{R}_{n_2}(\tilde{\mathcal{I}};\sigma) \le C n_2^{-\frac{1}{2}}(\sigma^2 + B\bar{R}_{n_2}(\tilde{\mathcal{I}};\sigma))^{\frac{\bar{b}}{2(\bar{b}+1)}} + \frac{1}{4}\sigma^2 \le \max\left\{3C n_2^{-\frac{1}{2}} \sigma^{\frac{\bar{b}}{\bar{b}+1}}, 3CB^{\frac{\bar{b}}{\bar{b}+2}} n_2^{-\frac{\bar{b}+1}{\bar{b}+2}}, \frac{1}{2}\sigma^2\right\}.$$

For the RHS to be $\le \sigma^2$ it suffices to consider the first two terms, leading to $\sigma^2 \gtrsim B^{\frac{\bar{b}}{\bar{b}+2}} n_2^{-\frac{\bar{b}+1}{\bar{b}+2}}$. Combining with (30) complete the proof. $\square$

## C.4 Average-Case Analysis

**Corollary C.6** (average-case approximation error). *In the setting of Theorem 4.1, with $\mathcal{D}^{(n_1)}$- probability $\to 1$ we have*

$$\mathbb{E}_{g^* \sim \mathcal{GP}} \|\tilde{g}^* - g^*\|_2 \le \mathbb{E}_{g^* \sim \mathcal{GP}} \epsilon_{n_1, m}(g^*),$$

*where $\epsilon_{n_1, m}$ is defined by replacing the $\xi_{n_1}$ in (6) with $m^{-1/2}\xi_{n_1}$, and $\|\tilde{g}^*\|_{\tilde{\mathcal{I}}}$ satisfies (5). In particular, for $m = [n_1^{1/(\bar{b}+1)}]$, we have*

$$\mathbb{E}_{g^* \sim \mathcal{GP}} \|\tilde{g}^* - g^*\|_2 = \tilde{O}(\xi_{n_1} + n_1^{-\frac{\bar{b}}{2(\bar{b}+1)}}). \tag{31}$$

While we will not use (31) for IV, it hints at the possibility of improvement, through more careful analyses and/or additional assumptions. Remark C.2 discusses this in more detail.

*Proof.* Recall the proof of Theorem 4.1, and the notations therein. We can easily check that all statements in the proposition still hold true, with the high-probability events now defined w.r.t. ($G$ and) the combined training samples. Thus, (21) will still hold, and it remains to prove the two approximation error bounds. We will improve Lemma C.2 and plug into (22). We can check the proof of the lemma also remains valid, and (25) is still good enough; thus, it suffices to improve (26) about the term $\|\tilde{e}^\top (G - \hat{G})\|_2$.

The proof of Theorem 4.1 only invokes Lemma C.2 with

$$\bar{e}_* = (\lambda_1^{-1/2} \langle g^*, \psi_1 \rangle_2, \dots, \lambda_{m'}^{-1/2} \langle g^*, \psi_{m'} \rangle_2),$$

where $\{(\lambda_i, \psi_i)\}$ are the Mercer decomposition of $k_x$. For $g^* \sim \mathcal{GP}(0, k_x)$, $\bar{e}_*$ will distribute as $N(0, I)$. We will thus modify Lemma C.2 into:

$$\mathbb{E}_{\bar{e}_* \sim N(0,I)} \|\bar{e}_* (\hat{\Psi} - \Psi)\|_2 \le c_r (m^{-1/2}\xi_n + m^{-(\bar{b}+1)/2}) \sqrt{\log n} \cdot \mathbb{E}_{\bar{e}_* \sim N(0,I)} \|\bar{e}_*\|_2. \tag{32}$$

(The statement is still restricted to a high-probability event w.r.t. training samples and $G$.) To prove the above, note that (25) still holds for any fixed $\bar{e}_*$, so the second term above remains correct. It remains to deal with the first term. Introduce the notation $S := \Xi(\Xi^\top \Xi)^{-1}$, so that $\tilde{e} = S\bar{e}_*$. We have, for *any fixed* $(\mathcal{D}^{(n_1)}, G)$,

$$\begin{aligned}
\mathbb{E}_{\bar{e}_*}(\|\tilde{e}^\top (G - \hat{G})\|_2^2) &= \mathbb{E}_{\bar{e}_*} \operatorname{Tr}(\tilde{e}\tilde{e}^\top (G - \hat{G})(G - \hat{G})^\top) \\
&= \mathbb{E}_{\bar{e}_*} \operatorname{Tr}(S\bar{e}_* \bar{e}_*^\top S^\top (G - \hat{G})(G - \hat{G})^\top) \\
&= \operatorname{Tr}(S \cdot \mathbb{E}_{\bar{e}_*}(\bar{e}_* \bar{e}_*^\top) \cdot S^\top (G - \hat{G})(G - \hat{G})^\top) \\
&= \operatorname{Tr}(SS^\top (G - \hat{G})(G - \hat{G})^\top) \\
&\le \|SS^\top\| \operatorname{Tr}((G - \hat{G})(G - \hat{G})^\top) = \|SS^\top\| \sum_{j=1}^m \|g^{(j)} - \hat{g}_{n_1}^{(j)}\|_2^2.
\end{aligned}$$

The inequality above is the von Neumann inequality $\operatorname{Tr}(AB) \le \|A\|_{op} \operatorname{Tr}(B)$. Conditioned on the $(\mathcal{D}^{(n_1)}, G)$-measurable event on which (17) and (20) hold, the first term above is bounded by $c_r^2 m^{-1}$, and the second by $m\xi_n^2 \log n$. Thus we have, with $(\mathcal{D}^{(n_1)}, G)$-probability $\to 1$,

$$\mathbb{E}_{g^*} \|\tilde{e}^\top (G - \hat{G})\|_2^2 \le c_r^2 \xi_n^2 \log n \lesssim c_r^2 \xi_n^2 \log n \frac{(\mathbb{E}_{\bar{e}_*} \|\bar{e}_*\|_2)^2}{m},$$

where the last inequality follows from Vershynin [77, Theorem 3.1.1]. Since $\mathbb{E}X^2 \ge (\mathbb{E}X)^2$, we complete the new bound for the first term in (32), and subsequently (32). Following the original proof, we can see that (22) becomes

$$\mathbb{E}_{g^* \sim \mathcal{GP}} \|\tilde{g}^* - g^*\|_2 \le \mathbb{E}_{g^* \sim \mathcal{GP}} \left[ c_r \|\operatorname{Proj}_{m'} g^*\|_{\mathcal{I}} (m^{-1/2}\xi_n + m^{-(\bar{b}+1)/2}) \sqrt{\log n} + \|g^* - \operatorname{Proj}_{m'} g^*\|_2 \right],$$

which proves the first claim.

For the second claim, from Jensen's inequality, we know $\mathbb{E}_{g^* \sim \mathcal{GP}}\|\text{Proj}_{m'} g^*\|_{\mathcal{I}} = \mathbb{E}_{\bar{e}_*}\|\bar{e}_*\|_2 \leq \sqrt{\mathbb{E}_{\bar{e}_*}\|\bar{e}_*\|_2^2} = m'$, and thus the first term in the above display is $\tilde{O}(\xi_n + m^{-\bar{b}/2})$. Similarly, we have

$$\mathbb{E}_{g^* \sim \mathcal{GP}}\|g^* - \text{Proj}_{m'} g^*\|_2 \leq \left(\mathbb{E}_{g^* \sim \mathcal{GP}}\|g^* - \text{Proj}_{m'} g^*\|_2^2\right)^{1/2} \overset{(23)}{\leq} O(m^{-\bar{b}/2}),$$

which completes the proof. $\qquad\square$

*Remark* C.2. The only change in this proof is a new bound for the estimation error from the oracle. The Cauchy-Schwarz inequality used in the previous bound (26) was is typically loose: it requires the vector $\tilde{e}$, determined by the test function, to be parallel with $(g^{(1)} - \hat{g}_{n_1}^{(1)}, \ldots, g^{(m)} - \hat{g}_{n_1}^{(m)})$, the estimation residuals. This is at odds with the intuition that residual functions for each independently drawn GP samples to be in some sense uncorrelated.

For an extreme example, suppose our target RKHS $\mathcal{I} = \text{span}\{\psi_1\}$ is one-dimensional, so that the standard GP prior draw can be written as $\epsilon\psi_1$ for some $\epsilon \sim \mathcal{N}(0, 1)$. Then, independent draws from $\mathcal{GP}(0, k_z)$ should have independent signs, and it seems very strange if a useful regression algorithm often returns residual functions with correlated signs determined by $\tilde{e}$, for such $g^{(\cdot)}$. The fact that such oracles are allowed by our assumption suggest there is room for improvement, although we do not pursue this path, since the suboptimality is relatively mild (see the end of Appendix C.5.2). Empirically, our estimator works well for $n_1 = n_2$.

Note that while similar problems have been studied in multi-task learning, such works often assume independent inputs for the different tasks, which corresponds to the different GP prior draws in our setting. Corollary C.6 appears new in this aspect, at least among works that established fast rates.

## C.5 Deferred Derivations and Additional Discussions

### C.5.1 Derivation for Example 4.1

We first derive the expression of $\xi_n$. For regression functions of the form $g = \bar{g} \circ \Phi$, where $\Phi$ is defined as in the text, and $\bar{g}$ is $\underline{\beta}_2$ Hölder regular and have $d_l$ inputs, Schmidt-Hieber [10] establishes the convergence rate

$$\xi_n \lesssim (n^{-\frac{\beta_1}{2\beta_1 + d_z}} + n^{-\frac{\beta_2}{\beta_2 + d_l}})\|\bar{g}\|_{C^{\underline{\beta}_2}} \log^{3/2} n + \epsilon_{opt}. \tag{33}$$

(We view $\|\Phi\|_{C^{\beta_1}}$ as a constant.) The result holds uniformly for all $\bar{g} \in C^{\underline{\beta}_2}$ with uniformly bounded norm.

Recall Example A.3: for any $\epsilon > 0$, there exists a version[13] of the Matérn GP $\Pi_z$ for $\bar{g}$ s.t. $\bar{g} \in C^{\beta_2 - \epsilon}$ a.s. We set $\underline{\beta}_2 = \beta_2 - \epsilon$. By Lemma A.4, the random variable $\|\bar{g}\|_{C^{\underline{\beta}_2}}$ has a subgaussian tail. Thus for any $p > 1$, we can choose some $C_1 > 0$ which depend on $p$ and $\epsilon$, leading to

$$\Pi_z(E) := \Pi_z(\{\bar{g} : \|\bar{g}\|_{C^{\underline{\beta}_2}}^2 \leq C_1 \log n\}) \geq 1 - n^{-p}.$$

Let the NN model in Schmidt-Hieber [10] be constructed for $g = \bar{g} \circ \Phi$ satisfying the above norm bound, and $\hat{g}_n$ denote the resulted estimator. Consider

$$\mathbb{E}_{\Pi_z}\|\hat{g}_n - g\|_2^2 \leq \mathbb{E}_{\Pi_z}[\mathbf{1}_E\|\hat{g}_n - g\|_2^2] + 2\mathbb{E}_{\Pi_z}[\mathbf{1}_{E^c}(\|\hat{g}\|_\infty^2 + \|g\|_2^2)]$$

$$\leq \mathbb{E}_{\Pi_z}[\mathbf{1}_E\|\hat{g}_n - g\|_2^2] + \frac{2\mathbb{E}_{\Pi_z}[\|\hat{g}\|_\infty^2 \mid E^c]}{n_2^p} + 2\mathbb{E}_{\Pi_z}[\mathbf{1}_{E^c}\|g\|_2^2]. \tag{34}$$

The first term clearly has the desired bound, with two extra logarithms. For the second term, note the estimator in Schmidt-Hieber [10] has sup norm bounded as $\|\hat{g}_n\|_\infty \leq C_1 \log n$, so it also has the desired bound. For the last term, another application of Lemma A.4 shows that $\|\bar{g}\|_2$ norm also has a subgaussian tail. Let $\Phi_2$ denote its CDF. Then we have

$$\mathbb{E}(\mathbf{1}_{E^c}\|g\|_2^2) \leq \int_{\Phi_2^{-1}(1-n_2^{-p})}^{+\infty} x^2 d\Phi_2(x) \lesssim \int_{C_2\sqrt{p\log n_2}}^{\infty} x^2 e^{-C_3 x^2} dx \lesssim n_2^{-C_4 p}\sqrt{p \log n_2}.$$

---

[13]We can work with any version of the GP prior since the regression oracle only accesses its evaluation on a finite number of points, which have the same distribution among all versions.

In the above, $C_2, C_3, C_4$ are determined by $\Pi_z$, so we can choose a sufficiently large $p$ so that the RHS is $\lesssim n^{-1}$. This completes the derivation for $\xi_n$. □

We now turn to the regression error using a fixed-form kernel. The Matérn process $\bar{g} \sim \mathcal{GP}(0, \bar{k}_z)$ takes value in the Hölder space $C^{\beta_2}_{-}(\bar{\mathcal{Z}})$ w.p.1, and the order cannot be made larger (Example A.3). Thus, the random function $g = \bar{g} \circ \Phi$ can only take value in $C^{\min\{\beta_2, \beta_1\}}(\mathcal{Z})$ w.p.1. Our claimed regression rate follows from [38] (for Matérn kernels), or [79] (for Gaussian/RBF kernels with an adaptive bandwidth).[14] □

### C.5.2  Derivations for Example 4.2

By properties of the Matérn RKHS (Example A.1), the latent-space kernel $\bar{k}_z$ has eigendecay $\lambda_i \lesssim i^{-(1+2\beta_2/d_l)}$. Thus we have $\bar{b} + 1 = 1 + 2\beta_2/d_l$, and

$$\xi_{n_1} = \tilde{\mathcal{O}}\Big(\epsilon_{fea,n_1} + n_1^{-\frac{\bar{b}-\epsilon}{2(\bar{b}-\epsilon+1)}}\Big) + \epsilon_{opt},$$

for all $\epsilon > 0$. We first establish the three approximation error bounds.

1. The first claim (Case (i) in the text) follows by observing $m^{-\bar{b}} = n_1^{-\bar{b}/(\bar{b}+1)} \ll \xi_{n_1}$.
2. For Case (ii), let $\tilde{g}_0 \in \tilde{\mathcal{I}}$ be the approximation returned by Corollary C.6, then we have

$$\mathbb{E}_g \|\tilde{g}_0 - g\|_2 \overset{(31)}{=} \tilde{O}(\xi_{n_1} + n_1^{-\frac{\bar{b}}{2(\bar{b}+1)}}) = \tilde{O}(\xi_{n_1}). \tag{35}$$

It remains to provide an RKHS norm bound. By convergence of the Karhunen-Loève expansion [66, Thm. 3.1] we have $\text{Proj}_m g = \sum_{i=1}^m \epsilon_i (\lambda_i^z)^{1/2} \psi_i^z$, where $\{(\lambda_i^z, \psi_i^z)\}$ denote the Mercer representation of $k_z$, and $\epsilon_i \sim \mathcal{N}(0, 1)$ are iid. As $(\lambda_i^z)^{1/2} \psi_i^z$ is an ONB of $\mathcal{H}$ (Lem. A.1), we have $\|\text{Proj}_m g\|_{\mathcal{H}}^2 = \sum_{i=1}^m \epsilon_i^2$. Thus, $\chi^2$-concentration bounds [46, Example 2.11] yields

$$\mathbb{P}(\|\text{Proj}_m g\|_{\mathcal{H}}^2 > 2\sqrt{2}m) \leq e^{-m} = \exp\big(-n^{\frac{1}{\bar{b}+1}}\big).$$

Thus, we let $\tilde{g} = 0$ on the above event, and $\tilde{g}_0$ otherwise. Repeating the argument of (34) we can show that the additional $L_2$ error is negligible, and $\tilde{g}$ satisfies the same bound of (35). In this case, the average-case $L_2$ rate of DNN is $\xi_{n_1}$, by definition.

3. Additionally, we claim that in Case (ii), for $m = [n^{\frac{\bar{b}}{(\bar{b}+1)^2}}]$, there exists $\tilde{g}^* \in \tilde{\mathcal{I}}$ s.t.

$$\mathbb{P}_{g^* \sim \mathcal{GP}}(\|\tilde{g}^*\|_{\tilde{\mathcal{I}}} \leq c_r n_1^{1/2(\bar{b}+1)}, \|\tilde{g}^* - g^*\|_2 = \tilde{\mathcal{O}}(m^{1/2}\xi_{n_1} + n_1^{-\bar{b}^2/2(\bar{b}+1)^2})) \geq 1 - e^{-m} \to 1.$$

To show this, we shall apply Theorem 4.1 to a different target function $g^*$. Let us denote by $g$ the GP prior draw to be approximated. Then we have the standard prior mass bound for the "sieve set" [36]: for some $C > 0$,

$$\forall \bar{n} \in \mathbb{N}, \mathbb{P}_{g \sim \mathcal{GP}(0, k_z)}(\exists g_{\bar{n}}^\dagger, \|g_{\bar{n}}^\dagger\|_{\mathcal{I}} \leq C\bar{n}^{1/2(\bar{b}+1)}, \|g_{\bar{n}}^\dagger - g\|_2 \leq C\bar{n}^{-\bar{b}/2(\bar{b}+1)}) \geq 1 - \exp\big(-\bar{n}^{\frac{1}{\bar{b}+1}}\big). \tag{36}$$

(This is proved by combining the Borell inequality [e.g., 71, Prop. 11.17] and the $L_2$ small-ball probability bound [e.g., 66, Cor. 4.9, with $\beta = 1$].) We set $\bar{n} := [n_1^{\bar{b}/(\bar{b}+1)}]$ and restrict to $g$ in the above event. Invoking Lemma A.3 with $g \leftarrow g_{\bar{n}}^\dagger, m \leftarrow [\bar{n}^{\frac{1}{\bar{b}+1}}]$ leads to

$$\|\text{Proj}_m g_{\bar{n}}^\dagger\|_{\mathcal{I}} \leq \|g_{\bar{n}}^\dagger\|_{\mathcal{I}} \lesssim \bar{n}^{\frac{1}{2(\bar{b}+1)}}, \quad \|\text{Proj}_m g_{\bar{n}}^\dagger - g_{\bar{n}}^\dagger\|_2^2 \lesssim \bar{n}^{-\frac{\bar{b}}{2(\bar{b}+1)}}.$$

Invoking Theorem 4.1 with $m \leftarrow 2[\bar{n}^{\frac{1}{\bar{b}+1}}], g^* \leftarrow \text{Proj}_m g_{\bar{n}}^\dagger$ yields

$$\|\tilde{g}\|_{\tilde{\mathcal{I}}} \leq c_r \bar{n}^{\frac{1}{2(\bar{b}+1)}}, \quad \|\tilde{g} - g^*\|_2 \leq c_r \bar{n}^{\frac{1}{2(\bar{b}+1)}} (\xi_{n_1} + \bar{n}^{-1/2}) \log n_1 + 0.$$

---

[14]Note that we did not rule out the possibility of a better rate being attainable: we did not prove a lower bound, instead only presented the best known upper rate given our Hölder regularity condition of $g$. However, improvement is most likely impossible: as discussed in introduction, separability results have been established in similar feature learning settings for certain fixed-form kernels; in the setting of this example, [10] established a lower bound for a wavelet model with a similar order.

Two applications of triangle inequality yield

$$\|\tilde{g} - g\|_2 \lesssim \bar{n}^{\bar{b}/2(\bar{b}+1)^2} (\xi_{n_1} + \bar{n}^{-1/2}) \log n + \bar{n}^{-\bar{b}/2(\bar{b}+1)} = \tilde{\mathcal{O}}\Big(\epsilon_{fea,n_1} + n_1^{-(\bar{b}^2-\epsilon)/2(\bar{b}+1)^2}\Big),$$

for all $\epsilon > 0$. This result is most useful for functions satisfying the conditions in (36), which may or may not be random GP prior draws.

In Case (ii) the average-case DNN rate is $\xi_{n_1}$, by definition. Case (iii) can thus be suboptimal when feature learning is easy, as the exponent of the second term above is worse by a multiplicative factor of $(\bar{b} - \epsilon)/(\bar{b} + 1)$. Note the suboptimality can also be removed if we can take $n_1 \gtrsim n_2^{(\bar{b}+1)/\bar{b}}$, at which point estimation error starts to dominate the combined regression error. The difference in order diminishes as $\bar{b} \to \infty$. The situation in Case (i) is less clear, since it is unclear if the DNN can take advantage of the improved regularity of $g^* \in \mathcal{I}$, see below. $\square$

*Remark* C.3 (optimal rate for $g^* \in \mathcal{I}$). We discuss whether the DNN oracle in [10] may estimate $g^* \in \mathcal{I}$ with a rate better than $\xi_n$.

The condition that $g^* \in \mathcal{I}$, or equivalently $\bar{g}^* \in \bar{\mathcal{I}}$, implies that $\bar{g}^*$ now possesses a better Sobolev regularity comparing with typical GP samples (Example A.1-A.3), but it provides no additional information on Hölder regularity, which is needed in [10]. If we additionally assume the Hölder regularity is also improved by the same amount (i.e., $\bar{g}^* \in C^{\underline{\beta}_2 + (d_l/2)}(\bar{\mathcal{Z}})$), the DNN rate will improve to $\tilde{\mathcal{O}}(\epsilon_{fea,n_1} + n_1^{-\bar{b}+1/2(\bar{b}+2)})$, by (33); this is smaller than $\xi_{n_1}$ if feature learning is sufficiently easy. Without the extra Hölder regularity, we can only invoke the Sobolev embedding theorem to obtain $\bar{g}^* \in C^{\underline{\beta}_2}(\bar{\mathcal{Z}})$; the derived DNN rate can only be infinitesimally better than $\xi_{n_1}$.

### C.5.3 Alternative Kernel Learning Schemes

Let us compare the proposed algorithm with a more obvious alternative. For simplicity, we assume the feature learning term in $\xi_{n_1}$ does not dominate, and restrict to Case (i) in Example 4.2 ($g^* \in \mathcal{I}$).

The established bound requires $O(n_1^{\bar{b}/(\bar{b}+1)^2})$ calls to the regression oracle.[15] In the IV setting, we have $\bar{b} \geq \max\{2p-1, b\}$, and possibly $\bar{b} = b + 2p$ (App. B.1). An alternative instrument learning procedure is to construct a $d$-dimensional approximation to $\mathcal{H}$, invoke the oracle on its basis functions, and use the estimates to construct an alternative $\tilde{\mathcal{I}}$. For an approximation error of $n_1^{-(b+1)/2(b+2)}$, we need $d \asymp n_1^{1/(b+2)}$ bases [80, 81], which becomes much worse than $n_1^{1/(\bar{b}+2)}$ when $p$ is moderately large. Moreover, finding such an approximation is difficult: the same references note that intuitive choices, such as uniformly sampled random feature, or Nyström inducing points, actually require $d \asymp n^{\frac{b}{b+1}} \to n$. It may be also possible that such an algorithm actually has worse performance, as we have not checked if its dependency on the oracle estimation error remains unchanged.

Our approximation gracefully adapts to the ill-posedness of the problem, while also being simpler to implement. Intuitively, this is because it directly focuses on the approximation of the optimal $\mathcal{I}$ as opposed to $\mathcal{H}$. In other words, it works with a basis $\{\varphi_i\} \subset \mathcal{H}$ that is not necessarily optimal for the approximation of $\mathcal{H}$, but is designed so that $\{E\varphi_i\}$ approximate $\mathcal{I}$ well. Interestingly, we do not require the knowledge of such a basis, instead automatically adapting to it. *This is made possible by the isotropy of the $\mathcal{GP}(0, k_x)$ prior*: the GP can be viewed as being defined by an aribtrary basis, convergence technicalities notwithstanding.

## D   Approximation and Estimation Results for Gaussian Process Regression

This section includes various technical results for using $\tilde{\mathcal{I}}$ in the "GP regime", i.e., when the regression function does not live in $\tilde{\mathcal{I}}$, but only satisfies an approximability condition like Asm. 2.2 *(iii)*.b. Regularization becomes weaker in the GP scheme than the kernel scheme; to see this, consider a regression task with data $\{(z_i, \bar{y}_i = g_0(z_i) + \epsilon_i) : i \in [\bar{n}]\}$. Observe that both the KRR estimate and the GP posterior mean estimator can have the form of

$$\hat{g}_n := \arg\min_{g \in \tilde{\mathcal{I}}} \frac{1}{\bar{n}} \sum_{i=1}^{\bar{n}} (g(z_i) - \bar{y}_i)^2 + \bar{\nu} \|g\|_{\tilde{\mathcal{I}}}^2,$$

---

[15]Or equivalently, solving a vector-valued regression problem with a similar output dimension.

but the kernel scheme uses $\bar{\nu} \gg \bar{n}^{-1}$ (unless the RKHS has fixed finite dimensionality), to dominate the critical radius of the RKHS [46], whereas the GP scheme uses $\bar{\nu} \asymp \bar{n}^{-1}$. See [38, 67] for more discussions. It is thus understandable that the GP scheme requires additional assumptions.

We first impose the following assumption on the *true RKHS* $\mathcal{I}$:

**Assumption D.1.** Let the Mercer eigenvalues of $\mathcal{I}$ be $\lambda_i \asymp i^{-(\bar{b}+1)}$. Then (EMB) holds for $\gamma = \frac{\bar{b}-1}{\bar{b}+1}$.

Recall the "top-level RKHS" $\bar{\mathcal{I}}$ and $\mathcal{I}$ satisfy embedding properties of the same order. As discussed in Example A.1, this assumption holds when $\bar{\mathcal{I}}$ is a Matérn RKHS with $\bar{b} > 2$, and its kernel is supported on a bounded, open subset of $\mathbb{R}^{d_l}$ with smooth boundaries.

The above assumption controls estimation error using the true RKHS $\mathcal{I}$. To see this, observe Lemma A.2 now implies

$$\|g\|_\infty \lesssim \|g\|_{\mathcal{I}}^{\frac{\bar{b}-1}{\bar{b}+1}} \|g\|_2^{\frac{2}{\bar{b}+1}}, \quad \forall g \in \mathcal{I}. \tag{37}$$

Fast-rate convergence analyses typically require sup norm bounds on a "relevant subset" of the hypothesis space, such as $\{g - g_0 : g \in \mathcal{I}, \|g - g_0\|_2 \text{ is small}\}$. The interpolation inequality (37) provides such a bound if we were using the true model $\mathcal{I}$. Note that such assumptions are not needed in the kernel regularization scheme, because as discussed above, in that case the ridge regularizer has a greater magnitude, thus providing more control for the RKHS norm on the relevant subset.

However, we will use the approximate model $\tilde{\mathcal{I}}$ for estimation, so to control similar sup norm quantities, we also need some crude bound on the sup-norm approximation error of $\tilde{\mathcal{I}}$. This, in turn, requires the following crude sup-norm error bound for the regression oracle.

**Assumption D.2.** On a high-probability event determined by $\mathcal{D}^{(n_1)}$, we have, for $m \asymp n^{1/\bar{b}+1}$, $\sum_{j=1}^m \|g^{(j)} - \hat{g}_{u,n_1}^{(j)}\|_\infty^2 \lesssim 1$.

The assumption requires sup norm error to scale at $n^{-1/2(\bar{b}+1)}$. We expect it to be mild, at least when $\bar{b}$ is reasonably large (and feature selection is not too hard). This is because in the classical kernel literature, the sup norm error rate for Sobolev kernels is $O(n^{-(\bar{b}-1)/2(\bar{b}+1)})$ [48]. Moreover, if the GP samples are $(\bar{b}-\epsilon)d/2$-Hölder regular as in the Matérn example, error rates in $L_2$ and sup norm will both be $O(n^{-(\bar{b}-\epsilon)/2(\bar{b}+1-\epsilon)})$.

## D.1 Sup Norm Approximation

All results in this subsection assume Assumption D.1.

**Lemma D.1.** Let $\Pi_g$ denote the standard GP prior defined by $k_z$, and $\tau_n = n^{-\frac{\bar{b}/2}{\bar{b}+1}}$. There exist constants $C, C' > 0$ s.t. (i) for all $n \in \mathbb{N}$,

$$\Pi_g(\Theta_n) := \Pi_g(\{g = g_h + g_e : \|g_h\|_{\mathcal{I}}^2 \leq Cn\tau_n^2, \|g_e\|_2^2 \leq C\tau_n^2, \|g_e\|_\infty^2 \leq Cn^{-\frac{1}{\bar{b}+1}}\}) \geq 1 - e^{-C'n\tau_n^2}. \tag{38}$$

(ii) In the above display, we can always choose $g_h = \mathrm{Proj}_m g$, for $m = \lceil n^{\frac{1}{\bar{b}+1}} \rceil$.

*Proof.* (i) is a consequence of Steinwart [66] and is proved in Wang et al. [26, Corollary 13, with $b \leftarrow \bar{b}, b' \leftarrow b - 1$]. To show (ii), it suffices to verify that

$$\tilde{g}_h := \mathrm{Proj}_m g_h + \mathrm{Proj}_m g_e, \quad \tilde{g}_e := \mathrm{Proj}_{>m} g_h + \mathrm{Proj}_{>m} g_e$$

still satisfy the above display. For $\tilde{g}_h$, let $\lambda_m \asymp m^{-(\bar{b}+1)}$ be the $m$-th Mercer eigenvalue for $k_z$. Then

$$\|\tilde{g}_h\|_{\mathcal{I}} \leq \|g_h\|_{\mathcal{I}} + \|\mathrm{Proj}_m g_e\|_{\mathcal{I}} \leq \sqrt{Cn\tau_n^2} + \lambda_m^{-1/2}\|\mathrm{Proj}_m g_e\|_2 \lesssim n^{\frac{1/2}{\bar{b}+1}} = \sqrt{n\tau_n^2}.$$

$$\|\mathrm{Proj}_{>m} g_h\|_2 \leq \lambda_m^{1/2}\|g_h\|_{\mathcal{I}} \lesssim \tau_n. \tag{39}$$

$$\|\tilde{g}_e\|_2 \leq \|g_e\|_2 + \|\mathrm{Proj}_{>m} g_h\|_2 \leq C\tau_n + \lambda_m^{1/2}\|g_h\|_{\mathcal{I}} \lesssim \tau_n.$$

$$\|\tilde{g}_e\|_\infty \leq \|g_e\|_\infty + \|\mathrm{Proj}_{>m} g_h\|_\infty \overset{(37)}{\leq} Cn^{-\frac{1/2}{\bar{b}+1}} + (\|\mathrm{Proj}_{>m} g_h\|_{\mathcal{I}})^{\frac{\bar{b}-1}{\bar{b}+1}} (\|\mathrm{Proj}_{>m} g_h\|_2)^{\frac{2}{\bar{b}+1}}$$

$$\leq Cn^{-\frac{1/2}{\bar{b}+1}} + n^{\frac{1/2}{\bar{b}+1} \cdot \frac{\bar{b}-1}{\bar{b}+1}} (\|\mathrm{Proj}_{>m} g_h\|_2)^{\frac{2}{\bar{b}+1}} \overset{(39)}{\lesssim} n^{-\frac{1}{2(\bar{b}+1)}}.$$

$\square$

**Corollary D.2.** *Let $g^{(1)}, \ldots, g^{(m)} \sim \Pi_g$ be i.i.d. samples from the GP prior. Then*

$$\Pi_g \left( \sum_{i=1}^m \|\mathrm{Proj}_{>m} g^{(i)}\|_\infty^2 \geq C \right) \leq m e^{-C'm} \to 0. \tag{40}$$

*Proof.* By Lemma D.1 with $n := m^{\bar{b}+1}$, and union bound. $\square$

**Proposition D.3.** *Suppose the conditions in Theorem 4.1 hold, and Assumption D.1, D.2 hold. Then on a high-probability event determined by $\mathcal{D}^{(n_1)}$, for any $g^* \in \mathcal{I}$, the approximation $\tilde{g}^* \in \tilde{\mathcal{I}}$ constructed in Theorem 4.1 will also satisfy*

$$\|\tilde{g}^* - g^*\|_\infty \leq \|g^* - \mathrm{Proj}_{m'} g^*\|_\infty + C m^{-1/2} \|\mathrm{Proj}_{m'} g^*\|_\mathcal{I},$$

*where the constant $C$ is independent of $g^*$.*

*Proof.* The proof parallels those of Theorem 4.1 and Lemma C.2. Recall that on the event defined in Theorem 4.1, we have $\|g^{(j)}\|_\infty \lesssim \sqrt{\log n}$, and thus $\|\hat{g}_{n_1}^{(j)} - g^{(j)}\|_\infty \leq \|\hat{g}_{u,n_1}^{(j)} - g^{(j)}\|_\infty$. Also recall the definitions of $\bar{e}_*, \tilde{e}$ therein, and $\tilde{g}^* = \bar{e}_*^\top \hat{\Psi}$.

Now we have

$$\|\tilde{g}^* - g^*\|_\infty \leq \|\bar{e}_*^\top (\hat{\Psi} - \Psi)\|_\infty + \|g^* - \mathrm{Proj}_{m'} g^*\|_\infty = \|\tilde{e}(\bar{G} - \hat{G})\|_\infty + \|g^* - \mathrm{Proj}_{m'} g^*\|_\infty$$

$$\leq \|\tilde{e}(\bar{G} - G)\|_\infty + \|\tilde{e}(G - \hat{G})\|_\infty + \|g^* - \mathrm{Proj}_{m'} g^*\|_\infty.$$

$$\|\tilde{e}(G - \hat{G})\|_\infty = \sup_{z \in \mathcal{Z}} \langle \tilde{e}, (G - \hat{G})(z) \rangle_2 \leq \|\tilde{e}\|_2 \left( \sum_{j=1}^m \|g^{(j)} - \hat{g}_{n_1}^{(j)}\|_\infty \right)^{1/2} \lesssim \|\tilde{e}\|_2. \quad \text{(Asm. D.2)}$$

$$\|\tilde{e}(G - \bar{G})\|_\infty \leq \|\tilde{e}\|_2 \left( \sum_{j=1}^m \|g^{(j)} - \mathrm{Proj}_m g^{(j)}\|_\infty \right)^{1/2} \overset{(40)}{\lesssim} \|\tilde{e}\|_2.$$

Combining the above displays, (27), and the fact that $\|\bar{e}_*\|_2 = \|\mathrm{Proj}_{m'} g^*\|_\mathcal{I}$ (see proof for Theorem 4.1) completes the proof. $\square$

## D.2 Estimation in Fixed-Design Regression

### D.2.1 Small-Ball Probability Bound

It is known that certain well-behaving entropy number bounds imply a small-ball probability bound [82]. For our purpose, however, we need to modify their proof, as our entropy number bound in Lemma C.4 is somewhat less regular.

**Lemma D.4.** *Let $\tilde{\mathcal{I}}$ be a finite-dimensional RKHS with a bounded reproducing kernel, supported on a bounded subset of $\mathbb{R}^{d_z}$. $\iota : \mathcal{I} \to L_2(P(dz))$ be the natural inclusion operator, $\Pi = N(0, \iota \iota^*)$ be the standard GP prior (see Claim B.1). Suppose the average entropy number bound in Lemma C.4 hold. Then there exists some constant $C > 0$, such that for*

$$\epsilon_{n_2} := C(n_2^{-\frac{\bar{b}/2}{\bar{b}+1}} \log n_2 + \chi_{n_1} \log^2 n_2) \lesssim (n_2^{-\frac{\bar{b}/2}{\bar{b}+1}} \log n_2 + (m^{-\frac{\bar{b}+1}{2}} + \xi_{n_1}) \sqrt{\log n_1} \log^2 n_2),$$

*we have, on a high-probability event determined by $\mathcal{D}^{(n_1)}$ and $Z^{n_2}$,*

$$-\log \Pi(\{g : \|g\|_{n_2} \geq \epsilon_{n_2}\}) \leq n_2 \epsilon_{n_2}^2.$$

*Proof.* Let $\iota_{n_2} : \tilde{\mathcal{I}} \to L_2(Z^{n_2})$ be the inclusion operator, $l_k(\iota_{n_2})$ be the $l$-approximation number. Li and Linde [82, Lemma 2.1] states that for some univeral $c_1, c_2$,[16]

$$l_k(\iota_{n_2}) \leq c_1 \sum_{j \geq c_2 k} e_j(\iota_{n_2}^\top) j^{-1/2} \quad \forall k \in \mathbb{N}.$$

---

[16]We dropped the logarithm factor therein because $L_2(Z^{n_2})$ is $K$-convex [83].

Pajor and Tomczak-Jaegermann [84, Theorem 3.2] states that, for some universal $c_3 > 0$,

$$e_{[c_3 j]}(\iota_{n_2}^\top) \leq 2 e_j(\iota_{n_2}) \quad \forall j \in \mathbb{N}.$$

Combining the two results, and taking expectation w.r.t. $Z^{n_2}$, we have

$$\mathbb{E}_{Z^{n_2}} l_k(\iota_{n_2}) \leq c_1 \sum_{j \geq c_2' k} j^{-1/2} \mathbb{E}_{Z^{n_2}} e_j(\iota_{n_2}), \quad \forall k \in \mathbb{N}.$$

Plugging in Lemma C.4, we can see that for $\chi_{n_1} = (m^{-\frac{\bar{b}+1}{2}} + \xi_{n_1})\sqrt{\log n_1}$,

$$
\begin{aligned}
\mathbb{E}_{Z^{n_2}} l_k(\iota_{n_2}) &\lesssim \sum_{j \geq c_2' k} j^{-(\bar{b}+\frac{3}{2})}(\min\{j, n_2\}^{\frac{\bar{b}+1}{2}} + \min\{j, n_2\}^{\bar{b}+\frac{1}{2}} \chi_{n_1}) \\
&= \sum_{j=c_2' k}^{n_2} j^{-(\bar{b}+\frac{3}{2})}(j^{\frac{\bar{b}+1}{2}} + j^{\bar{b}+\frac{1}{2}} \chi_{n_1}) + \sum_{j > n_2} j^{-(\bar{b}+\frac{3}{2})}(n_2^{\frac{\bar{b}+1}{2}} + n_2^{\bar{b}+\frac{1}{2}} \chi_{n_1}) \\
&\lesssim \left( \sum_{j=c_2' k}^{\infty} j^{-\frac{\bar{b}}{2}-1} + \sum_{j=c_2' k}^{n_2} j^{-1} \chi_{n_1} \right) + (n_2^{\frac{\bar{b}+1}{2}} + n_2^{\bar{b}+\frac{1}{2}} \chi_{n_1}) n_2^{-(\bar{b}+\frac{1}{2})} \\
&\lesssim k^{-\frac{\bar{b}}{2}} + n_2^{-\frac{1}{2}} + \chi_{n_1}(1 + \log n_2), \quad \forall k \in \mathbb{N}
\end{aligned}
$$
(41)

invoking Markov's inequality on (41),[17] with $k = [n_2]^{\frac{1}{\bar{b}+1}}$, we have, for some $C > 0$ and on a high-probability event determined by $Z^{n_2}$,

$$\ell_k(\iota_{n_2}) \leq C(n_2^{-\frac{\bar{b}/2}{\bar{b}+1}} \log n_2 + \chi_{n_1} \log^2 n_2) = \epsilon_{n,n_2}.$$

On this event we have

$$n(\epsilon_{n,n_2}) \leq k \leq n_2^{\frac{1}{\bar{b}+1}}, \quad \text{where} \quad n(\epsilon) := \max\{k : 4l_k(\iota_{n_2}) \geq \epsilon\} \tag{42}$$

Our conditions about $\tilde{\mathcal{I}}$ imply the GP prior has a Karhunen-Loève expansion; as $\tilde{\mathcal{I}}$ is finite-dimensional, the Karhunen-Loève expansion always converge. Therefore, by Li and Linde [82, Lemma 2.3, Proposition 2.3], we have, for any $\epsilon > 0$,

$$-\log \Pi(\{g : \|g\|_{n_2} \geq \epsilon\}) \leq n(\epsilon) \log \frac{n(\epsilon)}{\epsilon}.$$

Plugging (42) to the above completes the proof. □

### D.2.2 Regression with Fixed Design

Consider the regression problem with fixed design:

$$\bar{Y}_i = g_0(Z_i) + \varepsilon_i, \quad i = 1, 2, \ldots, n, \tag{43}$$

where $Z_1, \ldots, Z_n$ are fixed, $\varepsilon_1, \ldots, \varepsilon_n$ are independent 1-subgaussian random variables. Define $p_0$ to be the distribution of $\bar{Y} := (\bar{Y}_1, \ldots, \bar{Y}_n)$ and $p_g := \mathcal{N}(g(Z), I_n)$ for any function $g$. Let $\Pi_n$ be a GP prior and $\Theta$ be a parameter space such that $\Pi_n(\Theta) = 1$, we consider the fractional posterior for $\alpha \in (0, 1)$:

$$\Pi_{n,\alpha}(A \mid Z) := \frac{\int_A [p_g(\bar{Y})]^\alpha \Pi_n(dg)}{\int_\Theta [p_g(\bar{Y})]^\alpha \Pi_n(dg)}. \tag{44}$$

Define $r_n(g, g^\dagger) := \log p_{g^\dagger}(\bar{Y}) - \log p_g(\bar{Y})$ and $B_n(g^\dagger) := \{g \in \Theta : \mathbb{E}_{p_0} r_n(g, g^\dagger) \leq n\epsilon_n^2, \mathrm{Var}_{p_0} r_n(g, g^\dagger) \leq n\epsilon_n^2\}$ and $D_\alpha^{(n)}(g, g^\dagger) := -\frac{1}{1-\alpha} \log \mathbb{E}_{p_0} \left(\frac{p_g}{p_{g^\dagger}}\right)^\alpha$. By invoking the contraction theorem of the fractional posterior under the misspecified setting (since $p_0$ is non-Gaussian), we have the following theorem.[18]

---

[17] We note that the concentration should be much sharper: our entropy number bounds are based on tail sums of Gram matrix eigenvalues, which are $O(n_1^{-1})-$subgaussian [85]. For simplicity, however, we do not optimize the failure probability.

[18] In the original statement of Bhattacharya et al. [86, Corollary 3.7], $g^\dagger$ is the best KL-approximation of $p_0$ in $\Theta$. Following the same line of their proof, this corollary also holds for an arbitrary function $g^\dagger$ in our setting.

**Theorem D.5** ([86], Corollary 3.7). *For any $n \in \mathbb{N}$, let $\Pi_n$ be an $n$-dependent prior, $\alpha \in (0,1)$ be arbitrary, $\epsilon_n \in (0,1)$ be such that $n\epsilon_n^2 > 2$, $g^\dagger$ be a function (not necessarily in $\Theta$) such that*

$$- \log \Pi_n(B_n(g^\dagger)) \leq n\epsilon_n^2.$$

*Then with $p_0$ probability at least $1 - 2/(n\epsilon_n^2)$, we have*

$$\int \left\{ \frac{1}{n} D_\alpha^{(n)}(g, g^\dagger) \right\} \Pi_{n,\alpha}(dg \mid Z) \leq \frac{2\alpha + 1}{1 - \alpha} \epsilon_n^2.$$

Let $\Pi_n$ be the GP determined by $\tilde{\mathcal{I}}$, then the postrior mean is

$$g^* := \int_\Theta g \Pi_{n,\alpha}(dg \mid Z) = \underset{g \in \tilde{\mathcal{I}}}{\arg\min} \frac{1}{2\alpha n} \sum_{i=1}^n (\bar{Y}_i - g(Z_i))^2 + \frac{1}{n} \|g\|_{\tilde{\mathcal{I}}}^2, \tag{45}$$

Since $p_g$ is Gaussian and $p_0$ is subgaussian, we can explicitly compute $B_n(g^\dagger)$ and $D_\alpha^{(n)}$ in the above theorem and obtain the following corollary.

**Corollary D.6.** *Let $g^*$ be the GPR posterior mean estimator as in* (45), *for $\alpha = \frac{1}{2}$; and $\bar{Y}_i = g_0(Z_i) + \varepsilon_i$ where $\mathbb{E}(\varepsilon_i \mid Z_i) = 0$, and $\varepsilon_i$ is 1-subgaussian. Let $\Pi_n$ be the standard GP prior determined by $\tilde{\mathcal{I}}$, be arbitrary, $\epsilon_n \in (0,1)$ be such that $n\epsilon_n^2 > 2$, and*

$$- \log \Pi_n(\{g : \|g - g_0\|_n \lesssim \epsilon_n\}) \leq n\epsilon_n^2.$$

*Then we have, for some universal constant $c > 0$,*

$$\mathbb{P}(\|g^* - g_0\|_n \geq c\epsilon_n \mid Z) \leq \frac{1}{2n\epsilon_n^2}.$$

*Proof.* By the subgaussian properties, $\mathbb{E}_{p_0} \left( \frac{p_g}{p_{g^\dagger}} \right)^\alpha$ can be bounded as follows:

$$\begin{aligned}
& \mathbb{E}_{p_0} \exp\left( -\frac{n\alpha}{2} \left( \|\bar{Y} - g\|_n^2 - \|\bar{Y} - g^\dagger\|_n^2 \right) \right) \\
&= \exp\left( -\frac{n\alpha}{2} \left( \|g\|_n^2 - \|g^\dagger\|_n^2 - 2\langle g_0, g - g^\dagger \rangle_n \right) \right) \mathbb{E}_{\varepsilon_1, \dots, \varepsilon_n} e^{\alpha \sum_{k=1}^n \varepsilon_k (g(Z_k) - g^\dagger(Z_k))} \\
&\leq \exp\left( -\frac{n\alpha}{2} \left( \|g\|_n^2 - \|g^\dagger\|_n^2 - 2\langle g_0, g - g^\dagger \rangle_n \right) \right) e^{\frac{n\alpha^2}{2} \|g - g^\dagger\|_n^2} \\
&= \exp\left( -\frac{n\alpha}{2} \left( (1 - \alpha) \|g - g^\dagger\|_n^2 - 2\langle g_0 - g^\dagger g - g^\dagger \rangle_n \right) \right),
\end{aligned}$$

then it holds that

$$\begin{aligned}
D_\alpha^{(n)}(g, g^\dagger) &:= -\frac{1}{1 - \alpha} \log \mathbb{E}_{p_0} \left( \frac{p_g}{p_{g^\dagger}} \right)^\alpha \\
&\geq \frac{n\alpha}{2(1 - \alpha)} \left( (1 - \alpha) \|g - g^\dagger\|_n^2 - 2\langle g_0 - g^\dagger g - g^\dagger \rangle_n \right). \tag{46}
\end{aligned}$$

Similarly, since $r_n(g, g^\dagger) := \log p_{g^\dagger}(\bar{Y}) - \log p_g(\bar{Y})$, then

$$\mathbb{E}_{p_0} r_n(g, g^\dagger) = \frac{n}{2} \left( \|g - g^\dagger\|_n^2 - 2\langle g_0 - g^\dagger, g - g^\dagger \rangle_n \right), \quad \text{Var}_{p_0} r_n(g, g^\dagger) = n\|g - g^\dagger\|_n^2.$$

Setting $g^\dagger = g_0$, the set $B_n = \{g \in \Theta : \mathbb{E}_{p_0} r_n(g, g_0) \leq n\epsilon_n^2, \text{Var}_{p_0} r_n(g, g_0) \leq n\epsilon_n^2\}$ reduces to $\{g \in \Theta : \|g - g^\dagger\|_n \leq \epsilon_n^2\}$, and thus by the assumption we know $-\log \Pi_n(B_n) \lesssim n\epsilon_n^2$, which fulfills the requirement in Theorem D.5. Therefore, the following holds with probability $1 - \frac{2}{n\epsilon_n^2}$

$$\int \|g - g_0\|_n^2 \Pi_{n,\alpha}(dg \mid Z) \overset{(46)}{\lesssim} \int D_\alpha^{(n)}(g, g_0) \Pi_{n,\alpha}(dg \mid Z) \lesssim \epsilon_n^2. \tag{47}$$

Finally, Jensen's inequality yields that

$$\|g^* - g_0\|_n^2 = \left\| \int g \Pi_{n,\alpha}(dg \mid Z) - g_0 \right\|_n^2 \leq \int \|g - g_0\|_n^2 \Pi_{n,\alpha}(dg \mid Z) \lesssim \epsilon_n^2, \tag{48}$$

which completes the proof. $\qquad \square$

# E Deferred Proofs: IV Regression

## E.1 Proof for Proposition 5.1

We will use a slightly modified version of Theorem 1 in Dikkala et al. [19], which we state below with changed notations.

**Theorem E.1** (Dikkala et al. 19, adapted). *Let $\mathcal{H}, \mathcal{I}$ be normed function spaces, such that functions in $\mathcal{H}_{B_x}$ and $\mathcal{I}_{3U}$ has bounded ranges in $[-1, 1]$. Consider the estimator*

$$\hat{f}_n := \arg\min_{f \in \mathcal{H}} \max_{g \in \mathcal{I}} \frac{1}{n} \sum_{i=1}^n (y_i - f(x_i)) g(z_i) - \lambda \left( \|g\|_{\mathcal{I}}^2 + \frac{U}{\delta^2} \|g\|_{2,n}^2 \right) + \mu \|f\|_{\mathcal{H}}^2. \tag{49}$$

*Assume $f_0 \in \mathcal{H}$, and*

$$\forall f \in \mathcal{H} : \min_{g \in \mathcal{I}, \|g\|_{\mathcal{I}} \leq L\|f - f_0\|_{\mathcal{H}}} \|g - E(f - f_0)\|_2 \leq \eta_n \|f - f_0\|_{\mathcal{H}}. \tag{50}$$

*Let $\delta = c_1 \delta_n + c_2 \sqrt{\frac{\log(c_3/\zeta)}{n}}$, where $c_1, c_2, c_3 > 0$ are universal constants, and $\delta_n$ is an upper bound for the critical radii of $\mathcal{I}_{3U}$ and the function space*[19]

$$\mathcal{G} := \{(x, z) \mapsto (f - f_0)(x) g_f(z) : f - f_0 \in \mathcal{H}_{B_x}, g_f = \arg\min_{g \in \mathcal{I}_{L^2 B_x}} \|g_f - E(f - f_0)\|_2\}.$$

*If $\lambda > \delta^2/U, \mu > 2\lambda(1 + 4L^2 + 27U/B_x)$, we will have, w.p. $1 - 3\zeta$:*

$$\|E(\hat{f}_n - f_0)\|_2 \leq \left( 1025\delta + \eta_n + \frac{(3U + 54B_x^{-1}U + 8L^2 + 2)\lambda + \mu}{\delta} \right) \max\{1, \|f_0\|_{\mathcal{H}}^2\}.$$

**Proof for the modified theorem**   The only change we make is in (50), where we added the factor $\|f - f_0\|_{\mathcal{H}}$. We can see that the only use of its original version in Dikkala et al. [19] is on $f \leftarrow \hat{f}_n$ (p. 54 therein), so their proof will continue to hold after replacing $\eta_n$ with $\eta_n \|\hat{f}_n - f_0\|_{\mathcal{H}}$. Therefore, by the last display on their p. 54:[20]

$$\frac{\delta}{2} \|E(\hat{f}_n - f_0)\|_2 \leq 1025\delta^2 + \delta\eta_n \|\hat{f}_n - f_0\|_{\mathcal{H}} + 3\lambda U + 2 \sup_{g \in \mathcal{I}} \Psi^{\bar{\nu}/2}(f_0, g)$$

$$+ 27\delta^2 \frac{\|\hat{f}_n - f_0\|_{\mathcal{H}}^2}{B_x} + 4\lambda L^2 \|\hat{f}_n - f_0\|_{\mathcal{H}}^2 + \mu(\|f_0\|_{\mathcal{H}}^2 - \|\hat{f}_n\|_{\mathcal{H}}^2).$$

Since $\lambda > \max\{\delta^2/U, \delta\eta_n\}$, the sum of the second and last three terms in RHS are bounded by

$$\delta\eta_n + \lambda \left( 1 + \frac{27}{B} + 4L^2 \right) \|\hat{f}_n - f_0\|_{\mathcal{H}}^2 + \mu(\|f_0\|_{\mathcal{H}}^2 - \|\hat{f}_n\|_{\mathcal{H}}^2)$$

$$\leq \delta\eta_n + 2\lambda \left( 1 + \frac{27}{B} + 4L^2 \right) (\|\hat{f}_n\|_{\mathcal{H}}^2 + \|f_0\|_{\mathcal{H}}^2) + \mu(\|f_0\|_{\mathcal{H}}^2 - \|\hat{f}_n\|_{\mathcal{H}}^2).$$

Since $\mu \geq 2\lambda(1 + \frac{27U}{B_x} + 4L^2)$, the latter is bounded by $\delta\eta_n + (2\lambda(27B_x^{-1}U + 4L^2 + 1) + \mu)\|f_0\|_{\mathcal{H}}^2$. Following the next two displays on their p. 55, we have

$$\frac{\delta}{2} \|E(\hat{f}_n - f_0)\|_2 \leq 2 \sup_{g \in \mathcal{I}} \Psi^{\bar{\nu}/2}(f_0, g) + 1025\delta^2 + 3\lambda U + \delta\eta_n + (2\lambda(27B_x^{-1}U + 4L^2 + 1) + \mu)\|f_0\|_{\mathcal{H}}^2.$$

By our assumption that $f_0 \in \mathcal{H}$, their subsequent upper bound for $\Psi^{\bar{\nu}/2}$ becomes $\sup_{g \in \mathcal{I}} \Psi^{\bar{\nu}/2}(f_0, g) \leq 0$. Thus

$$\|E(\hat{f}_n - f_0)\|_2 \leq 1025\delta + 3\frac{\lambda U}{\delta} + \eta_n + \frac{2\lambda(27B_x^{-1}U + 4L^2 + 1) + \mu}{\delta} \|f_0\|_{\mathcal{H}}^2.$$

This completes the proof.  □

---

[19]We dropped a scaling in its definition since our $\mathcal{H}$ and $\mathcal{I}$ are star-shaped.

[20]We make the constant hidden in their big-O notation explicit.

**Proof for Proposition 5.1**  Let us set $n \leftarrow n_2, \mathcal{I} \leftarrow \tilde{\mathcal{I}}, \delta \asymp n^{-\frac{b+1}{b+2}}, B_x \asymp 1, U \asymp (\log n_1)^{-1}$, $\lambda \leftarrow \delta^2/U, \mu \leftarrow 2\lambda(1 + 4L^2 + 27U/B_x)$, where the constants hidden in $\asymp$ are determined by Assumption 2.2, 4.1 and Theorem 4.1, and are independent of any sample size. Then the estimator (49) becomes

$$\hat{f}_{n_2} := \arg\min_{f \in \mathcal{H}} \max_{g \in \tilde{\mathcal{I}}} \frac{1}{n_2} \sum_{i=1}^{n_2} (y_i - f(x_i) - g(z_i))g(z_i) - \lambda\|g\|_{\mathcal{I}}^2 + \mu\|f\|_{\mathcal{H}}^2. \tag{51}$$

We claim that with $\delta, \eta_{n_2}, L$ set as below, the conditions in Theorem E.1 will now hold in our setting, conditioned on the high-probability events in Theorem 4.1. This is because:

1. By Proposition 4.2, the squared critical radius for $\tilde{\mathcal{I}}_{3U}$ is bounded by $\tilde{O}(n_2^{-\frac{\bar{b}+1}{b+2}} + \xi_{n_1} + m^{-(\bar{b}+1)})$, for $\mathcal{G}$ we consider its entropy number in the empirical $L_2$ norm (Definition A.3). Denote by $L_2(Z^{n_2})$ the empirical $L_2$ space. Observe

$$e_{j+1}(\mathcal{G}, L_2(Z^{n_2})) \leq U^{-1}L^2B_x \cdot e_j(\mathcal{H}_{B_x}, L_2(Z^{n_2})) + e_j(\tilde{\mathcal{I}}_{L^2B_x}, L_2(Z^{n_2})),$$

since we can combine the coverings in the RHS to obtain a covering for $\mathcal{G}$. By Steinwart and Christmann [44, Exercise 7.7], we have

$$\mathbb{E}_{Z^{n_2}} e_j(\mathcal{H}_{B_x}, L_2(Z^{n_2})) \lesssim B_x j^{-(b+1)} \min\{j, n_2\}^{\frac{b+1}{2}},$$

Combining the above, Lemma C.4, and the fact that $\bar{b} \geq b$, we have

$$\mathbb{E}_{Z^{n_2}} e_j(\mathcal{G}, L_2(Z^{n_2})) \lesssim j^{-(b+1)} \min\{j, n_2\}^{\frac{b+1}{2}} \log n_1 + j^{-(\bar{b}+1)} \log n_1 \min\{j, n_2\}^{\bar{b}+\frac{1}{2}} \chi_{n_1},$$

where $\chi_{n_1}$ is defined therein. As the above bound has a similar structure to Lemma C.4, we can repeat the proof for Proposition 4.2 and find that $\delta_{n_2}^2 = \tilde{O}(n_2^{-\frac{b+1}{b+2}} + \xi_{n_1} + m^{-(\bar{b}+1)})$

2. The boundedness condition for $\mathcal{H}_{B_x}$ is satisfied by Assumption 2.2; that for $\mathcal{I}_{3U}$ is verified in Section 4.

3. It remains to determine $L$ and $\eta_n$ in (50). We claim (50) will hold by setting

$$L = c_\tau, \quad \eta_{n_2} \asymp (\xi_{n_1} + m^{-\frac{\bar{b}+1}{2}}) \log n_1.$$

To prove this, observe that the true $\mathcal{I}$ defined in Lemma 3.1 would satisfy (50) with $L = 1, \eta_{n_2} \equiv 0$; for our approximation $\tilde{\mathcal{I}}$, by Theorem 4.1 applied to $g^* = E(f - f_0)$, we know there exists some $\tilde{g}$

$$\|\tilde{g} - E(f - f_0)\|_{\tilde{\mathcal{I}}} \leq c_\tau\|E(f - f_0)\|_{\mathcal{I}} \leq c_\tau\|f - f_0\|_{\mathcal{H}},$$

$$\begin{aligned}
\|\tilde{g} - E(f - f_0)\|_2 &\leq c_\tau\|g^*\|_{\mathcal{I}}(\xi_{n_1} + m^{-\frac{\bar{b}+1}{2}}) \log n + \|g^* - \text{Proj}_m g^*\|_2 \\
&\overset{(10)}{\lesssim} \|g^*\|_{\mathcal{I}}[(\xi_{n_1} + m^{-\frac{\bar{b}+1}{2}}) \log n_1 + m^{-\frac{\bar{b}+1}{2}}] \\
&\leq 2\|f - f_0\|_{\mathcal{H}}(\xi_{n_1} + m^{-\frac{\bar{b}+1}{2}}) \log n_1.
\end{aligned}$$

Now all conditions in the theorem are fulfilled, and for $n_1 \geq n_2, m \geq n_2^{\frac{1}{b+2}}$, we get the convergence rate of

$$\begin{aligned}
\|E(\hat{f}_n - f_0)\|_2 &\leq \left(1025\delta + \eta_{n_2} + \frac{(3U + 54B_x^{-1}U + 8L^2 + 2)\lambda + \mu}{\delta}\right) \max\{1, \|f_0\|_{\mathcal{H}}^2\} \\
&= \tilde{O}\left((n_2^{-\frac{b+1}{2(b+2)}} + \xi_{n_1}) \max\{1, \|f_0\|_{\mathcal{H}}^2\}\right),
\end{aligned}$$

as claimed. $\qquad\square$

## E.2  Assumptions and Setup in Theorem 5.2

We have imposed Assumptions D.1, D.2. Asm. D.1 accounts for our different assumption about $\mathcal{I}$, comparing with [26] (see App. E.3 below), and is satisfied by Matérn kernels with suitable orders; Asm. D.2 requires the regression oracle to satisfy a mild sup norm error bound, which is used to control the sup norm approximation error of $\tilde{\mathcal{I}}$. App. D discusses these assumptions in detail. We also introduce the following two assumptions, both taken from [26]:

The following assumption is widely used in the NPIV literature [87, 40, 35]. It connects estimation error for $Ef_0$ to that for $f_0$.

**Assumption E.1** (link condition)**.** Let $\{\bar{\varphi}_i : i \in \mathbb{N}\}$ denote the Mercer eigenfunctions of $k_x$. Then we have, for all $f \in L_2(P(dx))$ and $j \in \mathbb{N}$,

$$j^{-2p}\|\mathrm{Proj}_j f\|_2^2 \lesssim \|Ef\|_2^2 \lesssim \sum_{i=1}^{\infty} i^{-2p}\langle f, \bar{\varphi}_i\rangle_2^2.$$

The following assumption first appears in a literature that analyzes kernel ridge regression in a "hard learning" scenario [68, 48]. As discussed in [26], it accounts for the different regularity between "typical" GP prior draws and the RKHS [38], which makes GP modeling to fall into this scenario. If it is known that $f_0 \in \mathcal{H}_0$ for some RKHS $\mathcal{H}_0$ with a bounded kernel and eigendecay $\lambda_i(T_{\mathcal{H}_0}) \lesssim i^{-b}$, in GP modeling we should specify as $\mathcal{H}$ the power RKHS $\mathcal{H}_0^{b+1+\epsilon/b}$ (Defn. A.1), so that both (an infinitesimally deteriorated version of) the GP scheme of Asm. 2.2 and this assumption can hold [26].[21] As discussed in Example A.1, this assumption is satisfied by all Matérn kernels satisfying Asm. 2.2 *(iii)*.b, given the requirement for $P(dx)$ therein.

**Assumption E.2** (embedding property)**.** For some (arbitrarily small) $\epsilon > 0$, (EMB) holds for $\mathcal{H}$ with $\gamma = \frac{b-\epsilon}{b+1}$.

The quasi-posterior is defined by a standard GP prior $\Pi = \mathcal{GP}(0, k_x)$, and its Radon-Nikodym derivative w.r.t. the prior:

$$\frac{\Pi(df \mid \mathcal{D}_{s2}^{(n_2)})}{\Pi(df)} \propto e^{-\frac{n_2}{\lambda}\ell_{n_2}(f)}, \quad \text{where} \quad \ell_{n_2}(f) := \sup_{g \in \tilde{\mathcal{I}}} \Big( \sum_{i=1}^{n_2} 2(f(x_i) - y_i)g(z_i) - g(z_i)^2 \Big) - \nu\|g\|_{\tilde{\mathcal{I}}}^2. \tag{52}$$

In the above, $\lambda, \nu \asymp 1$ are scaled ridge regularizers, and $\tilde{\mathcal{I}}$ is constructed as in Algorithm 1. As the log quasi-likelihood $\ell_{n_2}$ is quadratic in $f$, we can check the corresponding posterior mean estimator has a similar form to (49), with $\lambda, \delta^2 \leftarrow (2n_2)^{-1}\nu, U \leftarrow \frac{1}{2}$ and $\mu \leftarrow (2n_2)^{-1}\lambda$. Note this is an invalid choice for Proposition 5.1 which requires $\delta^2, \lambda, \mu$ to be $\tilde{\Theta}(n_2^{-b+1/b+2})$, demonstrating the difference in regularization scale.

## E.3  Proof for Theorem 5.2

We will modify the proof for Theorem 3 in [26] to allow for our different assumptions about $\mathcal{I}$, and account for the approximation error in $\tilde{\mathcal{I}}$. For the former, note that in [26] $\mathcal{H}$ is in the GP scheme, but $\mathcal{I}$ is in the kernel scheme: it has eigendecay $\lambda_i \asymp i^{-(b+2p)}$ and contains the image of the power RKHS $\mathcal{H}^{b/b+1}$ under $E$ (Assumption 7 therein). In contrast, our $\mathcal{I}$ is also in the GP scheme, having the eigendecay of $i^{-(b+2p+1)}$ and only containing the image of $\mathcal{H}$.

Still, all assumptions in [26] about $\mathcal{H}$ and $E$ are equivalent to ours, so their technical lemmas that do not involve $\mathcal{I}$ continue to hold. This leaves us with the final proofs of their Proposition 20 and the Theorem 3, which include the only occurrences of $\mathcal{I}$. We will address them in turn.

Throughout the proof, the denotation of the constants $(C, C', \dots)$ may change from line to line.

### E.3.1  Replaced Denominator Bound

This subsection replaces Proposition 20 in [26]. For all $n \in \mathbb{N}$, define $\delta_n := n^{-\frac{b+2p}{2(b+2p+1)}}$.

---

[21]The deterioration by $\epsilon$ can be removed if $[\mathcal{H}_0]^{1-\epsilon}$ can be embedded into $L_\infty(P(dx))$.

**Lemma E.2** (Local sub-Gaussian complexity). *Let $\bar{V}^{n_2} := \{\bar{v}_i : i \in [n_2]\}$ be a set of $1$-bounded rvs which are conditionally independent given $Z^{n_2}$, and have zero conditional mean. Let $\mathcal{G}(\tilde{\mathcal{I}}_1; \delta) = \mathbb{E}_{\bar{V}^{n_2}} \sup_{g \in \tilde{\mathcal{I}}_1, \|g\|_{n_2} \leq \delta} \left| \frac{1}{n_2} \sum_{i=1}^{n_2} g(z_i) \bar{v}_i \right|$. Then there exists $C_1 > 0$ s.t. for any $\delta \geq n_2^{-1/2}$, we have*

$$\mathbb{P}_{Z^{n_2}}\left( \mathcal{G}(\tilde{\mathcal{I}}_1; \delta) \leq C_1 n^{-\frac{1}{2}} \delta^{\frac{b+2p}{b+2p+1}} \right) \geq 1 - e^{-n_2^{\frac{1}{b+2p+1}}/\log^4 n_1}.$$

*Proof.* Recall $\bar{\mathcal{I}}$ as defined in Lemma C.3. As $\tilde{\mathcal{I}}_1 \subset \bar{\mathcal{I}}_1$, it suffices to establish the above for $\bar{\mathcal{I}}_1$.

Let $\{\hat{\lambda}_j\}, \{\lambda_j\}$ denote the eigenvalues of the integral operators w.r.t. the empirical and population measure. Let $m := [(\delta^2)^{-\frac{1}{b+2p+1}}]$. By Lemma C.3 and our choice of $n_1$, we have

$$\sum_{j=m}^{n_2} \lambda_j \lesssim m^{-(b+2p)} + \chi_{n_1}^2 \leq (\delta^2)^{\frac{b+2p}{b+2p+1}} + n_2^{-1}.$$

By Shawe-Taylor et al. [85, Theorem 5, Proposition 2],

$$\mathbb{P}_{Z^{n_2}}\left( \sum_{j=m}^{n_2} \hat{\lambda}_j \geq \sum_{j=m}^{n_2} \lambda_j + \epsilon \right) \leq \exp(-2n_2\epsilon^2/\bar{R}^4),$$

where $\bar{R} := \sup_z \bar{k}(z, z)$. By the construction of $\bar{k}$, boundedness of $\{\hat{g}_{n_1}^{(i)}\}, \{g_{n_1}^{(i)}\}$ (see (19)), and Corollary D.2, we can see that on the high $\mathbb{P}_{\mathcal{D}_{s1}^{(n_1)}}$-probability event the theorem conditioned on, $\bar{R} \lesssim \log n_1$. Combining the two displays above, and recalling $\delta^2 \geq n_2^{-1}$, we have, for some $C_0 > 2$,

$$\mathbb{P}_{Z^{n_2}}\left( \sum_{j=m}^{n_2} \hat{\lambda}_j \leq C_0(\delta^2)^{\frac{b+2p}{b+2p+1}} \right) \geq \mathbb{P}_{Z^{n_2}}\left( \sum_{j=m}^{n_2} \hat{\lambda}_j \leq \sum_{j=m}^{n_2} \lambda_j + (C_0 - 2)(\delta^2)^{\frac{b+2p}{b+2p+1}} \right)$$

$$\geq 1 - e^{-n_2^{\frac{1}{b+2p+1}}/\log^4 n_1}.$$

Plugging to Wainwright [46, Theorem 13.22; the proof holds for all bounded rvs], we have, on the above event,

$$\mathcal{G}(\bar{\mathcal{I}}_1; \delta) \leq \sqrt{\frac{2}{n_2}} \sqrt{\sum_{j=1}^{n_2} \min\{\delta^2, \hat{\lambda}_j\}} \leq \sqrt{\frac{2}{n_2}} \sqrt{m\delta^2 + \sum_{j=m}^{n_2} \hat{\lambda}_j} \leq C_1 n_2^{-\frac{1}{2}} \delta^{\frac{b+2p}{b+2p+1}}.$$

$\square$

**Lemma E.3** (KRR norm bound). *Let $V^{n_2} = \{v_i : i \in [n_2]\}$ be a set of $B$-bounded rvs which are conditionally independent given $Z^{n_2}$, and have zero conditional mean. Let $\hat{g}$ be the KRR estimate: $\hat{g} = \arg\min_{g \in \tilde{\mathcal{I}}} \sum_{i=1}^{n_2} (g_0(z_i) + v_i - g(z_i))^2 + \nu \|g\|_{\tilde{\mathcal{I}}}^2$. Let $E_n(Z^{n_2}, V^{n_2})$ denotes the event on which*

1. *There exists $\tilde{g} \in \tilde{\mathcal{I}}$ determined by $Z^{n_2}$, s.t. $\|\tilde{g}\|_{\tilde{\mathcal{I}}}^2 \lesssim n_2^{\frac{1}{b+2p+1}} \log n_2$, $\|\tilde{g} - g_0\|_{n_2}^2 \lesssim \delta_{n_2}^2 \log n_2$.*

2. *$\hat{g}$ satisfies $\|\hat{g} - g_0\|_{n_2}^2 \lesssim \delta_{n_2}^2 \log n_2$.*

*Then, on the intersection of $E_n$ and an event with probability $1 - \gamma_{n_2} \to 1$, we have*

$$\|\hat{g}\|_{\tilde{\mathcal{I}}}^2 \lesssim B^{\frac{b+2p+1}{b+2p+1/2}} n_2^{\frac{1}{b+2p+1}} \log n_2.$$

*Proof.* With an abuse of notation, we denote $\langle g, \epsilon \rangle_{n_2} := \frac{1}{n_2} \sum_{i=1}^{n_2} g(z_i) \epsilon_i$. We now proceed in two steps:

(Step 1) We build a peeling argument. By Wainwright [46, Theorem 3.24], we have, for some $C_1, C_2 > 0$ and any $\delta > 0$,

$$\mathbb{P}_{V^{n_2}}\left( \sup_{g \in \tilde{\mathcal{I}}_1, \|g\|_{n_2} \leq \delta} |\langle g, \epsilon \rangle_{n_2}| \geq \mathcal{G}(\tilde{\mathcal{I}}_1; \delta) + \frac{1}{2} C_1 B n_2^{-\frac{1}{2}} \delta^{\frac{b+2p}{b+2p+1}} \right) \leq \exp(-C_2 \delta^{-\frac{2}{b+2p+1}}).$$

Combining with Lemma E.2, we have, for any $\delta \in (n_2^{-1/2}, n_2^{-\frac{b+2p}{2(b+2p+1)}})$,

$$\mathbb{P}_{V^{n_2}}\left(\sup_{g \in \tilde{\mathcal{I}}_1, \|g\|_{n_2} \leq \delta} |\langle g, \epsilon \rangle_{n_2}| \geq C_1 B n_2^{-\frac{1}{2}} \delta^{\frac{b+2p}{b+2p+1}}\right)$$

$$\leq \exp\left(-(\log^{-4} n_1) n_2^{\frac{1}{b+2p+1}}\right) + \exp\left(-C_2 n_2^{\frac{b+2p}{(b+2p+1)^2}}\right) =: \eta_{n_2}^{(0)}.$$

With an union bound over $\{\delta = e^j n_2^{-1/2} : 0 \leq j \leq \log n_2^{\frac{1/2}{b+2p+1}}\}$, we have, with probability $\geq 1 - \eta_{n_2}^{(0)} \log n_2 \to 1$,

$$\sup\left\{|\langle g, \epsilon \rangle_{n_2}| : g \in \tilde{\mathcal{I}}_1, \|g\|_{n_2} \leq n_2^{-\frac{b+2p}{2(b+2p+1)}}\right\} \leq e C_1 B n_2^{-\frac{1}{2}} \max\{n_2^{-\frac{1}{2}}, \|g\|_{n_2}\}^{\frac{b+2p}{b+2p+1}}.$$

Applying the above to $\frac{1}{\max\{\|g\|_{\tilde{\mathcal{I}}}, 1\}} g$:

$$\sup\left\{|\langle g, \epsilon \rangle_{n_2}| : g \in \tilde{\mathcal{I}}, \|g\|_{n_2} \leq n_2^{-\frac{b+2p}{2(b+2p+1)}}\right\} \leq$$
$$e C_1 B \max\{n_2^{-\frac{1}{2}} \|g\|_{n_2}^{\frac{b+2p}{b+2p+1}} \|g\|_{\tilde{\mathcal{I}}}^{\frac{1}{b+2p+1}}, n_2^{-\frac{2(b+2p)+1}{2(b+2p+1)}} \|g\|_{\tilde{\mathcal{I}}}\}. \tag{53}$$

In particular, this applies to $(C \log n_2)^{-1}(\tilde{g} - \hat{g})$ for some $C > 0$.

(Step 2) As $\hat{g}$ minimizes the empirical loss, we have

$$\nu \|\hat{g}\|_{\tilde{\mathcal{I}}}^2 \leq n_2 \|\hat{g} - g_0\|_{n_2}^2 + \nu \|\hat{g}\|_{\tilde{\mathcal{I}}}^2 \leq n_2 \|\tilde{g} - g_0\|_{n_2}^2 + \nu \|\tilde{g}\|_{\tilde{\mathcal{I}}}^2 + 2 n_2 \langle \epsilon, \hat{g} - \tilde{g} \rangle_{n_2}.$$

Plugging in our conditions, we have

$$\|\hat{g}\|_{\tilde{\mathcal{I}}}^2 \leq C_3 n_2^{\frac{1}{b+2p+1}} \log n_2 + 2 n_2 \langle \epsilon, \hat{g} - \tilde{g} \rangle_{n_2}$$
$$\overset{(53)}{\leq} C_3'(n_2^{\frac{1}{b+2p+1}} \log n_2 + n_2 B \max\{n_2^{-\frac{1}{2}} \|\hat{g} - \tilde{g}\|_{n_2}^{\frac{b+2p}{b+2p+1}} \|\hat{g} - \tilde{g}\|_{\tilde{\mathcal{I}}}^{\frac{1}{b+2p+1}}, n_2^{-\frac{2(b+2p)+1}{2(b+2p+1)}} \|\hat{g} - \tilde{g}\|_{\tilde{\mathcal{I}}}\},$$
$$\|\hat{g} - \tilde{g}\|_{\tilde{\mathcal{I}}}^2 \leq 2(\|\hat{g}\|_{\tilde{\mathcal{I}}}^2 + \|\tilde{g}\|_{\tilde{\mathcal{I}}}^2)$$
$$\leq C_3''(n_2^{\frac{1}{b+2p+1}} \log n_2 + n_2 B \max\{n_2^{-\frac{1}{2}} \|\hat{g} - \tilde{g}\|_{n_2}^{\frac{b+2p}{b+2p+1}} \|\hat{g} - \tilde{g}\|_{\tilde{\mathcal{I}}}^{\frac{1}{b+2p+1}}, n_2^{-\frac{2(b+2p)+1}{2(b+2p+1)}} \|\hat{g} - \tilde{g}\|_{\tilde{\mathcal{I}}}\}.$$

By enumerating the dominating term and taking the maximum of the implied bounds, we conclude that

$$\|\hat{g} - \tilde{g}\|_{\tilde{\mathcal{I}}}^2 \leq 2 C_3'' B^{\frac{b+2p+1}{b+2p+1/2}} n_2^{\frac{1}{b+2p+1}} \log n_2.$$

Another triangular inequality completes the proof. $\qquad\square$

**Lemma E.4** (Bernstein's inequality). *For any function $g$ with $\|g\|_\infty \leq B$, $\|g\|_2 \leq s$,*

$$\mathbb{P}_{Z^{n_2}}(\|g\|_{n_2}^2 \leq 5s^2) \geq 1 - e^{-n_2 s^2/B^2}.$$

*Proof.* The random variable $g(\mathbf{z})^2$ is $\|g\|_\infty^2$ bounded and has mean $\|g\|_2^2$ and variance $\leq \mathbb{E} g(\mathbf{z})^4 \leq \|g\|_\infty^2 \|g\|_2^2$. The claim follows by Bernstein's inequality [e.g., 44, Theorem 6.12, with $\xi_i \leftarrow g(z_i)^2 - \|g\|_2^2$]. $\qquad\square$

**Lemma E.5** (replaces Lemma 19 in 26). *There exist constants $C_1, C_2 > 0$, so that for all $n \in \mathbb{N}$, there exists a function set $\Theta_{d0}$ with prior mass $\Pi(\Theta_{d0}) \geq e^{-C_1 n \delta_n^2}$, s.t. for all $f \in \Theta_{d0}$, we have*

$$\|E(f - f_0)\|_2^2 \leq C_2 \delta_n^2, \quad \|f - f_0\|_\infty^2 \leq C_2. \tag{54}$$

*Moreover, for such $f$ we have $f - f_0 = f_h + f_e$, where*

$$\|f_h\|_{\mathcal{H}}^2 \leq C_2 n \delta_n^2, \quad \|E f_e\|_2^2 \leq C_2 \delta_n^2, \quad \|f_e\|_\infty^2 \leq C_2. \tag{55}$$

*Proof.* Let $\Theta_{d0}$ be defined as Wang et al. [26, proof for Lemma 19]. As stated in that lemma, (54) holds. For such $f$, let $f_h' + f_e' = f - f_0$ be defined as in their proof, so that $\|f_h'\|_{\mathcal{H}}^2 \lesssim n \delta_n^2, \|f_e'\|_2^2 \lesssim n^{-\frac{b}{b+2p+1}}, \|f_e'\|_\infty \lesssim 1$; applying their Lemma 16 to $f_h'$ and $f_e'$ shows the existence of $f_h + f_e = f - f_0$ satisfying (55). $\qquad\square$

**Proposition E.6** (replaces Proposition 20 in [26]). *For some $C_{den} > 0, \eta'_{n_2} \to 0$ we have*

$$\mathbb{P}_{\mathcal{D}_{s2}^{(n_2)}}\left(\int e^{-n_2\ell_{n_2}(f)}\Pi(df) \geq \exp\left(-C_{den}\delta_{n_2}^2 n_2 \log^2 n_2\right)\right) \geq 1 - \eta'_{n_2} \to 1. \tag{56}$$

*Proof.* We proceed in two steps.

(Step 1) This step replaces Step 2 of the proof in Wang et al. [26]; it will establish that for some $C_5 > 0$ and a sequence $\eta_n \to 0$,

$$\inf_{f\in\Theta_{d0}} \mathbb{P}_{\mathcal{D}_{s2}^{(n_2)}}(\{\ell_{n_2}(f) \leq C_5\|E(f - f_0)\|_2^2 + C_5\delta_{n_2}^2 \log^2 n_2\}) \geq 1 - \eta_{n_2}. \tag{57}$$

Recall $\ell_{n_2}(f) = \sup_{g\in\tilde{\mathcal{I}}} \frac{1}{n_2} \sum_{i=1}^{n_2}(2(f(x_i) - y_i)g(z_i) - g(z_i)^2) - n_2^{-1}\nu\|g\|_{\tilde{\mathcal{I}}}^2$ and $\nu \asymp 1$. With some algebra,[22] we find

$$\ell_{n_2}(f) = \|\hat{g}_f\|_{n_2}^2 + n_2^{-1}\nu\|\hat{g}_f\|_{\tilde{\mathcal{I}}}^2, \tag{58}$$

where $\hat{g}_f$ is the optima for $\ell_{n_2}(f)$ and equals the KRR solution (45). We will use Corollary D.6 to bound the $\|\hat{g}_f\|_{n_2}$, and Lemma E.3 to bound $\|\hat{g}_f\|_{\tilde{\mathcal{I}}}$.

First note the residual $\mathbf{e}^f := f(\mathbf{x}) - \mathbf{y} - E(f - f_0)(\mathbf{z})$ is bounded by $2(B + C_2)$, and thus subgaussian: this is because $|y_i - f_0(x_i)| \leq B$ by Assumption 2.1, and $|f(x_i) - f_0(x_i)| \leq C_2$ by (54). Moreover, the true regression function $g_f := E(f - f_0)$ can be approximated in the true RKHS $\mathcal{I}$ using $\bar{g}_f := E(f_h)$, where $f_h$ is defined in Lemma E.5, so that by that Lemma and Lemma 3.2, we have

$$\|\bar{g}_f\|_{\mathcal{I}}^2 = \|f_h\|_{\mathcal{H}}^2 \leq C_2 n_2\delta_{n_2}^2, \quad \|\bar{g}_f - g_f\|_2^2 \leq C_2\delta_{n_2}^2, \quad \|\bar{g}_f - g_f\|_\infty^2 \leq C_2.$$

Combining the above with (54) and the inequality $\|a\|^2 \leq 2\|a\|^2 + 2\|a - b\|^2$ yields

$$\|\bar{g}_f\|_2^2 \leq 4C_2\delta_{n_2}^2, \quad \|\bar{g}_f\|_\infty^2 \leq 4C_2.$$

Applying Theorem 4.1 and Proposition D.3 to $\bar{g}_f \in \mathcal{I}$, and recalling $\xi_{n_1} \lesssim n_2^{-1}$, we can see that there exists $\tilde{g}_f \in \tilde{\mathcal{I}}$ s.t.

$$\|\tilde{g}_f\|_{\tilde{\mathcal{I}}}^2 \lesssim \|\bar{g}_f\|_{\mathcal{I}}^2 \lesssim n_2\delta_{n_2}^2, \quad \|\tilde{g}_f\|_2^2 \leq 2\|\bar{g}_f\|_2^2 + 2\|\tilde{g}_f - \bar{g}_f\|_2^2 \lesssim \delta_{n_2}^2 \log n_1, \quad \|\tilde{g}_f\|_\infty^2 \lesssim 1. \tag{59}$$

Now, a few applications of Lemma E.4 shows that, for some constant $C_3 > 0$, we have, for any $f \in \Theta_{d0}$,

$$\mathbb{P}_{Z^{n_2}}(\max\{\|g_f\|_{n_2}^2, \|\tilde{g}_f\|_{n_2}^2, \|\tilde{g}_f - g_f\|_{n_2}^2\} \leq C\delta_{n_2}^2 \log n_2) \geq 1 - 3e^{-C_3 n_2\delta_{n_2}^2} \to 1. \tag{60}$$

Therefore, for some constant $C'_3 > 0$ and $\Pi_{g,n_1}$ the standard GP prior defined by $\tilde{\mathcal{I}}$, we have

$$\Pi_{g,n_1}(\{g : \|g - g_f\|_{n_2} \leq C'_3\delta_{n_2} \log n_2) \geq \Pi_{g,n_1}(\{g : \|g - \tilde{g}_f\|_{n_2} \leq C_3\delta_{n_2} \log n_2)$$
$$\overset{(i)}{\geq} e^{-\|\tilde{g}_f\|_{\tilde{\mathcal{I}}}^2}\Pi_{g,n_1}(\{g : \|g\|_{n_2} \leq C_3\delta_{n_2} \log n_2)$$
$$\overset{(ii)}{\geq} \exp(-(\|\tilde{g}_f\|_{\tilde{\mathcal{I}}}^2 + n_2C_3\delta_{n_2}^2 \log^2 n_2)) \tag{61}$$
$$\overset{(59)}{\geq} \exp(-2n_2C_3\delta_{n_2}^2 \log^2 n_2),$$

where (i) can be found in Ghosal and Van der Vaart [71, Lemma I.28], and (ii) is by Lemma D.4 and holds on a high-probability event determined by $Z^{n_2}$.

The above display fulfills the condition in Corollary D.6, which now shows that, when $\nu \asymp 1$ is sufficiently large (determined by $B$ and $C_2$), for some $C_4 > 0$ and $\eta_{n_2}^{(1)} \to 0$, both independent of $f \in \Theta_{d0}$, the maximizer $\hat{g}^f$ of $\ell_{n_2}(f)$ above satisfies

$$\mathbb{P}_{\mathcal{D}_{s2}^{(n_2)}}(\|\hat{g}_f - g_f\|_{n_2} \leq C_4\delta_{n_2} \log n_2 \mid Z^{n_2}) \geq 1 - \eta_{n_2}^{(1)} \to 1. \tag{62}$$

(59), (60) and (62) fulfills the conditions in Lemma E.3, which now implies that

$$\mathbb{P}_{\mathcal{D}_{s2}^{(n_2)}}\left(\|\hat{g}_f\|_{\tilde{\mathcal{I}}}^2 \leq C_5 n_2^{\frac{1}{b+2p+1}} \log n_2\right) \geq 1 - \eta_{n_2}^{(2)} \to 1$$

---

[22]See e.g. the beginning of Appendix C.1.3 in Wang et al. [26] for a similar argument.

for some $C_5 > 0$. Plugging the two displays above into (58) yields (57).

(Step 2) By Wang et al. [26, proof for Proposition 20, step 1] applied to (57), we find

$$\mathbb{P}_{\mathcal{D}_{s2}^{(n_2)}}(\Pi\{f \in \Theta_{d0} : \ell_{n_2}(f) \leq C_5\|E(f - f_0)\|_2^2 + C_5\delta_{n_2}^2 \log^2 n_2\}) \geq 1 - 4\eta_{n_2}.$$

Combining the definition of $\Theta_{d0}$ and its prior mass bound in Lemma E.5, we have, on the above event,

$$\int e^{-n_2\ell_{n_2}(f)}\Pi(df) \geq \exp\left(-3C_5 n_2\delta_{n_2}^2 \log^2 n_2\right).$$

This completes the proof. $\qquad\square$

### E.3.2 Replaced Numerator Bound

Following Wang et al. [26], the proof proceeds in two steps. Only the second step contains essential changes.

**Step 1.** Recall the notations in Wang et al. [26, proof for Theorem 3]: in particular, $\epsilon_n := n^{-\frac{b/2}{b+2p+1}}$, $\delta_n := n^{-\frac{(b+2p)/2}{b+2p+1}}$, and $\{\Theta_n : n \in \mathbb{N}\}$ as defined by their Corollary 13. By that corollary, there exists some $C_{gp} > 0$ so that

$$\Pi(\Theta_r^c) \leq \exp\left(-2C_{den}n_2^{\frac{1}{b+2p+1}} \log^2 n_2\right), \quad \text{where} \quad r := \left[\left(C_{gp}^{-1}C_{den}n_2^{\frac{1}{b+2p+1}} \log^2 n_2\right)^{b+1}\right]. \quad (63)$$

By their Lemma 21,[23] when $M > 0$ is sufficiently large, for $\alpha := \frac{2b+1}{b}$, the two sets of functions

$$\text{err}_{n_2,1} := \{f \in \Theta_r : \|f - f_0\|_2 \geq M\epsilon_{n_2} \log^\alpha n_2\},$$
$$\text{err}_{n_2,2} := \{f \in \Theta_r : \|E(f - f_0)\|_2 \geq CM\delta_{n_2} \log^\alpha n_2\}$$

are "equivalent up to constants"; to be precise, we can have $\text{err}_{n_2,1} \subset \text{err}_{n_2,2}$ or $\text{err}_{n_2,2} \subset \text{err}_{n_2,1}$ by choosing $C > 0$ appropriately, independent of $M$ or $n_2$. Therefore, we do not distinguish between them below, and use $\text{err}_{n_2}$ to refer to both of them.

The analysis of (quasi)-posterior contraction rates is based on certain decompositions of the average posterior mass. We follow the decomposition in Wang et al. [26]: let $E_{den}$ denote the event defined in (56), and $A(f, \mathcal{D}_{s2}^{(n_2)})$ be an event to be defined below satisfying

$$\sup_{f \in \Theta_r \cap \text{err}_{n_2}} \mathbb{P}(A^c(f, \mathcal{D}_{s2}^{(n_2)})) \leq \exp\left(-2C_{den}n_2^{\frac{1}{b+2p+1}} \log^2 n_2\right). \quad (64)$$

Then

$$\mathbb{E}_{\mathcal{D}_{s2}^{(n_2)}}(\Pi(\text{err}_{n_2} \mid \mathcal{D}_{s2}^{(n_2)}))$$

$$= \mathbb{E}_{\mathcal{D}_{s2}^{(n_2)}} \frac{\int_{\text{err}_{n_2}} \Pi(df) \exp(-n_2\ell_{n_2}(f))}{\int \Pi(df) \exp(-n_2\ell_{n_2}(f))}$$

$$\leq \mathbb{P}_{\mathcal{D}_{s2}^{(n_2)}} E_{den}^c + \mathbb{E}_{\mathcal{D}_{s2}^{(n_2)}}\left(\mathbf{1}_{E_{den}} \frac{\int_{\text{err}_{n_2}} \Pi(df) \exp(-n_2\ell_{n_2}(f))}{\int \Pi(df) \exp(-n_2\ell_{n_2}(f))}\right)$$

$$\overset{(56)}{\leq} o_{n_2}(1) + \exp\left(C_{den}n_2^{\frac{1}{b+2p+1}} \log^2 n_2\right)\mathbb{E}_{\mathcal{D}_{s2}^{(n_2)}} \int_{\text{err}_{n_2}} \Pi(df) \exp(-n_2\ell_{n_2}(f))$$

$$\leq o_{n_2}(1) + \exp\left(C_{den}n_2^{\frac{1}{b+2p+1}} \log^2 n_2\right)\mathbb{E}_{\mathcal{D}_{s2}^{(n_2)}}\left[\Pi(\Theta_r^c) + \int_{\text{err}_{n_2} \cap \Theta_r} \Pi(df) \exp(-n_2\ell_{n_2}(f))\right]$$

$$\overset{(63)}{\leq} o_{n_2}(1) + \exp\left(C_{den}n_2^{\frac{1}{b+2p+1}} \log^2 n_2\right)\mathbb{E}_{\mathcal{D}_{s2}^{(n_2)}} \int_{\text{err}_{n_2} \cap \Theta_r} \Pi(df) \exp(-n_2\ell_{n_2}(f))$$

$$\overset{(64)}{\leq} o_{n_2}(1) + \exp\left(C_{den}n_2^{\frac{1}{b+2p+1}} \log^2 n_2\right)\mathbb{E}_{\mathcal{D}_{s2}^{(n_2)}}\left[\mathbf{1}_{A(f,\mathcal{D}_{s2}^{(n_2)})} \int_{\text{err}_{n_2} \cap \Theta_r} \Pi(df) \exp(-n_2\ell_{n_2}(f))\right].$$

---

[23] We invoke the lemma with $n \leftarrow [r^{\frac{b+2p+1}{b+1}}] \asymp n_2 \log^{2(b+2p+1)} n_2$, so the lemma actually provides a stronger statement; for any $\alpha > 0$ our claim immediately follows.

Therefore, it suffices to show that, for all $f \in \text{err}_{n_2} \cap \Theta_r$, on the event $A(f, \mathcal{D}^{(n_1)})$,

$$\ell_{n_2}(f) \geq 2C_{den}\delta_{n_2}^2 \log^2 n_2. \tag{65}$$

Following Wang et al. [26], we define

$$\Psi_{n_2}(f, g) := \frac{1}{n_2}\sum_{i=1}^{n_2}(f(x_i) - y_i)g(z_i), \quad \Psi(f, g) := \mathbb{E}_{\mathcal{D}_{s2}^{(n_2)}}\Psi_{n_2}(f, g),$$

$$A(f, \mathcal{D}_{s2}^{(n_2)}) := \left\{ \Psi_{n_2}(f, g) - \|\tilde{g}\|_{n_2}^2 \geq \Psi(f, g) - \frac{3\|\tilde{g}\|_2^2}{2} \right\},$$

where $\tilde{g} \in \tilde{\mathcal{I}}$ is a (replaced) approximation to $E(f - f_0) =: g$ to be defined below; we can still invoke Bernstein's inequality as in Wang et al. [26, end of p. 34], finding

$$\mathbb{P}(A^c(f, \mathcal{D}_{s2}^{(n_2)})) \leq \exp\left(-\frac{n_2\|\tilde{g}\|_2^2}{16(\|f - f_0\|_\infty + B + \|\tilde{g}\|_\infty)^2}\right). \tag{66}$$

We will bound the RHS below.

**Step 2.** We restrict to $f \in \Theta_r \cap \text{err}_{n_2}$ and the event above.

We first define our approximation $\tilde{g}$. Let $\Delta f = f - f_0, \widetilde{\Delta f} \in \mathcal{H}, g = E\Delta f, g_j = E(\text{Proj}_j \widetilde{\Delta f})$ be defined as in the proof of Wang et al. [26, Lemma 21].[24] By their Eq. 40 and Eq. 42 in the proof of Lemma 21, we have

$$\|\widetilde{\Delta f} - \Delta f\|_\infty \lesssim 1, \quad \|\widetilde{\Delta f} - \Delta f\|_2 \lesssim \epsilon_{n_2}, \quad \|\widetilde{\Delta f}\|_{\mathcal{H}} \lesssim n_2^{\frac{1/2}{b+2p+1}}\log n_2,$$
$$\|E(\widetilde{\Delta f} - \Delta f)\|_2 \lesssim \delta_{n_2}, \quad \|g - g_j\|_2 \lesssim \delta_{n_2}. \tag{67}$$

Combining the last two inequalities, we have

$$\|g_j - E(\widetilde{\Delta f})\|_2 \leq \|g - g_j\|_2 + \|E(\widetilde{\Delta f} - \Delta f)\|_2 \lesssim \delta_{n_2}.$$

By Lemma 3.2 we have $g_j, E(\widetilde{\Delta f}) \in \mathcal{I}$, and $\max\{\|E(\widetilde{\Delta f})\|_{\mathcal{I}}^2, \|g_j\|_{\mathcal{I}}^2\} \lesssim n_2\delta_{n_2}^2\log^2 n_2$. Thus their difference has the same $\mathcal{I}$-norm bound, and by (37) we have

$$\|g_j - E(\widetilde{\Delta f})\|_\infty \lesssim \|g_j - E(\widetilde{\Delta f})\|_2^{\frac{1}{b+2p+1}}\|g_j - E(\widetilde{\Delta f})\|_{\mathcal{I}}^{\frac{b+2p}{b+2p+1}} \leq \log n_2.$$
$$\|g_j - g\|_\infty \leq \|g_j - E(\widetilde{\Delta f})\|_\infty + \|E(\widetilde{\Delta f} - \Delta f)\|_\infty \lesssim \log n_2. \tag{68}$$

(We used the inequality $\|E(\cdot)\|_\infty \leq \|\cdot\|_\infty$ as we are working with a version of conditional expectation that is bounded under the sup norm.)

Now, by Theorem 4.1 and Proposition D.3, there exists $\tilde{g} \in \tilde{\mathcal{I}}$ s.t.

$$\|\tilde{g}\|_{\mathcal{I}}^2 \lesssim n_2^{\frac{1}{b+2p+1}}\log^2 n_2, \quad \|\tilde{g} - g_j\|_2^2 \lesssim n_2^{-\frac{b+2p}{b+2p+1}}\log^2 n_2, \quad \|\tilde{g} - g_j\|_\infty^2 \lesssim \log^2 n_2.$$

Combining with (67), (68):

$$\|\tilde{g}\|_{\tilde{\mathcal{I}}} \lesssim n_2^{\frac{1/2}{b+2p+1}}\log n_2, \quad \|\tilde{g} - g\|_2 \lesssim \delta_n \log n_2, \quad \|\tilde{g} - g\|_\infty \lesssim \log n_2. \tag{69}$$

Now we check the probability bound (66) satisfies (64). Define $U := \frac{\|f - f_0\|_2}{\epsilon_{n_2}\log^\alpha n_2}$, so that $U \geq \sqrt{M}$, and by Wang et al. [26, Lemma 21],

$$U\delta_{n_2}\log^\alpha n_2 \lesssim \|g\|_2 \leq \|\Delta f\|_2 \leq U\epsilon_{n_2}\log^\alpha n_2. \tag{70}$$

$$\|\widetilde{\Delta f}\|_2 \leq \|\Delta f - \widetilde{\Delta f}\|_2 + \|\Delta f\|_2 \overset{(67)}{\lesssim} U\epsilon_{n_2}\log^\alpha n_2.$$

---

[24]As in Step 1, we invoke the lemma with $n \leftarrow [r^{\frac{b+2p+1}{b+1}}] = n_2(\log^2 n_2)^{b+2p+1}$, so the various $L_2$ error terms there are smaller ($\epsilon_n \ll \epsilon_{n_2}, \delta_n \ll \delta_{n_2}$), while the RKHS norm terms in the scale of $n^{\frac{1/2}{b+2p+1}}$ now becomes $n_2^{\frac{1/2}{b+2p+1}}\log n_2$.

By the embedding property assumption:
$$\|f - f_0\|_\infty \leq \|\widetilde{\Delta f}\|_\infty + \|\Delta f - \widetilde{\Delta f}\|_\infty \overset{(67)}{\lesssim} \|\widetilde{\Delta f}\|_\infty + 1 \overset{(9)}{\lesssim} \|\widetilde{\Delta f}\|_{\mathcal{H}}^{\frac{b}{b+1}} \|\widetilde{\Delta f}\|_2^{\frac{1}{b+1}} \lesssim U^{\frac{1}{b+1}} (\log n_2)^{\frac{b+\alpha}{b+1}}.$$

Using the triangle inequality and the inequality $\|g\|_\infty = \|E(f - f_0)\|_\infty \leq \|f - f_0\|_\infty$, we bound (66) as

$$\mathbb{P}(A^c(f, \mathcal{D}^{(n_1)})) \leq \exp\left(-\frac{n_2(\|g\|_2 - \|\tilde{g} - g\|_2)^2}{16(\|f - f_0\|_\infty + B + \|\tilde{g} - g\|_\infty + \|g\|_\infty)^2}\right)$$

$$\leq \exp\left(-\frac{n_2(\|g\|_2 - \|\tilde{g} - g\|_2)^2}{16(2\|f - f_0\|_\infty + B + \|\tilde{g} - g\|_\infty)^2}\right)$$

$$\leq \exp\left(-\frac{n_2\delta_{n_2}^2(\log n_2)^{2\alpha}(U^2 - 1)}{C(U^{\frac{1}{b+1}}(\log n_2)^{\frac{b+\alpha}{b+1}} + B + \log n_2)^2}\right).$$

For $\alpha \geq \frac{2b+1}{b}$ and $U$ sufficiently large, the RHS is $\leq \exp(-CU^{\frac{2b}{b+1}} n_2 \delta_{n_2}^2 \log^2 n_2)$ and thus satisfies (64).

It remains to check (65), for $f \in \Theta_r \cap \mathrm{err}_{n_2}$ and on the event $A(f, \mathcal{D}_{s2}^{(n_2)})$. As $\tilde{g} \in \tilde{\mathcal{I}}$, the manipulations in Wang et al. [26, Eq. 50] continue to hold, with $g_j$ replaced by $\tilde{g}$; it leads to
$$\ell_{n_2}(f) \geq c_1 \|g\|_2^2 - C_2 \delta_n^2 - n_2^{-1}\nu\|\tilde{g}\|_{\mathcal{I}}^2,$$
where $c_1, C_2 > 0$ are constants, and we recall $\nu \asymp 1$. Plugging in (70) and (69), we have
$$\ell_{n_2}(f) \geq c_1 U^2 \delta_{n_2}^2 \log^{2\alpha} n_2 - C_2 \delta_n^2 - n_2^{-1}\nu\|\tilde{g}\|_{\mathcal{I}}^2 \geq c_1 U^2 \delta_{n_2}^2 \log^{2\alpha} n_2 - C_2 \delta_n^2 - C_3 \delta_n^2 \log^2 n_2.$$
Since $\alpha > 1$, (65) holds for sufficiently large $U$. This completes the proof for Theorem 5.2. $\square$

## E.4 Proof for Corollary 5.3

The idea here is not new (see e.g., [88]). By Proposition E.6, the upper bound holds on an event $E_{den}$ s.t. $\mathbb{P}_{\mathcal{D}_{s2}^{(n_2)}} E_{den} \to 1$. For the lower bound of $-\Pi(\mathcal{D}_{s2}^{(n_2)})$, observe

$$\Pi(\mathcal{D}_{s2}^{(n_2)}) = \int_{\mathrm{err}_{n_2}} \Pi(df) e^{-n_2 \ell_{n_2}(f)} + \int_{\mathrm{err}_{n_2}^c} \Pi(df) e^{-n_2 \ell_{n_2}(f)}$$

$$\leq \int_{\mathrm{err}_{n_2}} \Pi(df) e^{-n_2 \ell_{n_2}(f)} + \Pi(\mathrm{err}_{n_2}^c).$$

where $\mathrm{err}_{n_2}$ is defined in Appendix E.3.2. For the first part, we have (see Step 1 in Appendix E.3.2)

$$\mathbb{E}_{\mathcal{D}_{s2}^{(n_2)}} \mathbf{1}_{E_{den}} \int_{\mathrm{err}_{n_2}} \Pi(df) e^{-n_2 \ell_{n_2}(f)} \leq e^{-2C_{den} n_2^{\frac{1}{b+2p+1}} \log^2 n_2}.$$

As $\mathbb{P}(E_{den}) \to 1$, it must hold that

$$\mathbb{P}_{\mathcal{D}_{s2}^{(n_2)}}\left(\int_{\mathrm{err}_{n_2}} \Pi(df) e^{-n_2 \ell_{n_2}(f)} \leq e^{-C_{den} n_2^{\frac{1}{b+2p+1}} \log^2 n_2}\right) \to 1.$$

For the second part, recall that by definition we have

$$\mathrm{err}_{n_2}^c \subset \{f : \|f - f_0\|_2 \leq M\epsilon_{n_2} \log^{\frac{2b+1}{b}} n_2\}$$

for some constant $M > 0$, and $\epsilon_{n_2}^2 \asymp n_2^{-\frac{b}{b+2p+1}}$; for $j = [n_2^{\frac{b+1}{b+2p+1}}]$ and $f_j^\dagger$ defined in Assumption 2.2 *(iii)*.b, we have

$$\Pi(\mathrm{err}_{n_2}^c) \leq \Pi(\{f : \|f - f_0\|_2 \leq M\epsilon_{n_2} \log^{\frac{2b+1}{b}} n_2\})$$

$$\overset{(i)}{\leq} \Pi\left(\left\{f : \|f - f_j^\dagger\|_2 \leq \frac{M}{2}\epsilon_{n_2} \log^2 \frac{2b+1}{b} n_2\right\}\right)$$

$$\overset{(ii)}{\leq} e^{\|f_j^\dagger\|_{\mathcal{H}}^2} \Pi\left(\left\{f : \|f\|_2 \leq \frac{M}{2}\epsilon_{n_2} \log^2 \frac{2b+1}{b} n_2\right\}\right)$$

$$\overset{(i)}{\leq} e^{n_2^{\frac{1}{b+2p+1}}} \Pi\left(\left\{f : \|f\|_2 \leq \frac{M}{2}\epsilon_{n_2} \log^2 \frac{2b+1}{b} n_2\right\}\right).$$

In the above, (i) follows by the assumption on $f_j^\dagger$, and (ii) can be found in Ghosal and Van der Vaart [71, Lemma I.28]. Recall the $L_2$ small-ball probability bound: for $\mathcal{H}$ satisfying Assumption 2.2 and sufficiently small $\epsilon$ it holds that [see e.g., 66, Corollary 4.9]

$$-\log \Pi(\{\|f\|_2 \leq \epsilon\}) \asymp \epsilon^{-\frac{2}{b}}.$$

Combining these results complete the proof. $\qquad\square$

# F  Implementation Details

This section describes the implementation of the proposed method. Further experiment details, such as network architectures and the range of hyperparameters, are in Appendix H.

**The Algorithm**  Our algorithm is described in Algorithm 1. In our implementation, we combine the $m$ scalar regression tasks to a single vector-valued regression task, for which we train a neural network model with square loss. We draw approximate GP prior samples using random Fourier features.[25]

Our implementation makes a small modification to $\tilde{k}_z$: we extend it with a single linear feature $\hat{h}(\cdot)$, defined as the regression estimate of $\mathbb{E}(\mathbf{y} \mid \mathbf{z})$, with output truncated by $C \log n_1$. Thus, the kernel becomes $\tilde{k}_z(z, z') = \frac{1}{m} \sum_{i=1}^{m} \hat{g}_{n_1}^{(j)}(z) \hat{g}_{n_1}^{(j)}(z') + \hat{h}(z)\hat{h}(z')$. Clearly, the extension does not affect any approximation or estimation guarantee,[26] but it guard against potential misspecification of $\mathcal{H}$: when Assumption 2.2 (iii) fails to hold, the unmodified $\tilde{\mathcal{I}}$ can only estimate $Ef$ for $f \in \mathcal{H}$ (or $f \sim \mathcal{GP}(0, k_x)$), but not $\mathbb{E}(f(\mathbf{x}) - \mathbf{y} \mid \mathbf{z} = \cdot)$; the modified $\tilde{k}_z$ fixes this. This issue does not arise when the assumption holds.

The kernelized IV quasi-posterior admits the following closed-form expression [19, 26], which we restate for the reader's convenience. The point estimator (7) equals the posterior mean below, with $\lambda, \nu$ adjusted appropriately (see Appendix E.1, or [19, Appendix E]).

$$\Pi(f(x_*) \mid \mathcal{D}_{s2}^{(n_2)}) = \mathcal{N}(K_{*x}\Lambda Y, K_{**} - K_{*x}\Lambda K_{x*})$$
$$\text{where} \quad L = (K_{zz} + \nu I)^{-1}K_{zz}, \ \Lambda = (\lambda I + LK_{xx})^{-1}L.$$

In the above, $K_{(\cdot)}$ denotes the appropriate Gram matrix. As our $K_{zz}$ admits a known low-rank structure: $K_{zz} = \Phi\Phi^\top$, where $\Phi = (\hat{g}_{n_1}^{(1)}(Z), \ldots, \hat{g}_{n_1}^{(m)}(Z), \hat{h}(Z)) \in \mathbb{R}^{n_2 \times (m+1)}$, we use Woodbury identity to obtain $L = \Phi(\Phi^\top\Phi + \nu I)^{-1}\Phi^\top$ which can be computed in $\mathcal{O}(mn^2)$. Another application of Woodbury identity allows the computation of $\Lambda$ in $\mathcal{O}(mn^2)$ time [26, Appendix D.3]. Applying Nyström approximation to $k_x$ would improve the complexity to $\mathcal{O}(m^2n)$ [19], but we find the above procedure sufficient for our purposes.

**Hyperparameter Selection for Instrument Learning**  In principle, the regression oracle may conduct hyperparameter selection by further splitting its observed dataset, and perform cross validation. We can also compare different oracles on the said validation set. In practice we construct a heldout set using $\mathcal{D}_{s2}^{(n_2)}$ and $g^{(1)}, \ldots, g^{(m)}$ for the NN-based oracle, and use it for early stopping. For selecting the best oracle (which includes the selection of NN activation functions, optimizers, etc., in our setting), however, we find it slightly better to draw additional GP samples $g_v^{(1)}, \ldots, g_v^{(m_v)}$ and evaluate the different oracles by using the resulted kernel $\tilde{k}_z$ on the dataset $\{(z_i, g_v^{(j)}) : i \in [n_2], j \in [m_v]\}$. This can be viewed as a task generalization loss in the terminologies of multi-task learning (Sec. 6), and is more directly connected to the role of $\tilde{\mathcal{I}}$ in IV regression. It also allows data-dependent selection for $m$, although we use fixed choices of $m$ for simplicity.

Following the discussion in the last paragraph, we further guard against potential misspecification of $\mathcal{H}$, by adding to the validation statistics the regression error on $\{(z_i, y_i) : i \in [n_2]\}$. In summary, our

---

[25]More sophisticated sampling schemes exist, e.g., exact Matérn GP samples can be obtained using its state-space represetation [89]. But this is unnecessary for an one-off operation, where we can simply set the number of random features to be sufficiently large.

[26]for the latter, note the truncation of $\hat{h}$, which ensures that $\|g\|_\infty \lesssim \|g\|_{\tilde{\mathcal{I}}} \log n_1$ continue to hold.

first stage validation statistics is

$$m_v^{-1} \sum_{j=1}^{m_v} \mathsf{KRRTestMSE}(\{(z_i, g_v^{(j)}) : i \in [n_2]\}; \tilde{k}_z) + \mathsf{KRRTestMSE}(\{(z_i, y_i) : i \in [n_2]\}; \tilde{k}_z).$$

(71)

**Second-Stage Model Selection**    We provide the following expression for the log marginal quasi-likelihood:

$$
\begin{aligned}
\log \int \Pi(df) \exp(-\lambda^{-1} n_2 \ell_{n_2}(f)) &= \log \int \Pi(df) \exp\left(-\frac{1}{2\lambda}(Y - f(X))^\top L(Y - f(X))\right) \\
&= -\frac{1}{2}\left(Y^\top \Lambda Y + \log|\lambda^{-1}(L^{1/2} K_{xx} L^{1/2} + \lambda I)|\right) + \text{const}
\end{aligned}
$$

(72)

$$\text{where} \quad L = (K_{zz} + \nu I)^{-1} K_{zz}$$

$$\text{and} \quad \Lambda = (\lambda I + L K_{xx})^{-1} L.$$

In the above, $K_{xx}, K_{zz}$ denote the Gram matrices using $\mathcal{H}, \tilde{\mathcal{I}}$, respectively, and the first equality above can be found in [26]. Similar to [26] we exploit low-rank structures in $\Lambda$ and $L$ to accelerate computation, but with Nyström approximation replaced by the feature representation of $\tilde{\mathcal{I}}$.

Note that we cannot use the above marginal quasi-likelihood cannot to select $\lambda$; this is evident from the first expression $\int \Pi(df) e^{-\lambda^{-1} n_2 \ell_{n_2}(f)}$, as the log quasi-likelihood $\ell_n$ does not contain $\lambda$. This is a small price we pay for the simplified analysis: observe $\ell_n(f) = \sup_{g \in \tilde{\mathcal{I}}} \sum_{i=1}^{n_2} 2(f(x_i) - y_i)g(z_i) - g(z_i)^2 - \nu \|g\|_{\tilde{\mathcal{I}}}^2$ is equivalent to $\inf_{g \in \tilde{\mathcal{I}}} \sum_{i=1}^{n_2} (f(x_i) - y_i - g(z_i))^2 + \nu \|g\|_{\tilde{\mathcal{I}}}^2$ up to a Gaussian complexity term. The latter allows for a more complete quasi-Bayesian data generating process as in [29], to which we can add a hyperprior for $\lambda$ as in [29]. However, bounding the extra term would involve additional technicalities. We consider it beyond the scope of this work.

Once $\lambda \asymp 1$ is fixed, however, both versions of marginal likelihood should allow for principled selection of other hyperparameters, as both choices for $\ell_{n_2}$ can be viewed as approximating $\lambda^{-1} n_2 \|E(f - f_0)\|_2^2$; although in our setting, theoretical guarantees have only been established for the former choice. As for $\lambda$, we fix $\lambda = \frac{1}{n_2} \sum_{i=1}^{n_2} (y_i - \hat{h}(z_i))^2$, where $\hat{h}$ is an estimate for $\mathbb{E}(\mathbf{y} \mid \mathbf{z} = \cdot)$; this can be viewed as an *empirical Bayes* counterpart for the choice in [29]. Across all settings we find the selected value to be fairly stable. This is different from other hyperparameters such as kernel bandwidth or variance, for which careful likelihood-based selection is more needed.

# G    Extension for High-Dimensional Exogenous Covariates

In many applications we have access to additional exogenous covariates $\mathbf{w}$, which are independent with the unobservable confounder $\mathbf{u}$ [35, 6]. Denote the original instrument and treatment as $\mathbf{z}_o, \mathbf{x}_o$, the true outcome function then satisfies $\mathbb{E}(f_0(\mathbf{x}_o, \mathbf{w}) - \mathbf{y} \mid \mathbf{z}_o, \mathbf{w}) = 0$, meaning that such $\mathbf{w}$ can then be appended to both $\mathbf{x}$ and $\mathbf{z}$ in the formulation (1).

When both $\mathbf{x}_o$ and $\mathbf{w}$ have moderate dimensions, the prior knowledge about $f_0$ can often be characterized by a product kernel $k_x(x, x') := k_{x_o}(x_o, x'_o) k_w(w, w')$, where $k_w$ can be predetermined. In this case, Algorithm 1 will continue to apply. This connects to the classical tensor product basis model [e.g., 25], as the Mercer basis of $k_x$ equals $\{\psi_{(i,j)}^x = \psi_i^{x_o} \otimes \psi_j^w\}$. The inclusion of $\mathbf{w}$ breaks our assumption by making $E$ non-compact. However, it is common in theoretical works to ignore it for brevity [4, 35, 90, 24], also because $\mathbf{w}$ does not suffer from the ill-posedness issue, and thus is intuitively easier to handle.

When $\mathbf{w}$ is high-dimensional, it can be difficult to prescribe a correctly specified $k_w$ with low complexity. However, it is often realistic that there exist some low-dimensional informative features $\bar{w} = \Phi_w(w),$[27] in which case it is natural to consider kernels of the form $k_{x_o}(x_o, x'_o) k_w(\Phi_w(w), \Phi_w(w'))$, where $k_{x_o}$ is still available *a priori*, but $k_w$ and $\Phi_w$ needs to be learned. In this section we describe a

---

[27]Otherwise it would be unclear if efficient estimation is still possible at all.

simple algorithm, inspired by [17], which implicitly models such a $k_w$ with flexible NN models. For brevity, we will focus on intuition, and will not provide any formal error analysis. The algorithm will be evaluated in Appendix H.4.

To motivate the algorithm, suppose *for the moment* that we have access to $k_w$ and its Mercer basis. Then any $f \in \mathcal{H}$ can be expressed as $f = \sum_{i=1}^{\infty} \sum_{j=1}^{\infty} a_{ij} \psi_i^{x_o} \otimes \psi_j^w$, and the coefficients $a_{ij}$ should satisfy a fast decay, as determined by the assumed eigendecay of $k_{x_o}$ and the unknown, true $k_w$. It is thus possible to truncate the outer sum at $i = m$, and the truncation error will be vanishing fast for some slowly increasing $m$.

The proof of Theorem 4.1 relied on the fact that $2m$ random GP prior draws approximate the top $m$ Mercer basis well. Thus, we can draw $2m$ samples $f_i^{x_o} \sim \mathcal{GP}(0, k_{x_o})$, and perform an "approximate change of basis":

$$f = \sum_{i=1}^{m} \psi_i^{x_o} \otimes \Big( \sum_{j=1}^{\infty} a_{ij} \psi_j^w \Big) + \Delta_m(f) = \sum_{i=1}^{m} f_i^{x_o} \otimes f_i^w + \Delta'_m(f),$$

In the above, $f_i^w \in L_2(P(dw))$ can be obtained by rotating the functions $\{\sum_{j=1}^{\infty} a_{ij} \psi_j^w\}$,[28] and should have good regularity determined by $k_{x_o}$ and $k_w$. Now, observe that

$$Ef = \mathbb{E}\Big( \sum_{i=1}^{m} f_i^{x_o} \otimes f_i^w \mid \mathbf{z}_o, \mathbf{w} \Big) + E\Delta'_m(f) = \sum_{i=1}^{m} (Ef_i^{x_o}) \otimes f_i^w + E\Delta'_m(f).$$

The approximation error $\|E\Delta'_m(f)\|_2 \leq \|\Delta'_m(f)\|_2$ should continue to be small. We can efficiently approximate $Ef_i^{x_o}$ using the regression oracle, and *parameterize* $Ef$ as

$$(Ef)(z_o, w) \approx \sum_{i=1}^{m} \hat{g}_{n_1}^{(i)}(z_o) f_i^w(w; \theta),$$

where $f_i^w(w; \theta)$ denotes the $i$-th output of a DNN with parameter $\theta$. Since the true $f_0$ should satisfy $(Ef_0)(\mathbf{z}_o, \mathbf{w}) = \mathbb{E}(\mathbf{y} \mid \mathbf{z}_o, \mathbf{w})$, we can use the RHS of the above to regress $\mathbf{y}$, after which the estimate for $f_0$ can be read out by replacing $\hat{g}_{n_1}^{(i)}(z_o)$ with $f_i^{x_o}(x_o)$.

In summary, we constructed an algorithm which *implicitly* models a tensor product kernel. The full algorithm is summarized below:

---

**Algorithm 2** IV regression with learned instruments and exogenous covariates.

---

**Require:** $\mathcal{D}_{s1}^{(n_1)}, \mathcal{D}_{s2}^{(n_2)}$; regression algorithm Regress; treatment kernel $k_{x_o}$; $m \in \mathbb{N}$
1: **for** $j \leftarrow 1$ to $m$ **do**
2: $\quad$ Draw $f_j^{x_o} \sim \mathcal{GP}(0, k_{x_o})$
3: $\quad$ $\hat{g}_{u,n_1}^{(j)} \leftarrow$ Regress($\{((\tilde{z}_{o,i}, \tilde{w}_i), f_j^{x_o}(\tilde{x}_{o,i})) : i \in [n_1]\}$)
4: $\quad$ Define $\hat{g}_{n_1}^{(j)} := \min\{\hat{g}_{u,n_1}^{(j)}(\cdot), C \log m\}$
5: **end for**
6: Let $g(z_o, w; \theta) := \sum_{i=1}^{m} \hat{g}_{n_1}^{(i)}(z_o, w) f_i^w(w; \theta)$, where $(f_i^w)$ denote a multi-output DNN with parameter $\theta$. Optimize $\theta$ using (a possibly regularized variant of) the objective

$$\ell'_{n_2}(\theta; \mathcal{D}_{s2}^{(n_2)}) := \sum_{i=1}^{n_2} (g(z_{o,i}, w_i; \theta) - y_i)^2.$$

7: **return** $\hat{f}_{n_2}(x_o, w) := \sum_{i=1}^{m} f_j^{x_o}(x_o) f_i^w(w; \theta)$.

---

While we will not present a full analysis for brevity, it should be easy to provide faster-rate guarantees on $\|E(\hat{f}_{n_2} - f_0)\|_2$, for the NN model in [10]: it suffices to control the M-estimation error about $\ell'_{n_2}$ with local Rademacher analysis, and combine it with the regression error $\|\hat{g}_{n_1}^{(j)} - Ef_j^{x_o}\|_2$. In principle, uncertainty quantification can be conducted by viewing $\ell'_{n_2}$ as a log quasi-likelihood, and

---

[28] A properly scaled version of $\{\psi_i^{x_o}\}$ is well approximated by a rotated version of $\{f_i^{x_o}\}$ (App. C.2). Thus, $f_i^w$ can be obtained by inverting the rotation on $\{\sum_{j=1}^{\infty} a_{ij} \psi_j^w\}$.

modeling $f$ with a Bayesian neural network (BNN); investigation of posterior consistency may then follow [91].

Our algorithm is inspired by [17] which also employs a tensor product model, but additionally learns the treatment representation. From the theoretical perspective, their added flexibility comes with a hefty price: as noted in [17], it is unclear if their alternating optimization procedure minimizes the empirical risk, and only slow rate convergence has been established for the hypothetical empirical risk minimizer. Intuitively, our method avoids these issues by disentangling (the important parts of) the two stages, and carefully reducing them to regression-like problems. This is also the reason we are able to reduce – in principle – uncetainty quantification to a standard BNN inference problem, which is not possible with the formulations of [17, 19].

# H  Simulations

Code for all experiments can be found at `https://github.com/meta-inf/fil`.

## H.1  Evaluation of Hyperparameter Selection

We first evaluate the instrument learning procedure under a correctly specified second stage.

**Setup**   We set $f_0 \sim \mathcal{GP}(0, k_x)$ and generate $\mathbf{z}$ using NNs, with $N \in \{500, 2500, 5000\}, D \in \{40, 100\}$. $k_x$ is set to a RBF kernel with bandwidth determined set to the median distance between inputs (the "median trick").

For our method, we train DNNs using the square loss and the AdamW optimizer, and perform early stopping based on validation loss. We apply dropout with rate varying in $\{0, 0.05, 0.1, 0.2, 0.4\}$. We vary activation functions in relu, swish and tanh; hidden layers in [100], [100, 100], [100, 100, 100] and [100, 100, 100, 100]; and learning rate in $\{5 \times 10^{-3}, 10^{-3}, 5 \times 10^{-4}, 10^{-4}\}$.

We also include two baselines to verify the need for flexibly learned first stage:

1. A kernelized first stage using the RBF kernel, where the bandwidth is set to $\{1, 2, 4, 8\}$ times the median distance.[29]

2. A linear first stage, with basis set to the output of randomly initialized NNs with the same architecture.

Both baselines use the same correctly specified second stage.

We evaluate each setup on 5 independently generated datasets. For our method, training takes a total of 3.6 hours on 8 RTX 3090 GPUs. On average, each single experiment takes 11 seconds, and (at least) 6 experiments can be run in parallel on a single GPU.

**Results**   We plot the validation statistics against test error $\|\hat{f}_n - f_0\|_2^2$ in Figure 2. We can see that learnable NNs lead to the best performance across all sample size, and the first-stage validation statistics correlates well with test performance. Figure 3 provides further information on the influence

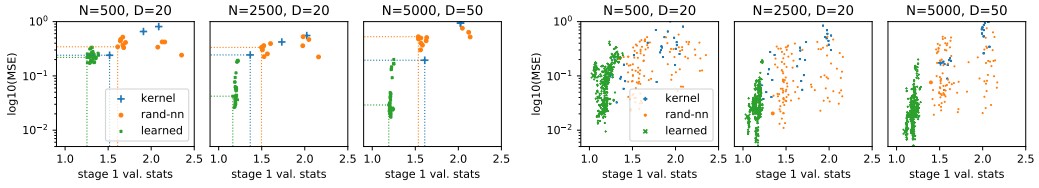

Figure 2: First-stage validation statistics vs. counterfactual MSE for all methods and choices of hyperparameters. Left: visualization of a single run; within each method the model with the best validation statistics is highlighted. Right: aggregated plots for 5 independent runs.

of various NN hyperparameters, by varying one hyperparameters (network activation, architecture,

---

[29]We find that on this dataset, a bandwidth smaller than the median distance is never optimal.

learning rate or dropout rate), and plotting the test and validation statistics with the others chosen to optimize the validation statistics. We can see that network depth and dropout rate have a larger impact.

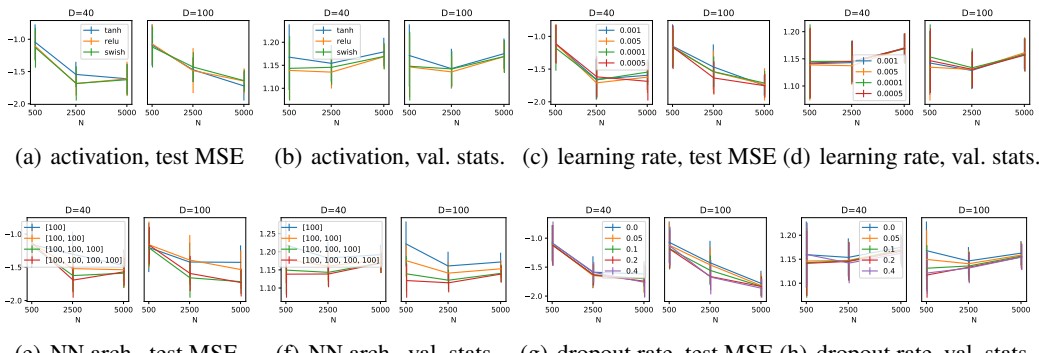

(a) activation, test MSE    (b) activation, val. stats.    (c) learning rate, test MSE (d) learning rate, val. stats.

(e) NN arch., test MSE    (f) NN arch., val. stats.    (g) dropout rate, test MSE (h) dropout rate, val. stats.

Figure 3: Test MSE and validation statistics grouped by different hyperparameters, for the experiment in Section H.1. The test MSEs are plotted in logarithm scale.

## H.2    Predictive Performance

In this section we evaluate the predictive performance of the point estimator.

**Low-dimensional and NN-generated instruments**    We first generate the observed $\mathbf{z}$ from the latent feature $\bar{\mathbf{z}}_1$ based on settings (i-ii) in the main text, and vary $f_0$ in the collection of fixed-form functions in [92]. For baselines, we use adversarial GMM [AGMM, 19] implemented with tree, DNN and RBF kernels. Note that AGMM-RBF is computationally similar to KernelIV [27], as can be seen from the closed-form expressions.

For our method, we determine hyperparameters of the first-stage NN based on the validation results in last subsection's experiments: we use MLPs with hidden layers $\{100, 100, 100\}$ and swish activation, and set the learning rate to $10^{-3}$, and dropout rate to 0.2. We still use a RBF kernel for the second stage, but choose its bandwidth from $\{0.5, 1, 1.5\}$ times the median distance of input, based on the marginal quasi-likelihood.

For baselines, the kernelized version of [19] is implemented as in last subsection, with its first-stage bandwidth determined by the validation statistics (71), and second-stage bandwidth chosen from the same grid as our method. For AGMM-NN and AGMM-tree, we use the official implementation in [19]; as no official instructions for hyperparameter selection are provided, for simplicity, we provide an optimistic estimate of their performance by enumerating hyperparameters in a reasonably defined grid (which always include the default setting in the official code), and reporting the configuration with the best test error. This amounts to

- AGMM-NN: we vary the learning rate hyperparameters in $\{1, 5\} \times \{10^{-3}, 10^{-4}\}$, and the $L^2$ regularization hyperparameters in $\{1, 5\} \times \{10^{-4}, 10^{-5}\}$. We evaluate all 5 variants of NN-based estimators in [19, Fig. 21].
- AGMM-Tree: we vary the depth of the tree in $\{2, 4, 6, 8\}$, and the number of iterations in $\{100, 200, 400, 800, 1600\}$.

For each setup we evaluate on 20 independently generated datasets. For our method, training takes a total of 2.7 hours on 8 RTX 3090 GPUs.

Full results are plotted in Figure 4. All methods have competitive performance in the low-dimensional setting, which is consistent with the report in previous work. As $D$ grows, however, the performance of all baselines worsens, and only the proposed method is able to maintain a similar level of precision, which idicates very good adaptivity. The deterioration is to be expected for the kernelized baselines which do not adapt to any type of informative latent structure. While the tree models may perform variable selection, which can be viewed as adapting to a special type of linear latent structures, less

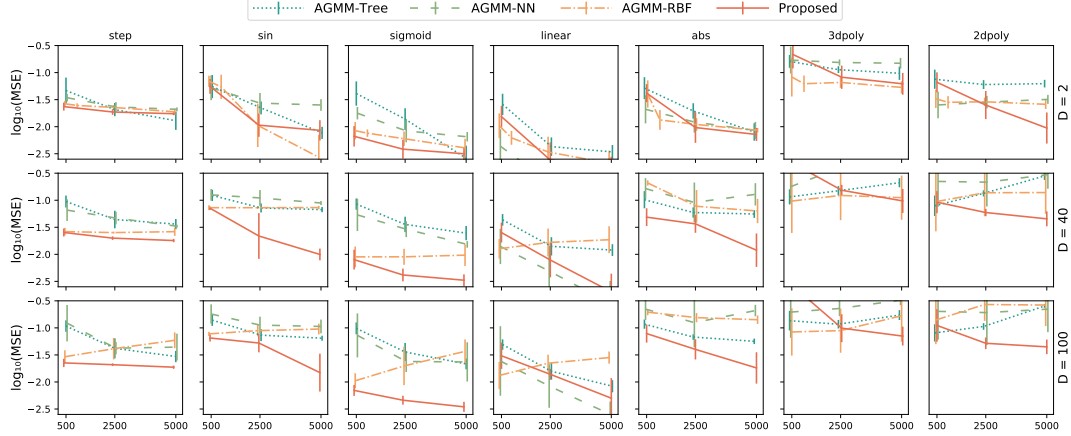

Figure 4: Predictive performance: full results for the low-dimensional and NN-generated instruments. Error bars indicate standard deviation from 20 independent experiments.

.

is known about its adaptivity to nonlinear latent structures. The observed deterioration suggests tree-based models are less competitive in this regime, although it may also be attributed to challenges in minimax optimization, which clearly explains the deteriorated performance of the AGMM-NN baseline.

**Image Instruments**   We now turn to image-based observations. Following previous work [92, 19, 20], we fix $f_0$ to be the abs function, and we map the latent feature $\bar{z}_1 \sim \text{Unif}[-3, 3]$ to $\lfloor \frac{3}{2} z_1 + 5 \rfloor \sim \text{Unif}\{0, \ldots, 9\}$, and use a random MNIST / CIFAR-10 image with the corresponding label as the observed instrument. We expect the MNIST setting to be less challenging than typical high-dimensional feature learning scenarios, as the image label explains a large proportion of variance. It is also known that kernel-based classifiers achieve very good accuracy on MNIST.[30]

We compare with AGMM implemented with DNNs, and the kernelized version of MMR-IV [39]. We use the official implementation of the baselines. For our method, we use a convolutional neural network for instrument learning. Its architecture follows [6], with the exceptions that we double all hidden dimensions and remove the dropout layer. We find the increase in capacity necessary to approximate GP prior draws. We also experimented with a ResNet-18 model which led to similar results. We use the same RBF second stage as before.

We generate $N_1 + N_2 = 10000$ samples. For our method and AGMM-NN, we split the samples evenly. MMV-IV does not require a separate validation set, so we use all generated samples for training.

All methods are evaluated on 10 independently generated datasets. For MMR-IV, its hyperparameter selection procedure is occasionally unstable, so for each randomly generated dataset, we repeat the procedure 20 times from random initial values. For our method, training takes a total of 1 hour on two TITAN X GPUs.

Results are presented in Table 3. We can see that our method is still the most competitive, although the kernel-based MMR-IV also performs well, especially in the MNIST setting.

| Test MSE | AGMM-NN | MMR-IV-Nystrom | Proposed |
|----------|---------|----------------|----------|
| MNIST    | 0.061 [0.056, 0.064] | 0.011 [0.008, 0.018] | **0.008** [0.007, 0.009] |
| CIFAR-10 | 0.117 [0.109, 0.128] | 0.024 [0.013, 0.045] | **0.012** [0.009, 0.013] |

Table 3: Image experiment: median, 25% and 75% percentile of test MSE. **Boldface** indicates the best result ($p < 0.05$ in Mann-Whitney U test).

---

[30]http://yann.lecun.com/exdb/mnist/ reports a test accuracy of $98.6\%$ for an SVM with RBF kernels.

|          | MNIST                  | CIFAR-10               |
|----------|------------------------|------------------------|
| Test MSE | 0.008 [0.007, 0.009]   | 0.012 [0.009, 0.013]   |
| CB. Rad. | 0.011 [0.009, 0.013]   | 0.041 [0.021, 0.060]   |

Table 4: Image experiment: test MSE and radius of the $90\%$ $L_2$ credible ball, for the proposed method. We report median and $25\%$ and $75\%$ quantiles.

### H.3 Uncertainty Quantification

**Setup**   We evaluate the quasi-Bayesian uncertainty estimates in two settings:

1. To rule out the influence from misspecification, we generate $f_0 \sim \mathcal{GP}(0, k_x)$, and construct credible sets using the GP quasi-posterior based on $\mathcal{GP}(0, k_x)$. Note this is still a non-trivial setting, due to the need for instrument learning.

2. For the evaluation of predictive performance, we vary $f_0$ in the collection of functions in [18]. Note the potential misspecification: the RBF kernel is only suited for approximating Hölder regular functions, i.e., functions with bounded (high-order) derivatives [79].
   In this setting we use (quasi-)Bayesian model averaging (BMA), constructed from a grid of GP priors based on RBF kernels $k_x(x, x') = \sigma^2 \exp(-(x - x')^2/2h^2)$, with varying $\sigma$ and $h$.

The observed instrument is generated under the low-dimensional or NN-based setting. For image instruments, our method provides reliable coverage in the setting of Table 3, as shown in Table 4. However, we did not experiment with other choices of $N, D$ or $f_0$ in the image setting, due to the increased computational cost.

Hyperparameters in the instrument learning algorithm are set as in Section H.2. The expression for the marginal likelihood is in (72). For BMA, we consider $\sigma \in \{0.5, 1, 1.5, 2, 2.5, 3\}$ and $h \in \{0.5, 1, 1.5\}$. We reweigh the hyperparameters on the two-dimensional grid, by imposing a (discretized) $\mathrm{InvGamma}(2, 2)$ prior for $\sigma$ and a $\mathrm{Gamma}(2, 1)$ prior for $h$.

As a baseline, we replace the learned first-stage with a fixed-form RBF kernel, with bandwidth determined by the first-stage validation statistics.

For each setup, evaluation is repeated on 300 independently generated datasets in the GP setting, and 20 generations in the setting of [18]. For our method, training takes a total of 1.8 hours in the GP setting, and 2.7 hours in the setting of [18]. The experiments are conducted on 8 RTX 3090 GPUs.

**Results**   Full results in the correctly specified setting are reported in Table 5. As we can see, both methods have similar behaviors in the low-dimensional setting. However, in high dimensions, only the proposed method produces reliable uncertainty estimates with the correct nominal coverage, whereas using a fixed-form kernel as first stage leads to undercoverage and lareger credible sets. The latter observation can be understood from the correspondence between the quasi-posterior marginal variance, and a certain worst-case prediction error on a simplified data generating process [26, Section 5]. The linear IV literature has also related the use of first stage models with insufficient predictive power with IV regression given weakly informative instruments [see e.g., 93].

For BMA, we visualize the resulted quasi-posterior in Figure 5, and report the radius of $90\%$ $L_2$ credible ball in Figure 6. We can see that using learned instruments lead to sharper uncertainty estimates than using a fixed-form kernel for first stage. The resulted uncertainty estimates are still slightly conservative when the model is correctly specified (e.g., sin or linear), or the misspecification is mild (e.g., abs); but under-coverage can occur in the presence of more severe misspecification, such as the step design, or the two polynomial designs. For the polynomial designs, note that our data distribution for $\mathbf{x}$ has a gaussian-like tail, so their unbounded growth of function values and derivatives can be problematic. Preliminary experiments show that polynomial kernels leads to significantly better marginal likelihood on the polynomial functions than the RBF kernels, and improved coverage and MSEs, without affecting the other designs. In aggregate, these results highlight the need for a correctly specified model for reliable uncertainty quantification.

| $D$ | $N_1 = N_2$ | MSE | 90% CB. Rad. | 90% CB. Cvg. | Avg. 90% CI. Cvg. |
|---|---|---|---|---|---|
| **Proposed** | | | | | |
| | 500 | .069 $\pm$.050 | .140 $\pm$.019 | .930 [.895, .954] | .919 $\pm$.125 |
| 2 | 2500 | .028 $\pm$.020 | .056 $\pm$.007 | .907 [.868, .935] | .914 $\pm$.124 |
| | 5000 | .020 $\pm$.014 | .038 $\pm$.004 | .910 [.872, .937] | .908 $\pm$.124 |
| | 500 | .091 $\pm$.064 | .179 $\pm$.022 | .907 [.868, .935] | .908 $\pm$.132 |
| 40 | 2500 | .033 $\pm$.024 | .067 $\pm$.007 | .910 [.872, .937] | .912 $\pm$.137 |
| | 5000 | .023 $\pm$.014 | .046 $\pm$.004 | .917 [.880, .943] | .908 $\pm$.125 |
| | 500 | .097 $\pm$.065 | .201 $\pm$.025 | .923 [.888, .948] | .915 $\pm$.123 |
| 100 | 2500 | .035 $\pm$.024 | .074 $\pm$.008 | .917 [.880, .943] | .908 $\pm$.127 |
| | 5000 | .024 $\pm$.016 | .049 $\pm$.004 | .920 [.884, .946] | .905 $\pm$.134 |
| **RBF first stage** | | | | | |
| | 500 | .071 $\pm$.052 | .144 $\pm$.021 | .920 [.884, .946] | .920 $\pm$.122 |
| 2 | 2500 | .028 $\pm$.020 | .057 $\pm$.007 | .916 [.879, .943] | .917 $\pm$.123 |
| | 5000 | .020 $\pm$.013 | .039 $\pm$.005 | .917 [.880, .943] | .913 $\pm$.117 |
| | 500 | .124 $\pm$.072 | .218 $\pm$.034 | .870 [.827, .903] | .907 $\pm$.107 |
| 40 | 2500 | .100 $\pm$.069 | .132 $\pm$.020 | .723 [.670, .771] | .837 $\pm$.187 |
| | 5000 | .094 $\pm$.067 | .108 $\pm$.017 | .660 [.605, .711] | .792 $\pm$.211 |
| | 500 | .431 $\pm$.192 | .240 $\pm$.036 | .187 [.147, .235] | .640 $\pm$.191 |
| 100 | 2500 | .176 $\pm$.089 | .175 $\pm$.023 | .517 [.460, .573] | .822 $\pm$.136 |
| | 5000 | .126 $\pm$.072 | .156 $\pm$.019 | .660 [.605, .711] | .855 $\pm$.143 |

Table 5: Full results for single-model uncertainty quantification: test MSE, estimated coverage of 90% $L_2$ credible ball and average coverage of pointwise 90% credible interval. For the CB coverage rate estimate we report its 95% Wilson score interval. For other statistics we report standard deviation. Results averaged over 300 independent runs.

## H.4  Exogenous Covariates

Finally, we evaluate our extended algorithm for exogenous covariates, developed in Appendix G.

**Setup**  We use the setup in [6], which simulates the prediction of airline demand. The structural function is

$$f_0(p, t, s) = 100 + (10 + p)s\psi(t) - 2p, \quad \text{where} \quad \psi(t) = 2\Big[\frac{(t-5)^2}{600} + e^{-4(t-5)^2} + \frac{t}{10} - 2p\Big].$$

The observational distribution is defined as

$$s \sim \text{Unif}\{1, \dots, 7\}, \quad t \sim \text{Unif}[0, 10], \quad (c, v) \sim \mathcal{N}(0, I), \quad p = 25 + (c + 3)\psi(t) + v,$$

$$u \sim \mathcal{N}(\rho v, 1 - \rho^2), \quad y = f_0(p, t, s) + u.$$

In our notations, $x_o = p$ is the treatment, $z_o = c$ is the instrument, and $w = (t, s)$ are the additional exogenous covariates. Following all previous work, we consider two variants:

1. In the *low-dimensional* setting, we directly observe $s$. Following [27, 20, 17] we use a univariate real-valued input as $s$.

2. In the *image* setting, we only observe a high-dimensional surrogate of $s$, defined as a random MNIST image of the respective class.

For our method, in the low-dimensional setting, we use an MLP with hidden layers $[128, 64, 32]$ and swish activation. (The architecture is changed to match [6].) In the high-dimensional setting, we first embed the image feature into a 64-dimensional representation, using ConvNet architecture in Appendix H.2; then we concatenate it with the other inputs and feed into the aforementioned MLP. The other hyperparameters follow the image experiment in Appendix H.2. We conduct early stopping by evaluating the reduced-form prediction error $\ell'_{n_2}$ (see Algorithm 2) on $\mathcal{D}_{s1}^{(n_1)}$.

We compare our method with DeepIV [6], DeepGMM [18], AGMM [19] instantiated with RBF kernel and DNN models, and DFIV [17]. For DeepIV, DeepGMM and DFIV we use the implementation in

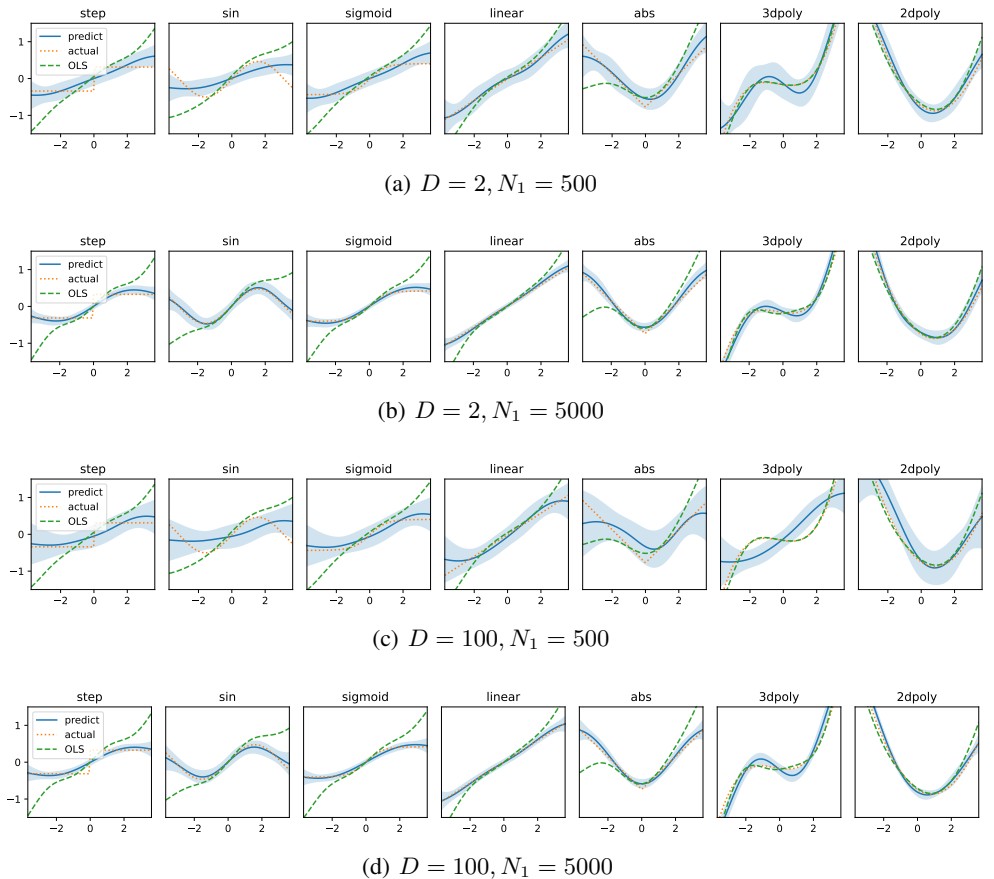

(a) $D = 2, N_1 = 500$

(b) $D = 2, N_1 = 5000$

(c) $D = 100, N_1 = 500$

(d) $D = 100, N_1 = 5000$

Figure 5: Visualization of pointwise 90% credible interval, using BMA and the learned instruments, for varying choices of $N_1 = N_2$ and $D$. OLS denotes a biased regression estimate using KRR.

| $n$ | DeepIV | DeepGMM | AGMM-RBF | AGMM-NN | DFIV | Proposed |
|---|---|---|---|---|---|---|
| Low-dimensional setting | | | | | | |
| 1000 | 3.76 [3.74, 3.77] | 3.97 [3.94, 3.99] | 3.75 [3.71, 3.79] | 3.42 [3.06, 3.99] | 3.00 [2.94, 3.10] | 2.94 [2.85, 3.06] |
| 5000 | 3.14 [3.10, 3.21] | 3.94 [3.91, 3.96] | 3.50 [3.46, 3.52] | 2.74 [2.66, 2.76] | 2.38 [2.31, 2.53] | 2.39 [2.30, 2.47] |
| Image setting | | | | | | |
| 5000 | 3.96 [3.93, 4.01] | 4.41 [4.38, 4.45] | 4.03 [4.02, 4.05] | 4.20 [4.10, 4.33] | 3.83 [3.78, 3.92] | 3.87 [3.85, 3.92] |

Table 6: Demand design: log test MSE vs the total sample size ($n = n_1 + n_2$). We report the median, 25% and 75% percentile over 20 replications.

[17]. For AGMM, we use the implementation in [26], as [19] did not experiment on this dataset. Note that AGMM-RBF has a similar form to KernelIV [27], and our result for it is consistent with [27].

All methods require two independent sets of observations, either directly used in the algorithm or for validation. We partition the training set evenly for this purpose.

All methods are evaluated on 20 independently generated datasets. For our method, all experiments take a total of 6 minutes on 4 Tesla A40 GPUs.

**Results** The results are presented in Table 6. We can see that our method has similar performance to [17], and outperforms the other baselines by a large margin. As discussed before, our method is more appealing than [17] from a theoretical perspective.

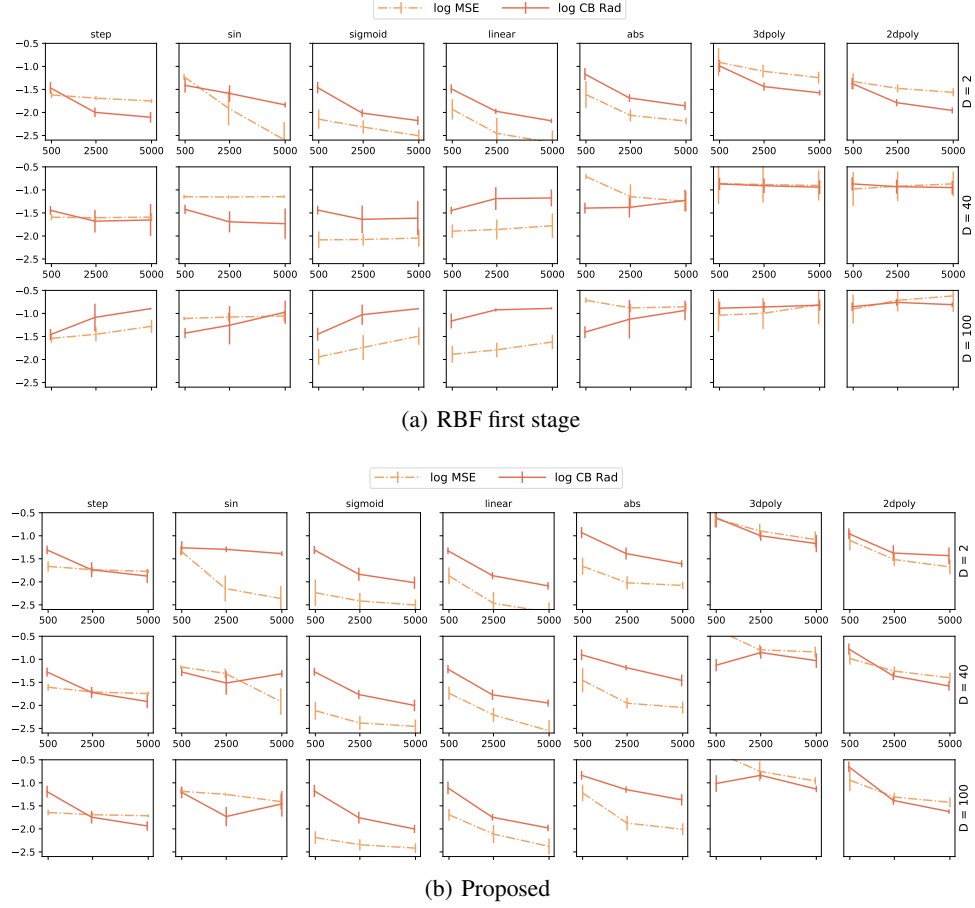

Figure 6: Results for Bayesian model averaging: counterfactual MSE vs the radius of $90\%$ $L_2$ credible ball.