# OpenReview forum: "Fast Instrument Learning with Faster Rates"
_NeurIPS.cc/2022/Conference — NeurIPS 2022 Accept_

### Official Review · Reviewer_Vtb2 · 2022-07-08

**Rating:** 4
**Confidence:** 3
**Soundness:** 2 fair
**Presentation:** 2 fair
**Contribution:** 2 fair

**Summary:**

Fast and provably adaptive instrument learning, through kernel learning with black-box ML models.

**Questions:**

Can you elaborate how this work is not just simply applying the "orthogonal ML" framework?

**Strengths And Weaknesses:**

The paper applies newly developed "orthogonal ML" techniques.

However, the contribution seems rather incremental.

---

> ### Comment · Reviewer_bZoU · 2022-07-28
> **Unqualified Review**
>
> This review seems to be completely unqualified and should not be taken into account.
>
> By no means does the paper apply orthogonal ML techniques.
>
> Orthogonal or double/debiased ML uses (conditional) moment restrictions to enforce a property called Neyman Orthogonality, which, in a two-stage estimation procedure, makes the final estimate robust against estimation errors from the first-stage estimation of high dimensional nuissance parameters.
>
> This paper here formulates the IV problem in terms of conditional moment restrictions (CMR) and shows how to avoid the minimax optimization which is typical of related CMR estimators.
>
> So both rely on conditional moment restrictions but there is no direct connection apart from that and the paper definitely does not apply the orthogonal ML method. CMR have been around for a long time, long before any mention of orthogonal ML in literature.
>
> This review is rather outrageous.

---

> > ### Author Response · Authors · 2022-08-09
> > **Thank you for the clarification**
> >
> > Dear Reviewer bZoU,
> >
> > Thank you very much for the clarification. We highly appreciate that! We are also very happy to see that most of the reviewers (including you) appreciated our contributions and found our response satisfactory.
> >
> > Best,
> > Authors

---

> ### Author Response · Authors · 2022-08-01
> **Authors' Response**
>
> Thank you for your feedback. You are concerned our work is an incremental application of the orthogonal machine learning framework.  Unfortunately, this is a major misunderstanding.  Our method only relates to the double/orthogonal ML methodology in that we both use black-box ML methods to estimate a nuisance parameter. *All theoretical results have been developed separately.*
>
> We have discussed how our work deviates from that literature in L304-308. To further clarify on the issue:
>
> - Our work does not fit into the semi-parametric double ML framework as in [52], because our target parameter, $f_0$, is infinite-dimensional, whereas semiparametric theory requires the target parameter to have fixed dimensionality.
>
> - To our knowledge, only Foster et al [53] studied an extended setting that allows for infinite-dimensional targets. However, *their results do not apply to any existing estimator for nonparametric IV*, as the authors stated in their footnote 1 [53, p.8].
>
> We also make the following observations, which should help to clarify this issue:
>
> 1. Our work does not fit into the double/orthogonal ML framework, because **we do not make use of Neyman orthogonality**, the central idea in that literature.
> We build upon the NPIV estimator in [19]. Consistent with the remarks in [53], we find it dificult, if not impossible, to fit this estimator into the orthogonal learning framework:
>     - To satisfy the requirements in [53], we first need to specify an uncollapsed loss functional $L_{\\mathcal D}$ which our estimator approximately minimizes. The only formulation we can find is to designate as the nuisance parameters $E$ and $g_0:=E f_0$, and define [^1]
> $L\_{\\mathcal D}(f, (\\tilde E, \\tilde g)) := \\|\\tilde E f - \\tilde g\\|\_2^2.$
> This is consistent with our unregularized population loss when the nuisance parameters are correctly specified [19, p.16]. However,
> the orthogonality assumption [53, Asm. 1] is now violated: we have
> $D_{\\tilde E} D_{f} L_{\\mathcal D}(f_0, (E, g_0))[f-f_0,\\tilde E - E] =\\langle(\\tilde E-E) f_0, E(f-f_0)\\rangle_2 \\ne 0,$
> for general choices of $f,\\tilde E$.
>
>     - Two "obvious" alternative choices of $L\_{\\mathcal D}$ also fail to satisfy the requirements: *(i)* if we use $L\_{\\mathcal D}(f, \\tilde E)=\\|\\tilde E(f-f_0)\\|\_2^2$, it would be difficult to bound the $\\mathrm{Rate}\_{\\mathcal D}$ quantity in [53]; intuitively it is because this choice ignores the interaction between confounding and estimation error in $\\tilde E$. *(ii)* We cannot designate as the nuisance parameter the linear map that takes $n$ observations $\\{f(x_i)-y_i:i\\in[n]\\}$ to an estimate of $E(f-f_0)$, because the nuisance parameter must take value in a fixed space independent of $n$.
>
> 2. Estimation of an infinite-dimensional, operator-valued nuisance parameter requires non-trivial effort. Our proofs make heavy use of properties of kernel and Gaussian process models. Besides, they are only made possible by a carefully designed algorithm, which uses randomly GP samples as the regression target (L63-66).
>
> 4. When the target parameter is infinite-dimensional, previous work [53] -- even though they do not apply in our setting -- could only provide estimation guarantees, whereas our results also lead to guarantees for uncertainty quantification.
>
> We will revise the discussion on related work, to incorporate and better emphasize the points above.
>
> [^1]: Note this is a valid population objective, because [53] only requires $f_0$ to minimize the objective when the nuisance parameters are fixed to their true values. Also note that while our finite-sample objective only appears to maintain a single $g = \tilde E(f-f_0)$, we can separate $\tilde E f$ and $\tilde E f_0$ since our $\tilde E$ is a linear map.

---

> > ### Comment · Area_Chair_KyKy · 2022-08-06
> > **Discussion needed**
> >
> > Dear Reviewer Vtb2,
> >
> > The authors have provided a detailed rebuttal to your question on how this work differs from the orthogonal ML framework. Please kindly react to the rebuttal provided by the authors.
> >
> > Best,
> > AC

---

> ### Author Response · Authors · 2022-08-05
> **Looking forward to further feedback**
>
> Dear Reviewer Vtb2,
>
> Thank you again for the great efforts and valuable comments. We have carefully addressed the main concerns in detail. We hope you might find the response satisfactory and appreciate our contributions as the other reviewers. As the discussion phase is about to close, we are very much looking forward to hearing from you about any further feedback. We will be very happy to clarify further concerns (if any).
>
> Best,
> Authors

---

### Official Review · Reviewer_PEsV · 2022-07-10

**Rating:** 6
**Confidence:** 2
**Soundness:** 3 good
**Presentation:** 3 good
**Contribution:** 3 good

**Summary:**

The authors proposed to perform the first-stage instrument learning via an adaptive regression on random draws from a GP prior. And they use kernelized IV  methods to estimate the second stage casual effect functions. They show that the algorithm has faster convergence rates and avoids expensive minimax optimization procedures. For kernel first-stage models, the algorithm can also be applied to evaluate quasi-posterior and thus provide uncertainty quantification.

**Questions:**

I didn’t check the technical details. But should the condition of $n_1$ in line 268 be $n_1^{-(b+2p)/(b+2p+1)}$?

The results of the paper hinge on the assumption that $f_0$ can be characterized by a RKHS $\mathcal H$. Can the authors elaborate more on how practical is this assumption and what are the consequences if the model is mis-specified or the assumption is violated?


**Limitations:**

The authors have included discussions on limitations and future directions in their paper.

**Strengths And Weaknesses:**

The presentation of the paper is good, with clear explanations of the intuitions behind the conditions and assumptions. The algorithm itself is easy to implement and efficiently distills informative structures of the instruments. The first-stage can accommodate flexible ML methods. It also enables fast quasi-Bayesian uncertainty quantification with improve flexibility. They also provide extensive discussion on the connection to related work.

The content of the paper is very compact due to page limits. One may wish to see more on the simulation study in the main manuscript. It would be interesting see low-dimensional sparsity structure design.

---

> ### Author Response · Authors · 2022-08-01
> **Authors' Response**
>
>
> Thank you for your positive and constructive feedback. We address your questions below.
>
> 1. **Typo on L268**: You are correct. Thanks for spotting this. We'll correct it and thoroughly check every detail.
>
> 2. **RKHS assumption on $f_0$**: Thanks for the suggestion. We will add more discussion along the following lines in the final version:
>     - The assumption is usually practical when $x$ has moderate dimensions: e.g., when $f_0$ satisfies certain differentiability conditions, we can choose $\mathcal H$ as a suitable Matern RKHS, as discussed in Example A.1-A.2.
>     - Note that we can relax the requirement on $f_0$ by replacing it with some of its approximations, as in Asm. 2.2 (iii) b, and that Appendix G discusses an extension that allows $\mathbf{x}$ to include additional, high-dimensional exogenous covariates.
> On the flip side, the assumption will be harder to justify when the *treatment* is high-dimensional and $f_0$ contains sparsity patterns. However, this is also the case for classical NPIV methods based on Nadaraya-Watson smoothing or orthogonal series.  In fact, kernelized IV methods can be viewed as a generalization of the series model [Singh et al, 27].
>     - When the assumption is violated, estimation error will become suboptimal and uncertainty estimates will become less meaningful. For the latter, you may refer to the analysis in the simplified setting of Bayesian inverse problems [Knapik et al, 41].

---

> > ### Comment · Reviewer_PEsV · 2022-08-09
> > **Thank you for your response.**
> >
> > My questions have been addressed. My rating remains the same.

---

> > > ### Author Response · Authors · 2022-08-09
> > > **Thank you**
> > >
> > > Thank you very much for the feedback. We're happy that you found our response satisfactory.
> > >
> > > Best,
> > > Authors

---

> ### Comment · Area_Chair_KyKy · 2022-08-08
> **response needed**
>
> Dear Reviewer PEsV,
>
> Please kindly respond to the rebuttal provided by the authors and/or engage in the discussion with them. If it addresses your concerns, please react accordingly. Otherwise, please elaborate in your review on why you think the rebuttal/discussion is inadequate. Thank you.
>
> Best, AC

---

### Official Review · Reviewer_Kxn5 · 2022-07-12

**Rating:** 7
**Confidence:** 3
**Soundness:** 4 excellent
**Presentation:** 4 excellent
**Contribution:** 3 good

**Summary:**

This paper proposed an instrumental variable regression algorithm, which authors claimed to allow a faster convergence and the adaptivity to the dimensionality of informative latent features. This paper also presented extensive theoretical analysis to analyze the algorithm and proposed to bridge the notable gap in the literature.

**Questions:**

I have some specific questions on the experiments:
1. Why would you choose a randomly initialized NN as one of the baseline? Could you be more specific on "linear"?
2. Have you compared the time consumption of the proposed algorithm versus the baselines of your choice in different experiments? I am not sure if I see that being addressed in the main paper and appendix.

**Limitations:**

Authors adequately presented the assumptions of the algorithm and all related analysis, and I do not see potential negative societal impact of their work.

**Strengths And Weaknesses:**

The strength of this paper is on proposing the simple yet faster converging algorithm, and it is well-supported by theoretical analysis and numerical experiments. Authors provided an extensive literature review and paved way of a comprehensive understanding towards literature for audience even not familiar with the specific sub-field. The presentation of this paper is very well organized with detailed explanations in steps on the motivation and design of the algorithm. It is also notable that the appendix is very well organized with detailed supplementary materials covering both theoretical analysis and experiments. The weakness is not obvious to me. But, I'd like to point out that I am not an expert in this sub-field. The only major concern that I have is that there might lack significant novelty of this work comparing with existing literature.

---

> ### Author Response · Authors · 2022-08-01
> **Authors' Response**
>
> Thank you for your positive and constructive feedback. We address your questions below.
>
>
> 1. **Simulation setup**:
>     - Regarding baselines, we only compared to the randomly initialized NN baseline in the first experiment; the comparison serves as an ablation study which highlights the benefits of a learned model. For predictive performance, we primarily compare with AGMM [19], a family of methods with strong empirical performance and theoretical guarantees. Appendix H.3 - H.4 also compare our proposed method with [6, 17, 18, 57].
>     - The `linear` design is where the true structural function is linear; more concretely, $f_0(x)=x$ in the data generating process below L330.
>
>
> 2. **Run-time**: the revision will add the following run-time results, for the predictive experiment (Fig. 1) with $N=2500, D=100$. We can see that our method compares favorably to the two adaptive baselines.
>
> | Method          | AGMM-RBF  | AGMM-Tree  | AGMM-NN    | Proposed   |
> | --------------- | --------- | ---------- | ---------- | ---------- |
> | **Run-time / s** | 6.7 (0.1) | 1374 (418) | 303 (15.6) | 25.9 (5.6) |
>
> (All methods are tested on a server with 2 Xeon E5-2683v3 CPUs (28 cores) and 1 GTX Titan X GPU. AGMM-Tree only uses the CPU. The reported run-time is for a single set of hyperparameters. We report the standard deviation estimated from 5 runs.)

---

> > ### Comment · Reviewer_Kxn5 · 2022-08-08
> > **Thank you for the response**
> >
> > My questions were addressed properly. Thank you for the response!

---

> > > ### Author Response · Authors · 2022-08-09
> > > **Thank you**
> > >
> > > Thank you very much for the feedback. We're happy that you found our response satisfactory.
> > >
> > > Best, Authors

---

### Official Review · Reviewer_AFbo · 2022-07-25

**Rating:** 7
**Confidence:** 4
**Soundness:** 4 excellent
**Presentation:** 2 fair
**Contribution:** 4 excellent

**Summary:**

The paper proposes a Gaussian-process-based IV estimation procedure. In a nonparametric IV regression (NPIV) problem, y = f(x) + e, E[e|z]=0, the traditional approach to consists of two stages: estimate x' = E[x|z] (stage 1), then regression y on x' (stage 2). The kernel IV paper [27] proposed to used two pre-specified RKHS for each stage. Here the paper proposed to estimate stage 1 with a procedure based on sampling from the Gaussian process. Concretely, an RKHS is learned for stage 1 instead of being pre-specified. The advantage of such a procedure is to avoid minimax formulation of the problem and provide an inferential procedure due to the Bayesian nature of the method.

**Questions:**

Minor technical questions:

1. In Assumption 2.2, does a smaller b correspond to a smaller (smoother) function space or the opposite?

2. In Line 5 of Algorithm 1, is the estimated $\tilde k_z$ an estimate for the latent space kernel $\bar k_z$ or $k_z$?

**Limitations:**


1. Citations of a few closely related works are outdated, e.g., 5, 20, 27. These works are published in neurips or other conferences and yet only arxiv versions of these works are cited.

2. Overall the presentation of the paper is very dense. It could improve the paper presentation if the authors provide a guideline for the theorems and examples presented in the revised version of the paper.



**Strengths And Weaknesses:**

Strength

The paper provides an extensive overview of related works and rigorous theoretical derivation. The work is original; inferential methods for the nonparametric IV problem are new in the related literature as far as I know. The proposed method is neat and enjoys good theoretical guarantees.

Weakness

The presentation of the paper is a bit dense and could be improved by including an overview of the results in the paper.

---

> ### Author Response · Authors · 2022-08-01
> **Authors' Response**
>
> Thank you for your appreciation of our contributions and the constructive feedback. We address your questions below.
>
>
> 1. **Presentation**: We agree that the presentation of the paper can still be improved, and will incorporate your feedback into the final revision.
>
>
> 2. **Assumption 2.2**: A smaller $b$ corresponds to a larger function space, as the Mercer eigenvalues will have a slower decay. We'll make this more explicit.
>
>
> 3. **Algorithm 1**: $\tilde k_z$ corresponds to $k_z$. Our algorithm does not keep track of the latent space kernel, $\bar k_z$, which is only introduced to illustrate the assumptions and adaptive properties of our approach.
>
>
> 4. **Outdated citations**: Thanks for spotting this. We will check all citations, and replace outdated citations to point to the published version of the works.

---

> > ### Comment · Reviewer_AFbo · 2022-08-06
> > **Thank you for your response**
> >
> > The authors have addressed my issues. I keep my ratings unchanged.

---

> > > ### Author Response · Authors · 2022-08-09
> > > **Thank you**
> > >
> > > Thank you very much for the feedback. We're happy that you found our response satisfactory.
> > >
> > > Best, Authors

---

### Official Review · Reviewer_bZoU · 2022-07-27

**Rating:** 7
**Confidence:** 3
**Soundness:** 3 good
**Presentation:** 3 good
**Contribution:** 4 excellent

**Summary:**

The paper proposes an instrument learning approach for non-linear instrumental variable regression with high dimensional instruments.
State-of-the-art IV estimators with fast convergence rates generally rely on computationally costly minimax procedures. However, assuming that the true function of interest lies in an RKHS, the kernel versions of these estimators often allow for closed form solutions at the cost of using less flexible RKHS models as adversarial/instrument functions compared to e.g. neural networks or random forests. The authors combine the simplicity of the kernel versions of these estimators with the predictive power of modern machine learning models by first learning a kernel whose features are represented by arbitrary ML models and then using this kernel with the kernel-based closed form solution of the SOTA minimax approaches. The paper provides theoretical guarantees and fast convergence rates similar to the ones of the minimax approaches.

**Questions:**

I appreciated if the authors clarified the following points for me:
- Why does regressing $f(x)$ on $z$ provide denoised samples from $\mathcal{GP}(0,k_z)$?
- Why does this procedure focus on the informative latent features?
- How restrictive is the assumption of $f_0$ being an RKHS function in practice?


**Limitations:**

I do not see any immediate potential for any negative societal impact.

**Strengths And Weaknesses:**

As a disclaimer, I was assigned to this paper as an additional reviewer very shortly before the start of the rebuttal period. In the interest of not further delaying the process I was only able to spend a short amount of time on preparing the review and therefore might have overlooked some important aspects or errors. I have not checked the proofs and my review is based on the assumption that the theorems are correct.


Strengths
- The problem the paper solves is a very interesting and relevant one and to the best of my knowledge the approach is new
- While there is a growing literature on addressing conditional moment restrictions via minimax formulations, approaches like the proposed one that avoid the rather involved saddle-point optimization problems seem to be very useful and well-motivated
- While the final estimator builds on the kernel-kernel version of the minimax formulation of [19], the kernel over the instrumental variables is formed from arbitrary ML models and thus the proposed approach can be combined with (future) SOTA ML models tailored to the problem at hand
- Large parts of the paper are well and clearly written
- The appendix provides a lot of useful additional information


Weaknesses / Possible improvements

- The clarity of presentation could be improved, from reading the text it was not obvious to me that the proposed algorithm is the one described in Algorithm 1
- Without being familiar with the work in reference [19] and their closed form solution for the kernel version, it might be difficult to understand what problem the paper tries to solve and how the final estimator is obtained. The explanation in Appendix F focusses on the more involved estimator based on [26]. I think it would be helful also to discuss the simpler implementation of [19].
- I did not understand why regressing $f(x)$ on $z$ provides denoised samples from $\mathcal{GP}(0,k_z)$, however this is a crucial part of the method, so I think it should be explained more clearly
- Connected to the previous point, it is also not clear to me why this procedure focusses on the most informative latent features
- The paper has a lot of content and might feel a bit too dense. To some extent the methods based on [19] and [26] seem to be independent and the paper might be easier to understand if one of the methods were moved to the appendix (or to a second paper) but this impression might be owed to the limited time I have spent on the paper
- The experiments show that the method performs sometimes better and sometimes on par with the minimax approach of [19]. I think it would be insightful to report e.g. the computation time, to see if the supposedly simpler computations of the proposed method actually result in faster estimation

---

> ### Author Response · Authors · 2022-08-01
> **Authors' Response**
>
>
> Thank you for your positive and constructive feedback. We will carefully consider the presentation issues you have raised in preparing the final revision. We address your questions below.
>
> ---
>
> **Q1.** Discuss the implementation of [19]:
>
> **A1.** Thanks for the suggestion. The kernelized point estimators in [19] and [26] are equivalent up to the choices of constants (L231), so it suffices to discuss one of them. We will revise Appendix F to further clarify that the closed-form expression listed there can be modified to cover the estimator in [19].
>
> ---
>
> **Q2.** Why regressing $f(x)$ provides denoised samples from $GP(0, k_z)$:
>
> **A2.** Regressing $f(x)$ against $z$ provides an approximation to $E f = \mathbb{E}(f(\mathbf x)\mid\mathbf z=\cdot)$, as the latter minimizes the population loss for least square regression. As we proved in Lemma 3.1, for $f\sim GP(0, k_x)$, $E f$ distributes as $GP(0, k_z)$, up to null sets. Thus, regressing $f\sim GP(0,k_x)$ against $z$ provides an approximate sample from $GP(0, k_z)$.
> We refer to the regression procedure as "denoising", because it (approximately) removes the unpredictable term of $f(\mathbf{x})-(Ef)(\mathbf{z})$, which can be viewed as noise (L144-145). We'll make this more explicit.
>
> ---
>
> **Q3.** Why our method adapts to the informative latent structure:
>
> **A3.** This claim is ultimately supported by the comparison of our rate in Proposition 5.1 with that of kernelized IV with fixed-form kernels (L251-254), and the observation that the improvement comes from the adaptivity of the black-box regression algorithm (Ex 4.1).
> Section 3-4 provided intuition on why the improvement over fixed-form, non-adaptive kernels had been possible. We summarize the discussions here and will make them more explicit in the final version:
> 1. Example 3.1 shows the complexity of the optimal first-stage model, $\mathcal I$, only depends on the informative latents.
> 2. Section 4 show that the complexity (Prop. 4.2) and approximation error (Thm. 4.1) of the learned $\tilde{\mathcal I}$ additionally depend on the error rate $\xi_{n_1}$ of the regression oracle, which, as we show in Example 4.1, will adapt to the informative latent structure if we use DNN oracles.
> Thus, we can *intuitively* understand that using $\tilde{\mathcal I}$ instead of $\mathcal I$ would incur little extra error, and the combined error should still be adaptive to the informative latent structure.
> 3. As it is not immediately clear how much improvement -- over non-adaptive models -- has been quantified by these general results, we first instantiate them in Example 4.2, which considers a simpler regression task in the same informative latent setting as Example 4.1. The resultant rates can outperform fixed-form kernels by a large margin (L225-226).
>
> ---
>
> **Q4.** How restrictive the RKHS assumption on $f_0$ is:
>
> **A4.** The revision will add more discussion along the following lines:
> - The assumption should not be restrictive when $x$ has moderate dimensions: e.g., when $f_0$ satisfies certain differentiability conditions, we can choose $\mathcal H$ as a suitable Matern RKHS, as discussed in Example A.1-A.2.
> - Note that we can relax the requirement on $f_0$ by replacing it with some of its approximations, as in Asm. 2.2 (iii) b, and that Appendix G discusses an extension that allows $\mathbf{x}$ to include additional, high-dimensional exogenous covariates. On the flip side, the assumption will be less reasonable when the *treatment* is high-dimensional and $f_0$ contains sparsity patterns. However, this is also the case for classical NPIV methods based on Nadaraya-Watson smoothing or series models.  In fact, kernelized IV methods can be viewed as a generalization of the series model [Singh et al, 27].
>
> ---
>
> **Q5.** Computation time:
>
> **A5.** The revision will add the following run-time results, for the predictive experiment (Fig. 1) with $N=2500, D=100$. We can see that our method compares favorably to the two adaptive baselines.
>
> | Method          | AGMM-RBF  | AGMM-Tree  | AGMM-NN    | Proposed   |
> | --------------- | --------- | ---------- | ---------- | ---------- |
> | **Run-time / s** | 6.7 (0.1) | 1374 (418) | 303 (15.6) | 25.9 (5.6) |
>
> (All methods are tested on a server with 2 Xeon E5-2683v3 CPUs (28 cores) and 1 GTX Titan X GPU. AGMM-Tree only uses the CPU. The reported run-time is for a single set of hyperparameters. We report the standard deviation estimated from 5 runs.)

---

> > ### Comment · Reviewer_bZoU · 2022-08-08
> > **Thanks for the clarifications**
> >
> > Thanks for the clarifications. They have confirmed my viewpoint that this paper is a good contribution addressing a relevant and interesting problem. I have raised my score to 7 (Accept).

---

> > > ### Author Response · Authors · 2022-08-09
> > > **Thank you**
> > >
> > > Thank you very much for appreciating our contributions as well as the update!
> > >
> > > Best,
> > > Authors

---

### Meta-Review · Area_Chair_KyKy · 2022-08-24

**Recommendation:** Accept
**Confidence:** Certain

**Metareview:**

An instrumental variable (IV) regression for high-dimensional data is a challenging problem as it involves either a cumbersome two-stage method or a complex minimax procedure. Kernel methods recently arise as promising tools that enable the solutions to be computed in closed form. Unfortunately, the convergence rate of the vanilla estimators is sub-optimal as it depends on the choice of kernel functions on the instruments (i.e., similar to how the second-stage estimation depends on the first-stage estimation). This paper contributes by showing that the convergence rate can be improved if the kernels on instruments are learned from data. The expert reviewers agree that the paper provides a novel approach to learning the instrument kernels, rigorous theoretical analyses, and convincing empirical results. The major limitation, which can be improved in future work, is that most of the theoretical analyses rely on the assumption that the true structural function $f_0$ lies in a reproducing kernel Hilbert space (RKHS).

**Award:**

No

---

### Decision · Program_Chairs · 2022-09-14

Accept